# Generalization, Expressivity, and Universality of Graph Neural Networks on Attributed Graphs

**Levi Rauchwerger[1], Stefanie Jegelka[2,3], Ron Levie[1]**
[1]Technion – IIT, Faculty of Mathematics, [2]MIT, Dept of EECS and CSAIL,
[3]TUM, School of CIT, MCML, MDSI
`levi.r@campus.technion.ac.il`, `stefanie.jegelka@tum.de`,
`levieron@technion.ac.il`

## ABSTRACT

We analyze the universality and generalization of graph neural networks (GNNs) on attributed graphs, i.e., with node attributes. To this end, we propose pseudometrics over the space of all attributed graphs that describe the fine-grained expressivity of GNNs. Namely, GNNs are both Lipschitz continuous with respect to our pseudometrics and can separate attributed graphs that are distant in the metric. Moreover, we prove that the space of all attributed graphs is relatively compact with respect to our metrics. Based on these properties, we prove a universal approximation theorem for GNNs and generalization bounds for GNNs on any data distribution of attributed graphs. The proposed metrics compute the similarity between the structures of attributed graphs via a hierarchical optimal transport between computation trees. Our work extends and unites previous approaches which either derived theory only for graphs with no attributes, derived compact metrics under which GNNs are continuous but without separation power, or derived metrics under which GNNs are continuous and separate points but the space of graphs is not relatively compact, which prevents universal approximation and generalization analysis.

## 1 INTRODUCTION

Graph neural networks (GNNs) have become a widely used tool in science and industry due to their ability to capture complex relationships in graph-structured data. This makes them particularly useful (Zhou et al., 2020) in domains such as computational biology (Stokes et al., 2020; Atz et al., 2021), molecular chemistry (Wang et al., 2023), network analysis (Yang et al., 2023), recommender systems (Fan et al., 2019), weather forecasting (Keisler, 2022) and learnable optimization (Qian et al., 2024; Cappart et al., 2023). As a result, there has been substantial interest in understanding theoretical properties of GNNs, such as expressivity (Xu et al., 2019), stability (Ruiz et al., 2021) or robustness (Ruiz et al., 2021), and generalization (Verma & Zhang, 2019; Yehudai et al., 2020; Oono & Suzuki, 2020; Li et al., 2022; Tang & Liu, 2023; Levie, 2023; Maskey et al., 2022; 2024).

Initial works analyzing expressivity of GNNs focused on the Weisfeiler-Leman (WL) graph isomorphism test as a criterion to distinguish graphs (Xu et al., 2019; Morris et al., 2019; Zhang et al., 2023a; 2024b), others used criteria such as subgraph or homomorphism counts or biconnectivity (Zhang et al., 2024a; 2023b; Chen et al., 2020; Tahmasebi et al., 2023). However, the WL test merely considers distinguishability of graphs, while analyses of robustness or generalization need a metric, i.e., a quantification of similarity. Usually, for graphs, this is a pseudometric, since GNNs typically cannot distinguish all graphs.

In this paper, we define a pseudometric for attributed graphs which is highly related to the type of computation message passing GNNs (MPNNs) perform. Our construction enables a unified analysis of expressivity, universal approximation and generalization of MPNNs on graphs with node attributes. This work extends and unifies several prior approaches and, to the best of our knowledge, this is the first work to enable this analysis in such a general setting. Both generalization and expressivity analysis rely on a few prerequisites that we need to attain via an appropriate (pseudo-)metric: one must identify the finest topology in which (1) MPNNs *separate points*, i.e., for any two different points in space, there exists an MPNN that can distinguish between them; (2) MPNNs are *Lipschitz continuous*; (3) the space of inputs, i.e., attributed graphs, is a *compact space*.

Given the interest in understanding GNN robustness and generalization, a few recent works studied pseudometrics on graphs. Most of them reflect the computation structure of MPNNs in defining the distance, essentially aiming to view graphs from the viewpoint of the GNN. Message passing, when unrolled, leads to a tree-structured *computation tree* rooted at each node for computing the feature of that node. The pseudometrics then define distances between computation trees, often via optimal transport (Wasserstein distance). Chen et al. (2022; 2023) prove that MPNNs separate points and are Lipschitz continuous over the Weisfeiler-Lehman (WL) distance between hierarchies of probability measures. Since the space of graphs is not compact under their metric, they achieve universal approximation by limiting the analysis to an arbitrary compact subspace. To quantify stability and domain transfer of GNNs, Chuang & Jegelka (2022) define the *Tree Mover's Distance* between finite attributed graphs. However, the above pseudometrics on finite graphs do not yield the desired compactness, and hence no universal approximation over the entire space of attributed graphs. Moreover, a robustness-type generalization theorem over the entire space, which depends on the space having a finite covering number, is not attainable. To solve this, we focus on graph limit theories in which the space of graphs is completed to a compact space, i.e., graphon theory. Inspired by Chen et al. (2022), Böker et al. (2023) took the first steps towards solving the aforementioned problem, by extending the expressivity analysis to graphons and using *iterated degree measures* (Grebík & Rocha, 2021) to represent an analog of computation trees and the 1-WL test on infinite objects. While this enables proofs that MPNNs separate points and are Lipschitz over a compact space, and hence have universal approximation, their pseudometric is restricted to graphs without attributes. In contrast, Levie (2023) defined a limit object of attributed graphs, i.e., *graphon-signals*. Their pseudometric, an extension of the *cut distance*, a common metric between attributed graphs, allows to analyze generalization via compactness and Lipschitz continuity, but is too fine to allow MPNNs to separate points, so does not allow universal approximation. The position paper (Morris et al., 2024) identifies these limitations, posing them as open problems. For additional related work, see Appendix B.

In this work, we close these gaps via a unified approach that allows for an analysis of expressivity, universal approximation, and generalization. Inspired by prior works, we too base our pseudometrics on Wasserstein distance (or Prokhorov metric) between distributions of computation tree analogs. To accommodate attributed graphs and graphons, we extend the theory of iterated degree measures – a hierarchy of measures that reflects a computation tree structure – to graphon-signals, and then define an appropriate extension of MPNNs and appropriate distance between our continuous analogs of computation trees. We then prove that our pseudometric leads to a topology with the three desiderata above: (1) MPNNs separate points; (2) MPNNs are Lipschitz continuous; and (3) the input space of attributed graphons is compact. This enables us to invoke the Stone-Weierstrass theorem to show universal approximation for continuous functions on attributed graphons, and hence, graphs. Compactness and Lipschitzness enable a uniform Monte Carlo estimate to compute a generalization bound for MPNNs. Our generalization bound makes no distributional assumptions on the data and number of parameters of the MPNN. Empirically, our pseudometric correlates with output perturbations of the MPNN, allowing to judge stability.

**Contributions.** We propose the first metric for attributed graphs under which the space of attributed graphs is compact and MPNNs are Lipschitz continuous and separate points. Our construction leads to the first theory of MPNNs that unifies expressivity, universality, and generalization on any data distribution of attributed graphs. In detail:

- We show a *fine-grained* metric version of the separation power of MPNNs, extending the results of Böker et al. (2023) to attributed graphs: two graph-signals are close in our metric if and only if the outputs of all MPNNs on the two graphs are close. Hence, the geometry of graphs (with respect to our metric) is equivalent to the geometry of the graphs' representations via MPNNs (with respect to the Euclidean metric).

- We prove that the space of attributed graphs with our metric is compact and MPNNs are Lipschitz continuous (and separate points). This leads to two theoretical applications: (1) a universal approximation theorem, i.e., MPNNs can approximate any continuous function over the space of attributed graphs; (2) a generalization bound for MPNNs, akin to robustness bounds (Xu & Mannor, 2012), requiring no assumptions on the data distribution or the number of GNN weights.

## 2 BACKGROUND

We begin with some background and notation, for additional background and fundamental concepts in topology see Appendix A. An index is available in Appendix N.

**Basic Notation.** Throughout this text, $\lambda$ denotes the *Lebesgue measure* on $[0, 1]$, and we consider measurability with respect to the Borel $\sigma$-algebra. For any metric space $\mathcal{X}$, we denote by $\mathcal{B}(\mathcal{X})$ its *standard Borel $\sigma$-algebra*. Given a measure $\mu$ on $\mathcal{X}$, we define its *total mass* as $\|\mu\| := \mu(\mathcal{X})$. For a standard Borel space $(\mathcal{Y}, \mathcal{B}(\mathcal{Y}))$ and a measurable map $f : \mathcal{X} \to \mathcal{Y}$, we define the *push-forward* $f_*\mu$ of $\mu$ via $f$ as $f_*\mu(\mathcal{A}) := \mu(f^{-1}(\mathcal{A}))$ for any $\mathcal{A} \in \mathcal{B}(\mathcal{Y})$. *Inequality between two measures* $\mu \leq \nu$ on some space $\mathcal{X}$ means that for any set $\mathcal{A} \subseteq \mathcal{X}$, it holds that $\mu(\mathcal{A}) \leq \nu(\mathcal{A})$. Given a vector $\vec{x} = (x_\alpha)_{\alpha \in \Lambda}$, where $\Lambda$ can be any countable set, we denote by $x_{\alpha_0}$ and $x(\alpha_0)$ the element at index $\alpha_0 \in \Lambda$ of $\vec{x}$. For a finite set $\mathcal{A}$, $|\mathcal{A}|$ is the cardinality. For $K \in \mathbb{N}_0$, we denote $[K] = \{0, 1, \ldots, K\}$.

In our notation, a function is denoted by $f : \mathcal{A} \to \mathcal{C}$. When $f$ is evaluated at a point $x \in \mathcal{A}$ we write $f(x)$ or $f_x$. We may also write $f(-)$ or $f_-$, which simply means $f$. We define $\|f\|_\infty := \sup_{x \in [0,1]} |f(x)|$. The *covering number* of a metric space $(\mathcal{X}, d)$ is the smallest number of open balls of radius $\epsilon$ needed to cover $\mathcal{X}$. The notation $\mathcal{K}$ represents any *compact space*, i.e., a topological space $\mathcal{X}$ in which every cover that consists only of open sets of $\mathcal{X}$ has a finite subcover. Note that compact spaces always have finite covering number. We consider throughout the paper the following fixed compact space: $\mathbb{B}_r^d := \{x \in \mathbb{R}^d : \|x\|_2 \leq r\} \subset \mathbb{R}^d$ for a fixed $r > 0$.

**The weak$^*$ Topology.** Let $\mathscr{M}_{\leq 1}(\mathcal{X})$ and $\mathscr{P}(\mathcal{X})$ denote the space of all nonnegative Borel measures with total mass at most one, and the space of all Borel probability measures on $\mathcal{X}$, respectively. We use $C_b(\mathcal{X})$ to denote the set of all bounded continuous real-valued functions on $\mathcal{X}$. We endow $\mathscr{M}_{\leq 1}(\mathcal{X})$ and $\mathscr{P}(\mathcal{X})$ with the topology generated by the maps $\mu \mapsto \int_\mathcal{X} f d\mu$ for $f \in C_b(\mathcal{X})$, called the weak$^*$ topology in functional analysis (Kechris, 2012, Section 17.E), (Bogachev, 2007, Chapter 8). Under this topology, both spaces are standard Borel spaces, and if $\mathcal{K}$ is a compact metric space, then $\mathscr{M}_{\leq 1}(\mathcal{K})$, $\mathscr{P}(\mathcal{K})$ are compact metrizable (Kechris, 2012, Theorem 17.22). See Appendix E.2 for more details. Under the weak$^*$ topology, for a sequence of measures $(\mu_i)_i$ and a measure $\mu$, we have convergence $\mu_i \to \mu$ if and only if $\int_\mathcal{X} f d\mu_i \to \int_\mathcal{X} f d\mu$ for every $f \in C_b(\mathcal{X})$. Similarly, for measures $\mu$ and $\nu$, we have equality $\mu = \nu$ if and only if $\int_\mathcal{X} f d\mu = \int_\mathcal{X} f d\nu$ for every $f \in C_b(\mathcal{X})$.

**Optimal Transport.** We will use Optimal Transport to construct a metric between graphs and between graph limits. *Unbalanced Optimal Transport* (Séjourné et al., 2023), also called *Unbalanced Earth Mover's Distance* and *Unbalanced Wasserstein Distance*, is a distance function defined by the minimal transportation cost between two distributions. The transport is described by a *coupling* $\gamma$ between two measures $\mu$, $\nu$ on measure spaces $\mathcal{X}$, $\mathcal{Y}$, respectively, i.e., a nonnegative joint measure on $\mathcal{X} \times \mathcal{Y}$ such that $(p_\mathcal{X})_*\gamma = \mu$, $(p_\mathcal{Y})_*\gamma \leq \nu$, given that $\|\mu\| \leq \|\nu\|$. Here, $p_\mathcal{X}$ and $p_\mathcal{Y}$ are the projections from $\mathcal{X} \times \mathcal{Y}$ to $\mathcal{X}$ and $\mathcal{Y}$, i.e., to the first and the second component, respectively, $(p_\mathcal{X})_*\gamma(A) = \gamma(A \times \mathcal{Y})$ and $(p_\mathcal{Y})_*\gamma(B) = \gamma(\mathcal{X} \times B)$.

**Definition 1** (Unbalanced Optimal Transport/Wasserstein Distance). *Let $(\mathcal{X}, d)$ be a metric Polish space. The* Unbalanced Earth Mover's Distance *between two measures $\mu, \nu \in \mathscr{M}_{\leq 1}(\mathcal{X}, d)$ is*

$$\mathbf{OT}_d(\mu, \nu) = \inf_{\gamma \in \Gamma(\mu, \nu)} \left( \int_{\mathcal{X} \times \mathcal{X}} d(x, y) d\gamma(x, y) \right) + |\|\mu\| - \|\nu\||,$$

*where $\Gamma(\mu, \nu)$ is the set of all couplings of $\mu$ and $\nu$.*

An intuitive way to think about such couplings is as transportation plans from one distribution of points in a space to another, where the cost is given by the overall travel distance.

**Product Metric.** A *product metric* $d$ is a metric on the Cartesian product of finitely many metric spaces $(\mathcal{X}_1, d_{X_1}), \ldots, (\mathcal{X}_n, d_{X_n})$ which metrizes the product topology (see Appendix A.4.5 and Appendix A.4.2). Here we use product metrics that are defined as the $\ell_1$-norm of the vector of distances measured in $n$ subspaces: $d((x_1, \ldots, x_n), (y_1, \ldots, y_n)) := \|(d_{X_1}(x_1, y_1), \ldots, d_{X_n}(x_n, y_n))\|_1$.

**Graph- and Graphon-Signals.** A graph $G = (V, E)$ consists of a set of nodes (vertices) $V = V(G)$ connected by edges $E = E(G) \subseteq V \times V$. We denote by $\mathcal{N}(v)$ the set of neighbors of $v \in V$. A *graph-signal* (alternatively *attributed graph*) $(G, \mathbf{f})$ is a graph $G$ with node set $V = \{1, \ldots, N\}$, and a signal $\mathbf{f} = (\mathbf{f}_j)_{j=1}^N \in \prod_{j=1}^N \mathbb{B}_r^d$ that assigns the value $\mathbf{f}_j \in \mathbb{B}_r^d$ to each node $j \in \{1, \ldots, N\}$.

A graphon (Lovász, 2012) may be viewed as a generalization of a graph, where instead of discrete sets of nodes and edges, there are an infinite sets indexed by the sets $V(W) := [0, 1]$ for nodes and $E(W) := V(W)^2 = [0, 1]^2$ for edges. A graphon is defined as a measurable symmetric function $W : E(W) \to [0, 1]$, i.e., $W(x, y) = W(y, x)$. Each value $W(x, y)$ describes the probability or intensity of a connection between points $x$ and $y$. The graphon is used to study limit behavior of large graphs (Lovász, 2012). A *graphon-signal* Levie (2023); Rauchwerger & Levie (2025) is a pair $(W, f)$ where $W$ is a graphon and $f : V(W) \mapsto \mathbb{B}_r^d$ is a measurable function with respect to the Borel $\sigma$-algebra $\mathcal{B}(V(W))$. Note that any graph/graphon without a signal can be seen as a graph/graphon-signal by simply setting the signal to the constant map, i.e., $\forall x \in V(W) : f(x) = c$. We denote by $\mathcal{WL}_r^d$ *the space of all graphon-signals*, with signals $f : V(W) \mapsto \mathbb{B}_r^d$.

Any graph-signal can be identified with a corresponding graphon-signal as follows. Let $(G, \mathbf{f})$ be a graph-signal with node set $\{1, \dots, N\}$ and adjacency matrix $\mathbf{A} = \{a_{i,j}\}_{i,j \in \{1,\dots,N\}}$. Let $\{I_k\}_{k=1}^N$ with $I_k = [(k-1)/N, k/N)$ be the equipartition of $[0, 1]$ into $N$ intervals. The graphon-signal $(W, f)_{(G,\mathbf{f})} = (W_G, f_{\mathbf{f}})$ induced by $(G, \mathbf{f})$ is defined by $W_G(x, y) = \sum_{i,j=1}^N a_{ij} \mathbb{1}_{I_i}(x) \mathbb{1}_{I_j}(y)$ and $f_{\mathbf{f}}(z) = \sum_{i=1}^N f_i \mathbb{1}_{I_i}(z)$, where $\mathbb{1}_{I_i}$ is the indicator function of the set $I_i \subset [0, 1]$. We write $(W, f)_{(G,\mathbf{f})} = (W_G, f_{\mathbf{f}})$ and identify any graph-signal with its induced graphon-signal.

**Message Passing Neural Networks.** *Message Passing Neural Networks (MPNNs)* (Gilmer et al., 2017) are a class of neural networks designed to process graph-structured data, where nodes may have attributes. Via a message passing process, MPNNs iteratively update each node's features by aggregating (processed) features from its neighbors. Various aggregation methods exist, e.g., summation, averaging, or coordinate-wise maximum. Our focus is on normalized sum aggregation, which in practice achieves comparable performance to standard sum aggregation (Levie, 2023).

**Computation Trees.** Computation trees capture and characterize local structure of graphs by describing the data propagation through neighboring nodes via MPNNs' successive layers (Morris et al., 2019; Arvind et al., 2020; Garg et al., 2020; Xu et al., 2020). Since graphons can be seen as graphs with uncountable number of nodes, the concept of computation trees can be extended in a natural way to graphons, by recursively defining computation trees as objects composed of a root node and a distribution of sub-trees induced by the node adjacency of the graphon. The resulting hierarchy of probability measures (Chen et al., 2022; Böker et al., 2023) connects to iterated degree measures. These easily integrate with MPNNs and allow us to identify the finest topology in which MPNNs separate points, which is needed to prove a universal approximation theorem for graphons.

**Iterated Degree Measures and the 1-WL Test for Graphons.** Grebík & Rocha (2021) define iterated degree measures (IDMs) to generalize the 1-Weisfeiler-Leman graph isomorphism test (1-WL) (Appendix A.1) and its characterizations to graphons. The 1-WL test performs message passing to uniquely encode (color) the type of computation tree rooted at each node. The collection of trees helps determine graph isomorphism for many pairs of graphs. Initially, the unattributed nodes are indistinguishable, and the test starts with a constant coloring. For graphons, measures replace node colorings, and iterated measures encode computation trees. Analogous to constant colorings, the base measure is defined as $\widetilde{\mathcal{M}}^0 := \{*\}$, where $*$ is any value, e.g., 0. At level $L \geq 0$, the tree is encoded as the product $\widetilde{\mathcal{H}}^L := \prod_{j \leq L} \widetilde{\mathcal{M}}^j$ over levels, and the space of next-level features as a measure over features of level $\leq L$: $\widetilde{\mathcal{M}}^{L+1} := \mathscr{M}_{\leq 1}(\widetilde{\mathcal{H}}^L)$. We endow $\widetilde{\mathcal{H}}^L$ with the product topology and $\widetilde{\mathcal{M}}^{L+1}$ with the weak* topology, as these are natural topologies for product spaces and spaces of measures.

Grebík & Rocha (2021) connect each graphon $W$ with its corresponding node colorings through the maps $\widetilde{\gamma}_{W,L} : [0, 1] \mapsto \widetilde{\mathcal{H}}^L$, which we call *computation iterated degree measures* (computation IDMs). This differs from both Grebík & Rocha (2021) and Böker et al. (2023) use the name IDM for infinite sequences of Borel measures. These maps send each graphon node $x \in [0, 1]$ to its iterated degree measure, i.e., its coloring. We start with a constant map to a tree that encodes a single color, $\widetilde{\gamma}_{W,0} : [0, 1] \mapsto \widetilde{\mathcal{H}}^0$. The "color" $\widetilde{\gamma}_{W,L}(x)$ at level $L > 0$ is a vector, where the $j$-th entry encodes the color of node $x$ after $j$ coloring iterations. That is, letting $\alpha(x)(j)$ be the $j$-th entry of the vector $\alpha(x)$, we set $\widetilde{\gamma}_{W,L}(x)(j) = \widetilde{\gamma}_{(W,f),L-1}(x)(j)$, for every $j < L$. Reflecting the WL-test, the last entry $\widetilde{\gamma}_{W,t}(x)(L)$ is a measure recursively defined as $\widetilde{\gamma}_{W,t}(x)(L)(A) = \int_{\widetilde{\gamma}_{W,L-1}^{-1}(A)} W(x, -) \, d\mu$ for any subset $\mathcal{A} \subseteq \widetilde{\mathcal{H}}^{L-1}$ of the $(L-1)$-th level colors. Namely, we are aggregating colors over the

neighborhood of node $x$, according to the connectivity encoded by $W(x, -)$, obtaining a measure over level $(L - 1)$ colors, analogous to the discrete 1-WL algorithm. Analogously to computation trees, computation IDMs capture the graphons' connectivity. In Section 3, we generalize IDMs to additionally incorporate signal information, thus moving from graphon analysis to graphon-signal analysis. We note that this definition is taken from Grebík & Rocha (2021). It differs from the 1-WL test in Böker et al. (2023), where they do not concatenate all previous colorings.

## 3 GRAPHON-SIGNAL METRICS THROUGH ITERATED DEGREE MEASURES

To analyze expressivity and generalization, we need to define an appropriate metric between attributed graphons. Hence, in this section, we first extend the IDM definition in Section 2 to capture both signal values and graphon topology. We then define distributions of iterated degree measures (DIDMs) and metrics between IDMs and DIDMs. These induce distance measures on the space of attributed graphs/graphons, which are polynomial time computable. We essentially transition from our computation trees that capture only graphon structure to computation trees that capture attributes as well. In Section 4, we will see how this allows us to analyze MPNNs on attributed graphs.

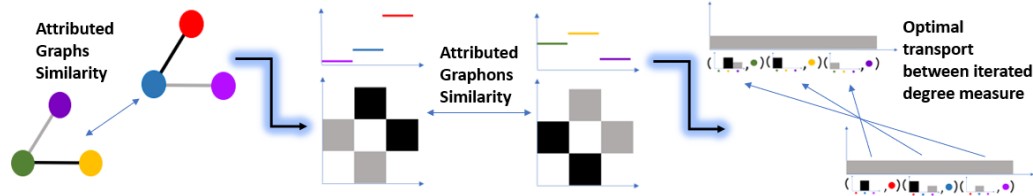

Figure 1: Measuring similarity between graph-signals on the left is translated into measuring similarity between graphon-signals and, lastly, to computing optimal transport between two IDMs, which comprise in the figure of a signal value and a distribution over signal values induced by the graphon's adjacency. Edges colors depict edge weights and node colors depict signal values.

**Computation IDMs and Distributions of IDMs.** We first expand the definitions of IDM and computation IDM of Section 2 from unattributed to attributed graphons. In hindsight, our approach is in line with an idea outlined in Section 5 of Böker et al. (2023). We change the IDMs' base space from a one point space, $\{*\}$, to the space of node attributes $\mathbb{B}_r^d = \{x \in \mathbb{R}^d : \|x\|_2 \leq r\} \subset \mathbb{R}^d$ (for a fixed $r > 0$). Thus, incorporating signal values inherently into the IDMs' structure. Explicitly, we define the *space of iterated degree measures* of order-$L$, $\mathcal{H}^L$, inductively by first defining $\mathcal{M}^0 := \mathbb{B}_r^d$. Then, for every $L \geq 0$, let $\mathcal{H}^L = \prod_{i \leq L} \mathcal{M}^i$ and $\mathcal{M}^{L+1} = \mathscr{M}_{\leq 1}(\mathcal{H}^L)$, where the topologies of $\mathcal{H}^L$ and $\mathcal{M}^{L+1}$ are the product and the weak* topology, respectively. We call $\mathscr{P}(\mathcal{H}^L)$ the *space of distributions of iterated degree measures (DIDMs)* of order-$L$. We denote by $p_{L,j} : \mathcal{H}^L \mapsto \mathcal{H}^j$ and $p_L : \mathcal{H}^L \to \mathcal{M}^L$ the canonical projections where $j \leq L < \infty$. Recall that $\alpha(x)(j)$, refers to the $j$-th entry of the vector $\alpha(x)$. Next, we define a graphon-signal 1-WL analog.

**Definition 2.** *Let* $[0, 1]$ *be the interval with the standard Borel $\sigma$-algebra $\mathcal{B}$ and let $(W, f)$ be a graphon-signal. We define* $\gamma_{(W,f),0} : [0, 1] \to \mathcal{H}^0$ *to be the map* $\gamma_{(W,f),0}(x) := f(x)$ *for every* $x \in [0, 1]$. *Inductively, we define* $\gamma_{(W,f),L+1} : [0, 1] \to \mathcal{H}^{L+1}$ *such that*

*(a)* $\gamma_{(W,f),L+1}(x)(j) = \gamma_{(W,f),L}(x)(j)$, *for every* $j \leq L$ *and*

*(b)* $\gamma_{(W,f),L+1}(x)(L+1)(\mathcal{A}) = \int_{\gamma_{(W,f),L}^{-1}(\mathcal{A})} W(x, -) \, d\mu$, *whenever* $\mathcal{A} \subseteq \mathcal{H}^L$ *is a Borel set.*

*Finally, for every* $L \in \mathbb{N}_0$, *let* $\Gamma_{(W,f),L}$ *be the push-forward of* $\lambda$ *via* $\gamma_{(W,f),L}$. *We call* $\gamma_{(W,f),L}$ *a* computation iterated degree measure (computation IDM) *of order-$L$ and* $\Gamma_{(W,f),L}$ *a* distribution of computation iterated degree measures (computation DIDM) *of order-$L$.*

In Appendix E.2 we prove that the spaces of IDMs and DIDMs are compact, which, together with the continuity of MPNNs (Theorem 13), and the separation power of MPNNs (Theorem 14), allows us to use the Stone-Weierstrass theorem (Appendix A.5) to show universality and to use uniform Monte Carlo estimation to compute a generalization bound for MPNNs.

**Theorem 3.** *The spaces $\mathcal{H}^L$ and $\mathscr{P}(\mathcal{H}^L)$ are compact spaces for any $L \in \mathbb{N}_0$.*

The proof of Theorem 3 inductively uses the fact that given any compact space $\mathcal{K}$, the spaces $\mathscr{M}_{\leq 1}(\mathcal{K})$ and $\mathscr{P}(\mathcal{K})$ endowed with the weak$^*$ topology are compact metrizable spaces. Tychonoff's theorem, which states that the product of any collection of compact topological spaces is compact with respect to the product topology, completes the argument.

**DIDM Mover's Distance.** Next, we define a distance between graphons, viewed as distributions of computation IDMs. Inspired by the tree mover's distance of Chuang & Jegelka (2022), we do this by optimal transport with a ground metric between IDMs, i.e., trees, as both computational IDMs and trees can be seen to represent the MPNNs' computational procees. Thus, we explicitly construct a metric that metrizes the topology of $\mathcal{H}^L$. We define the *IDM distance* of order-0 on $\mathcal{M}^0 = \mathcal{H}^0 = \mathbb{B}_r^p$, for $p \in \mathbb{N}_0$, by $d_{\mathrm{IDM}}^0 := \|x - y\|_2$, and denote by $\mathbf{OT}_{d_{\mathrm{IDM}}^0}$ the optimal transport distance on $\mathcal{M}^1 = \mathscr{M}_{\leq 1}(\mathcal{H}^0)$. We define $d_{\mathrm{IDM}}^L$ recursively as the product metric on $\mathcal{H}^L = \prod_{j \leq L} \mathcal{M}^j$ when the distance on $\mathcal{M}^j = \mathscr{M}_{\leq 1}(\mathcal{H}^{j-1})$ for $0 < j < L-1$ is $\mathbf{OT}_{d_{\mathrm{IDM}}^{j-1}}$. Explicitly written,

$$
d_{\mathrm{IDM}}^L(\mu, \nu) := \begin{cases} \|\mu_0 - \nu_0\|_2 + \sum_{j=1}^L \mathbf{OT}_{d_{\mathrm{IDM}}^{j-1}}(\mu_j, \nu_j) & : \text{if } \infty > L > 0 \\ \|\mu_0 - \nu_0\|_2 & : L = 0 \end{cases}
$$

for $\mu = (\mu_j)_{j=0}^L, \nu = (\nu_j)_{j=0}^L \in \mathcal{H}^L$. The next theorem states that, for every $L \in \mathbb{N}_0$, both the IDM distance $d_{\mathrm{IDM}}^L$ and optimal transport distance $\mathbf{OT}_{d_{\mathrm{IDM}}^L}$ fit naturally to the topologies of IDMs and DIDMs defined in Section 2.

**Theorem 4.** *Let $L \in \mathbb{N}_0$. The metrics $d_{\mathrm{IDM}}^L$ on $\mathcal{H}^L$ and $\mathbf{OT}_{d_{\mathrm{IDM}}^L}$ on $\mathscr{P}(\mathcal{H}^L)$ and $\mathscr{M}_{\leq 1}(\mathcal{H}^L)$ are well-defined. Moreover, $\mathbf{OT}_{d_{\mathrm{IDM}}^L}$ metrizes the weak$^*$ topologies of $\mathscr{M}_{\leq 1}(\mathcal{H}^L)$ and $\mathscr{P}(\mathcal{H}^L)$.*

We now use the distance between IDMs to define a distance between graphons, viewed as distributions of computation IDMs. Specifically, we define the *DIDM Mover's Distance* between graphons as the optimal transport cost between their DIDMs, with ground metric $d_{\mathrm{IDM}}^L(\cdot, \cdot)$:

**Definition 5** (DIDM Mover's Distance). *Given two graphon-signals $(W_a, f_a)$, $(W_b, f_b)$ and $L \geq 1$, the* DIDM Mover's Distance *between $(W_a, f_a)$ and $(W_b, f_b)$ is defined as*

$$
\delta_{\mathrm{DIDM}}^L((W_a, f_a), (W_b, f_b)) := \mathbf{OT}_{d_{\mathrm{IDM}}^L}(\Gamma_{(W_a, f_a), L}, \Gamma_{(W_b, f_b), L}).
$$

Intuitively, $\delta_{\mathrm{DIDM}}^L$ is the minimum cost required to transport node-wise IDMs from one graphon to another.

Given two attributed graphs, $(G, \mathbf{f})$ and $(H, \mathbf{g})$, the DIDM mover's distance, $\delta_{\mathrm{DIDM}}^L((G, \mathbf{f}), (H, \mathbf{g})) := \delta_{\mathrm{DIDM}}^L((W_G, f_{\mathbf{f}}), (W_H, f_{\mathbf{g}}))$ can be computed in polynomial time, as shown next.

**Theorem 6.** *For any fixed $L \in \mathbb{N}_0$, $\delta_{\mathrm{DIDM}}^L$ between any two graph-signals $(G, \mathbf{f})$ and $(H, \mathbf{g})$ can be computed in time polynomial in $L$ and the size of $G$ and $H$, namely $O(L \cdot N^5 \log(N))$, where $N = \max(|V(G)|, |V(H)|)$.*

The theorem is proven in Appendix L.2 with the following reasoning. While $\delta_{\mathrm{DIDM}}^L$ is defined using the induced graphon, it is computed directly on the graph. Moreover, we do not use a data-structure for representing IDMs directly. Instead, each time, we compute cost matrices derived from the cost matrices of the previous layer, avoiding explicit representations of the IDMs. We first compute a cost matrix $D_0$, containing the $L_2$ distances between the nodes' attributes, and then use it as a cost matrix of an OT problem. We repeatedly solve OT problems based on the previous cost matrix and the adjacency matrices of the graphs using linear programming and sum the results with the previous cost matrix to get the next cost matrix. Each OT problem is solved in $O(N^3 \log(N))$ time (Flamary et al., 2021; Chapel et al., 2020). At each step we solve $N^2$ problems. This means $O(N^2 \cdot N^3 \log(N))$ per step. After computing $D_L$, we use it to solve a single OT problem between two uniform distributions, one over $V(G)$ and the other over $V(H)$ to get $\delta_{\mathrm{DIDM}}^L$'s value, which again takes $O(N^3 \log(N))$. Hence, the total computation time is $O(L \cdot N^2 \cdot N^3 \log(N))$. In Section 5, we evaluate the DIDM Mover's Distance empirically.

**Alternative Approach.** For the sake of completeness, we present in Appendix L the Prokhorov metric, which can be used for defining alternative metrics to optimal transport which metrize $\mathcal{H}^L$,

similarly to the constuction of Böker et al. (2023). All of our results can be equivalently stated with the alternative metrics.

## 4   MPNNs and Their Relation to DIDM Mover's Distance

We next integrate MPNNs into our framework and define a general message passing scheme for computing features of attributed graphons, which generalizes standard graph MPNNs. Equivalently, by defining aggregated features via IDMs and DIDMs, we can define the MPNN directly on IDMs and DIDMs. In Appendix G, we show the equivalence between MPNNs defined on graphon-signals and MPNNs defined on IDMs and DIDMs. By analyzing MPNNs on IDMS and DIDMs, we prove Theorem 13 and Theorem 14, which state that two graphon-signals are close to each other in our metrics if and only if the two outputs of any MPNN on the two graphon-signals are close-by in $L_2$ distance.

An MPNN consists of feature initialization, which is a learnable Lipschitz continuous mapping $\varphi^{(0)} : \mathbb{R}^p \mapsto \mathbb{R}^{d_0}$, followed by $L$ layers, each of which consists of two steps: a *message passing layer* (MPL) that aggregates neighborhood information, followed by a node-wise *update layer*. Here, we assume the MPL to be normalized sum pooling when applied on graph-signals or graphon-signals. The update layer consists of a learnable Lipschitz continuous mapping $\varphi^{(t)} : \mathbb{R}^{2d_{t-1}} \mapsto \mathbb{R}^{d_t}$ where $0 < t \leq L$ is the layer's index. Each layer computes a representation of each node. For predictions on the full graph, a *readout layer* aggregates the node representations into a single graph feature and transforms it by a learnable Lipschitz function $\psi : \mathbb{R}^{d_L} \mapsto \mathbb{R}^d$ for some $d \in \mathbb{N}_0$. For the readout, with use average pooling.

**Definition 7** (MPNN Model). *Let $L \in \mathbb{N}_0$ and $p, d_0, \ldots, d_L, d \in \mathbb{N}_0$. We call any collection $\varphi = (\varphi^{(t)})_{t=0}^L$ of Lipschitz continuous functions $\varphi^{(0)} : \mathbb{R}^p \mapsto \mathbb{R}^{d_0}$, and $\varphi^{(t)} : \mathbb{R}^{2d_{t-1}} \mapsto \mathbb{R}^{d_t}$, for $1 \leq t \leq L$, an $L$-layer MPNN model, and call $\varphi^{(t)}$ update functions. For Lipschitz continuous $\psi : \mathbb{R}^{d_L} \mapsto \mathbb{R}^d$, we call the tuple $(\varphi, \psi)$ an MPNN model with readout, where $\psi$ is called a readout function. We call $L$ the depth of the MPNN, $p$ the input feature dimension, $d_0, \ldots, d_L$ the hidden feature dimensions, and $d$ the output feature dimension.*

An MPNN model processes graph-signals as a function as follows.

**Definition 8** (MPNNs on graph-signals). *Let $(\varphi, \psi)$ be an $L$-layer MPNN model with readout, and $(G, \mathbf{f})$ be a graph-signal where $\mathbf{f} : V(G) \mapsto \mathbb{B}_r^p$. The application of the MPNN on $(G, \mathbf{f})$ is defined as follows: initialize $\mathfrak{g}_-^{(0)} := \varphi^{(0)}(\mathbf{f}(-))$ and compute the hidden node representations $\mathfrak{g}_-^{(t)} : V(G) \to \mathbb{R}^{d_t}$ at layer $t$, with $1 \leq t \leq L$ and the graph-level output $\mathfrak{G} \in \mathbb{R}^d$ by*

$$\mathfrak{g}_v^{(t)} := \varphi^{(t)}\Big(\mathfrak{g}_v^{(t-1)}, \frac{1}{|V(G)|} \sum_{u \in \mathcal{N}(v)} \mathfrak{g}_u^{(t-1)}\Big) \quad and \quad \mathfrak{G} := \psi\Big(\frac{1}{|V(G)|} \sum_{v \in V(G)} \mathfrak{g}_v^{(L)}\Big).$$

To clarify the dependence of $\mathfrak{g}$ and $\mathfrak{G}$ on $\varphi$ and $(G, \mathbf{f})$, we often denote $\mathfrak{g}(\varphi)_v^{(t)}$ or $\mathfrak{g}(\varphi, G, \mathbf{f})_v^{(t)}$, and $\mathfrak{G}(\varphi, \psi)$ or $\mathfrak{G}(\varphi, \psi, G, \mathbf{f})$. Here, we use normalized sum aggregation over neighborhoods to be directly compatible with the graphon version. To extend this MPNN to graphons, we transition from a discrete set of nodes to a continuous set by converting the normalized sum into an integral.

**Definition 9** (MPNNs on Graphon-Signals). *Let $(\varphi, \psi)$ be an $L$-layer MPNN model with readout, and $(W, f)$ be a graphon-signal where $f : V(W) \mapsto \mathbb{B}_r^p$. The application of the MPNN on $(W, f)$ is defined as follows: initialize $\mathfrak{f}_-^{(0)} := \varphi^{(0)}(f(-))$, and compute the hidden node representations $\mathfrak{f}_-^{(t)} : V(W) \to \mathbb{R}^{d_t}$ at layer $t$, with $1 \leq t \leq L$ and the graphon-level output $\mathfrak{F} \in \mathbb{R}^d$ by*

$$\mathfrak{f}_x^{(t)} := \varphi^{(t)}\Big(\mathfrak{f}_x^{(t-1)}, \int_{[0,1]} W(x,y)\mathfrak{f}_y^{(t-1)} d\lambda(y)\Big) \quad and \quad \mathfrak{F} := \psi\Big(\int_{[0,1]} \mathfrak{f}(\varphi)_x^{(L)} d\lambda(x)\Big).$$

As before, we often denote $\mathfrak{f}(\varphi)_v^{(t)}$ or $\mathfrak{f}(\varphi, W, f)_v^{(t)}$, and $\mathfrak{F}(\varphi, \psi)$ or $\mathfrak{F}(\varphi, \psi, W, f)$.

The following definition generalizes MPNNs to IDMs and DIDMs using the canonical projections $p_{L,j} : \mathcal{H}^L \mapsto \mathcal{H}^j$ and $p_L : \mathcal{H}^L \to \mathcal{M}^L$, where $j \leq L < \infty$.

**Definition 10** (MPNNs on IDMs and DIDMs). *Let $(\varphi, \psi)$ be an $L$-layer MPNN model with readout. The application of the MPNN on IDMs and DIDMs is defined as follows: initialize $\mathfrak{h}_-^{(0)} := \varphi^{(0)}(-)$,*

*and compute the hidden IDM representations* $\mathfrak{h}_{-}^{(t)} : \mathcal{H}^t \to \mathbb{R}^{d_t}$ *on any order-t IDM* $\tau \in \mathcal{H}^t$, *and the DIDM-level putput* $\mathfrak{H} \in \mathbb{R}^d$ *on an order-L DIDM* $\nu \in \mathscr{P}(\mathcal{H}^L)$, *by*

$$\mathfrak{h}_\tau^{(t)} := \varphi^{(t)} \left( \mathfrak{h}_{p_{t,t-1}(\tau)}^{(t-1)}, \int_{\mathcal{H}^{t-1}} \mathfrak{h}_{-}^{(t-1)} dp_t(\tau) \right) \qquad and \qquad \mathfrak{H} := \psi \left( \int_{\mathcal{H}^L} \mathfrak{h}_{-}^{(L)} d\nu \right).$$

We also denote $\mathfrak{h}(\varphi)_\tau^{(t)}$ and $\mathfrak{H}(\varphi, \psi)$ or $\mathfrak{H}(\varphi, \psi, \nu)$. We name MPNNs' hidden representations and outputs *features*. MPNNs on IDMs and DIDMs are canonical extensions of MPNNs on graphon-signals as follows.

**Lemma 11.** *Let* $(W, f)$ *be a graphon-signal and* $(\varphi, \psi)$ *an L-layer MPNN model with readout. Then, given the computation IDMs* $\{\gamma_{(W,f),t}\}_{t=0}^L$ *and DIDM* $\Gamma_{(W,f),L}$, *we have that* $\mathfrak{f}(\varphi, W, f)_x^{(t)} = \mathfrak{h}(\varphi)_{\gamma_{(W,f),t}(x)}^{(t)}$ *for any* $t \in [L]$, $x \in [0, 1]$. *Similarly,* $\mathfrak{F}(\varphi, \psi, W, f) = \mathfrak{H}(\varphi, \psi, \Gamma_{(W,f),L})$.

That is, graphon-signals' hidden representations are computed through their computation IDMs and DIDMs, i.e., the product of the graphon-signals 1-WL algorithm (Definition 2). Similarly, graph-signals' hidden representations are computed through their induced graphon-signal.

Corollary 12 is a consequence of Theorem 15, one of our main results, presented in Section 5. It reveals a strong connection between MPNN features and the weak* topology of $\mathscr{P}(\mathcal{H}^L)$.

**Corollary 12.** *Let* $L \in \mathbb{N}_0$ *and* $d > 0$ *be fixed. Let* $\nu \in \mathscr{P}(\mathcal{H}^L)$ *and* $(\nu_i)_i$ *be a sequence with* $\nu_i \in \mathscr{P}(\mathcal{H}^L)$. *Then,* $\nu_i \to \nu$ *if and only if* $\mathfrak{H}(\varphi, \psi, \nu_i) \to \mathfrak{H}(\varphi, \psi, \nu)$ *for all L-layer MPNN models* $\varphi$ *with a readout function* $\psi : \mathbb{R}^{d_L} \to \mathbb{R}^d$.

We now present Theorem 13 and Theorem 14. Together, they establishe a bidirectional fundamental connection between our metrics and the outputs of all possible MPNNs, where the second direction is phrased as a delta-epsilon relation. Specifically, Theorem 13 states a Lipschitz bound for MPNNs with respect to the IDM and DIDM mover's distance. This quantifies stability as in the finite case in Chuang & Jegelka (2022).

**Theorem 13.** *Let* $\varphi$ *be an L-layer MPNN model. Then there exists a constant* $C_\varphi$ *that depends only on the number of layers L and the Lipschitz constants of the update functions, such that*

$$\|\mathfrak{h}(\varphi, \alpha)^{(L)} - \mathfrak{h}(\varphi, \beta)^{(L)}\|_2 \le C_\varphi \cdot d_{\mathrm{IDM}}^L(\alpha, \beta)$$

*for all* $\alpha, \beta \in \mathcal{H}^L$. *If* $\varphi$ *has a readout function* $\psi$, *then, for all* $\mu, \nu \in \mathscr{P}(\mathcal{H}^L)$, *there exists a constant* $C_{(\varphi, \psi)}$ *that depends only on* $C_\varphi$ *and the Lipschitz constant of the model's readout function, such that*

$$\|\mathfrak{H}(\varphi, \psi, \mu) - \mathfrak{H}(\varphi, \psi, \nu)\|_2 \le C_{(\varphi, \psi)} \cdot \mathbf{OT}_{d_{\mathrm{IDM}}^L}(\mu, \nu).$$

In our analysis, Theorem 13 is vital for the generalization analysis in Section 5. The following theorem is roughly the "topological converse" of Theorem 13, and is based on Corollary 12.

**Theorem 14.** *Let* $d > 0$ *be fixed. For every* $\varepsilon > 0$, *there are* $L \in \mathbb{N}_0$, $C > 0$, *and* $\delta > 0$ *such that, for all DIDMs* $\mu, \nu \in \mathscr{P}(\mathcal{H}^L)$, *if* $\|\mathfrak{H}(\varphi, \psi, \mu) - \mathfrak{H}(\varphi, \psi, \nu)\|_2 \le \delta$ *holds for every L-layer MPNN model* $\varphi$ *with readout function* $\psi : \mathbb{R}^{d_L} \to \mathbb{R}^d$ *when* $C_{(\varphi, \psi)} \le C$, *then* $\mathbf{OT}_{d_{\mathrm{IDM}}^L}(\mu, \nu) \le \varepsilon$.

Note that the constants $L$, $C$, and $\delta$ are independent of the DIDMs $\mu$ and $\nu$. The combination of Theorem 13 and Theorem 14 implies that we can not only bound MPNNs' output perturbations with $\mathbf{OT}_{d_{\mathrm{IDM}}^L}$, but also estimate closeness in $\mathbf{OT}_{d_{\mathrm{IDM}}^L}$ via MPNNs' output closeness.

**Empirical Evaluation.** As a proof of concept, we empirically test the correlation between $\delta_{\mathrm{DIDM}}^L$ and distance in the output of MPNNs. For the graphs, we use stochastic block models (SBMs), which are random graph generative models. We generated a sequence of 50 random graphs $\{G_i\}_{i=0}^{49}$, each with 30 vertices. Each graph is generated from a SBM with two blocks (communities) of size 15 with $p = 0.5$ and $q_i = 0.1 + 0.4i/49$ probabilities of having an edge between each pair of nodes from the same block different blocks, respectively. We denote $G := G_{49}$, which is an Erdős–Rényi model. We plot $\delta_{\mathrm{DIDM}}^2(G_i, G)$ against distance in the output of randomly initialized MPNNs. We conducted the experiment twice, once with a constant feature for all nodes and once with a signal which has a different constant value on each community of the graph. Each of these two values is randomly sampled from a uniform distribution over $[0, 1]$. Figure 2 shows the results when varying the hidden dimension of the GNN. The results show a strong correlation between input distance and GNN output distance. More empirical results are presented in Appendix M. The code is available at https://github.com/levi776/GNN-G-E-U.

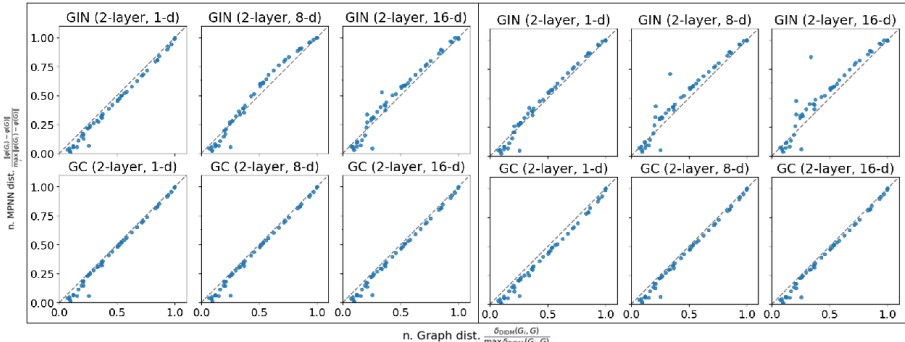

Figure 2: Correlation between $\delta_{\mathrm{DIDM}}^2$ and distance in the output of a randomly initialized MPNN. A convergent sequence of graphs. The graphs are generated by a stochastic block models. In the six leftmost figures the signal is constant. In the six rightmost figures, the signal values are constant each graph's community. Each signal value is sampled from a uniform distribution over $[0, 1]$.

## 5 THEORETICAL APPLICATIONS

Next, we state the main theoretical applications of the compactness of the space of attributed graphs and the Lipschitz and separation power of MPNNs with respect to DIDM mover's distance. First, we show a universal approximation theorem for MPNNs on IDMs and DIDMs, which means that MPNNs can approximate any continuous function on IDMs and DIDMs. This entails universal approximation of MPNNs on attributed graphs/graphons as well, generalizing the results of Böker et al. (2023) to attributed graphs. Second, we introduce a uniform generalization bound for MPNNs.

**Universal Approximation.** We define the set $\mathcal{N}_L^{d_L} := \{\mathfrak{h}(\varphi)_-^{(L)} : \mathcal{H}^L \mapsto \mathbb{R}^{d_L} | \varphi$ is an $L$-layer MPNN model$\} \subseteq C(\mathcal{H}^L, \mathbb{R}^{d_L})$, where $C(\mathcal{H}^L, \mathbb{R}^{d_L})$ is the set of all continuous functions, $\mathcal{H}^L \mapsto \mathbb{R}^{d_L}$. Similarly we define the set $\mathcal{NN}_L^d := \{\mathfrak{H}(\varphi, \psi, -) : \mathscr{P}(\mathcal{H}^L) \mapsto \mathbb{B}_r^d | (\varphi, \psi)$ is an $L$-layer MPNN model with readout$\} \subseteq C(\mathscr{P}(\mathcal{H}^L), \mathbb{R}^d)$.

**Theorem 15** (Universal Approximation). *Let $L \in \mathbb{N}_0$. Then, the set $\mathcal{N}_L^1$ is uniformly dense in $C(\mathcal{H}^L, \mathbb{R})$ and the set $\mathcal{NN}_L^1$ is uniformly dense in $C(\mathscr{P}(\mathcal{H}^L), \mathbb{R})$.*

Combining Theorem 15 with Lemma 11 leads in a very straightforward way to universal approximation of continuous functions from graph-signals to graph-signals embeddings. Our theory implies that if we use all MPNNs, we can distinguish between all attributed graphs with positive graph distance (since they could take different function values). Indeed, Table 1 and 3 in Böker et al. (2023) illustrate that using sufficiently many MPNNs provides enough discriminative power for graph classification tasks on both attributed and unattributed graphs. Our results supply the theoretical background for their experiments with attributed graphs.

Note that Theorem 15 states that any continuous function from DIDMs to scalars can be approximated by an MPNN on DIDMs. To infer a universal approximation result for functions from graph-signals to vectors we emphasize the following considerations. Define the set $\mathcal{NN}_L^d(\mathcal{WL}_r^d) := \{\mathfrak{F}(\varphi, \psi, -, -) : \mathcal{WL}_r^d \mapsto \mathbb{R}^d | (\varphi, \psi)$ is an $L$-layer MPNN model with readout$\}$. First, note that the space of computation DIDMs, $\Gamma_{(W,f),L}(\mathcal{WL}_r^d)$, is a strict subset of $\mathscr{P}(\mathcal{H}^L)$, which is not dense (w.r.t $\delta_{\mathrm{DIDM}}^L$) in view of Theorem 64. Indeed, there are DIDMs that do not come from any graphon-signal, and a closed strict subset cannot be dense. So, the space of DIDMs of graph-signals is also not dense. Hence, Theorem 15 does not imply that any continuous function (w.r.t $\delta_{\mathrm{DIDM}}^L$) from graph-signal to vector can be approximated using an MPNN model. Rather, any function on graph-signals that can be extended to a continuous function on DIDMs can be approximated can be approximated using an MPNN model, which is a weaker form of universality. Fortunately, we can directly prove a universal approximation theorem directly for the space $\mathcal{WL}_r^d$, which in terms gives a universal approximation theorem for continuous functions from graph-signals to vectors by a density argument. For this, in Appendix I.3 we prove that graph-signals are dense in $\mathcal{WL}_r^d$ w.r.t. $\delta_{\mathrm{DIDM}}^L$.

**Theorem 16.** *Let $L \in \mathbb{N}_0$. Then, the set $\mathcal{NN}_L^1(\mathcal{WL}_r^d)$ is uniformly dense in $C(\mathcal{WL}_r^d, \mathbb{R})$.*

**Generalization Bounds for MPNNs.** Consider $K$-class classification, i.e., the data is drawn from a distribution $(\mathcal{X} \times \{0, 1\}^K, \Sigma, \tau)$, where $\Sigma$ is the Borel $\sigma$-algebra and $\tau$ is a probability mea-

sure, where $\mathcal{X}$ is either $(\mathscr{P}(\mathcal{H}^L), \mathbf{OT}_{d_{\mathrm{IDM}}^L})$ or $(\mathcal{WL}_r^d, \delta_{\mathrm{DIDM}}^L)$, and $\mathcal{E}$ is a Lipschitz loss function with a Lipschitz constant $C_{\mathcal{E}}$. Define the formal bias of a function $f : \mathbb{R}^{d_1} \mapsto \mathbb{R}^{d_2}$ to be $\|f(0)\|_2$ (Maskey et al., 2022) and denote the smallest Lipschitz constants of $f$ by $\|f\|_L$ (see Appendix A.12). Fix $L \in \mathbb{N}$ and $A_1, A_2 > 0$. Let $\Theta$ be the set of all $L$-layer MPNN models with readout $\left(\left(\varphi^{(t)}\right)_{t\in[L]}, \psi\right)$ such that the Lipschitz constants $\left\|\varphi^{(t)}\right\|_L$ and $\|\psi\|_L$ are bounded by $A_1$ and the formal biases $\left\|\varphi^{(t)}(0)\right\|_2$ and $\|\psi(0)\|_2$ are bounded by $A_2$. Let $\mathrm{Lip}(\mathcal{X}, C, B)$ be the set of all continuous functions $f : \mathcal{X} \mapsto \mathbb{R}^{d_L}$ with bounded Lipschitz constants $\|f\|_L \leq C$ and bounded norms $\|f\|_\infty \leq B$. In Appendix K.3, we show that there exist $C_\Theta, B_\Theta > 0$ that depend on $L, A_1, A_2$ such that $\Theta \subseteq \mathrm{Lip}(\mathcal{X}, C_\Theta, B_\Theta)$. As a result, if we prove a generalization bound for the hypothesis class $\mathrm{Lip}(\mathcal{X}, C_\Theta, B_\Theta)$, the bound would also be satisfied for the hypothesis class $\Theta$. The following generalization theorem uses techniques from Levie (2023), deriving generalization bounds for Lipschitz continuous functions over domains with finite covering. For us, both the spaces of DIDMs and of graphon-signals have finite covering numbers as they are compact, and MPNNs are Lipschtiz over these domains. The technique is also similar to robustness-type bounds (Xu & Mannor, 2012).

**Theorem 17** (MPNN generalization theorem). *Consider the above classification setting with $\mathcal{X}$ being either $(\mathscr{P}(\mathcal{H}^L), \mathbf{OT}_{d_{\mathrm{IDM}}^L})$ or $(\mathcal{WL}_r^d, \delta_{\mathrm{DIDM}}^L)$. Let $C := C_{\mathcal{E}} \max(C_\Theta, 1)$, $B := C_{\mathcal{E}}(B_\Theta + 1) + |\mathcal{E}(0,0)|$, and $\{X_i\}_{i=1}^N$ be independent random samples from the data distribution $(\mathcal{X} \times \{0,1\}^K, \Sigma, \tau)$. Then, for every $p > 0$, there exists an event $\mathcal{U}^p \subset (\mathcal{X} \times \{0,1\}^K)^N$ regarding the choice of $\mathbf{X} = (X_1, \ldots, X_N)$, with probability $\tau^N(\mathcal{U}^p) \geq 1 - p$, in which for every function $\mathfrak{M}_{\mathbf{X}}$ in the hypothesis class $Lip(\mathcal{X}, C_\Theta, B_\Theta)$, we have*

$$\left| \mathcal{R}(\mathfrak{M}_{\mathbf{X}}) - \hat{\mathcal{R}}_{\mathbf{X}}(\mathfrak{M}_{\mathbf{X}}) \right| \leq \xi^{-1}(N) \left( 2C + \frac{1}{\sqrt{2}} B \left( 1 + \sqrt{\log(2/p)} \right) \right), \tag{1}$$

*where $\xi(\epsilon) = \frac{\kappa(\epsilon)^2 \log(\kappa(\epsilon))}{\epsilon^2}$, $\kappa$ is the covering number of the compact space $\mathcal{X} \times \{0,1\}^K$ and $\xi^{-1}$ is the inverse function of $\xi$.*

Theorem 17 means that when minimizing the empirical risk on a training set drawn from the data distribution, the statistical risk is guaranteed to be close to the empirical risk in high probability, where no assumptions on the data and number of parameters of the MPNN is required (the only assumption is Lipschitz continuity of the update and readout functions). Indeed, the term $\xi^{-1}(N/2C)$ in Equation (1) approaches zero when we take $N$ to infinity. Moreover, since the space $\mathcal{X} \times \{0,1\}^C$ is compact (Theorem 26), for $\mathcal{X} \in \{\mathscr{P}(\mathcal{H}^L), \mathcal{WL}_r^d\}$, its covering number $\kappa$ is finite.

## 6 DISCUSSION

MPNNs were historically defined constructively, as specific types of computations on graphs, without a proper theory of MPNN function spaces over properly defined domains. Our work provides a *functional basis* for MPNNs, which leads to *machine learning results* like universal approximation and generalization for attributed graphs.

The only non-standard part of our construction is the choice of normalized sum aggregation in our MPNN architecture, while most MPNNs use sum, mean, or maximum aggregation. We justify this choice as follows. First, experimentally, in Böker et al., 2023, Tables 1, 3 and Levie, 2023, Table 2, it is shown that MPNNs with normalized sum aggregation generalize well, compared to MPNNs with sum aggregation. We believe that this is the case since in most datasets most graphs have roughly the same order of vertices. Hence, the normalization by $N^2$ is of a constant order of magnitude and can be swallowed by the weights of the MPNN, mimicking the behavior of an MPNN with sum aggregation. Future work may explore extensions of our theory to other aggregation schemes (see Appendix A.3 and Appendix C.2 for more details).

Our theory is meaningful only for dense graphs, as sparse graphs are always considered close to the empty graph under our metrics (see Appendix C.1). Future work may focus on deriving fine-grained expressivity analyses for sparse attributed graphs. Moreover, the current theory is designed around graph level tasks. However, potentially, one can also use IDMs to study node level tasks, as IDMs represent the computational structure corresponding to each node. For example, one can potentially use our construction for analyzing the stability of node-level GNNs to perturbations to the structure of the graph and its features, where the magnitude of the perturbation is modeled via IDM distance. Lastly, since our proofs are specific to MPNNs with normalized sum aggregation, future research could provide extension of our theory to other aggregation functions.

## 7 ETHICS STATEMENT

As this paper mainly focuses on theory, and uses only datasets that do not involve humans or animals, it does not present an apparent ethical problem.

## 8 REPRODUCIBILITY STATEMENT

We hereby declare that the proofs of all theorems presented in the article are available in the appendix. The code we used in this paper will be available upon the acceptance of this paper, with full explanation on how to perform all the experiments present in this article. This means that all the experiments presented both in the main article and the appendix could be reproduced.

## 9 ACKNOWLEDGMENTS

This research was supported by a grant from the United States-Israel Binational Science Foundation (BSF), Jerusalem, Israel, and the United States National Science Foundation (NSF), (NSF-BSF, grant No. 2024660) , by the Israel Science Foundation (ISF grant No. 1937/23), and by an Alexander von Humboldt professorship.

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

## A   ADDITIONAL BACKGROUND

We provide here additional background.

### A.1   WEISFEILER LEMAN GRAPH ISOMORPHISM TEST ON GRAPHS

The Weisfeiler-Leman-1 (WL-1) test, which was developed by Boris Weisfeiler and Andrei Leman, also known as color refinement, is an algorithm which aimed to effectively approximate the solution of the graph isomorphism problem. As graph isomorphism problem is not known to be solvable in polynomial time, the algorithm cannot distinguish all non-isomorphic graphs.

Given a graph $G = (V, E)$ with initial labeling $L_0 : V \mapsto \mathbb{N}_0$ (also referred to as coloring), composed of $V$, a the set of nodes and $E \subseteq V \times V$, a the set of edges, the WL-1 algorithm iteratively updates the nodes labels based on the labels of neighboring vertices.

**Definition 18** (Weisfeiler-Leman Iteration). *Given a graph $G = (V, E)$ with an initial labeling $L_0 : V \to \mathbb{N}_0$, each iteration $t > 0$ of the WL algorithm computes a new labeling $L_t$ as follows:*

$$L_t(v) = Hash\left(L_{t-1}(v), \{\!\{L_{t-1}(u) : u \in \mathcal{N}(v)\}\!\}\right)$$

*where $\mathcal{N}(v)$ denotes the neighbors of vertex $v$, and Hash is an injective function mapping the previous label and multiset of neighbor labels to a new unique label.*

This process runs on two graphs in parallel and continues on until a stable labeling of one of the two graphs is achieved, meaning that for some $t \in \mathbb{N}_0$ the label of one of the graphs doesn't change, or a discrepancy is found between the label of the two graphs being compared. One of these two events is always reached.

### A.2   MESSAGE PASSING NEURAL NETWORKS ON GRAPHS

Message Passing Neural Networks (MPNNs) Gilmer et al. (2017) are a neural networks class designed to process graph-structured data with and without attributes. Given a graph $G = (V, E)$, and a signal $\mathbf{f} : V \mapsto \mathbb{R}^d$ (in the case of a graph without attributes the signal is the constant function $\mathbf{f}(v) = 1$, MPNNs iteratively update node embeddings through the exchange of messages between nodes through edges. We restate here Definition 7.

**Definition 7** (MPNN Model). *Let $L \in \mathbb{N}_0$ and $p, d_0, \ldots, d_L, d \in \mathbb{N}_0$. We call any collection $\varphi = (\varphi^{(t)})_{t=0}^L$ of Lipschitz continuous functions $\varphi^{(0)} : \mathbb{R}^p \mapsto \mathbb{R}^{d_0}$, and $\varphi^{(t)} : \mathbb{R}^{2d_{t-1}} \mapsto \mathbb{R}^{d_t}$, for $1 \le t \le L$, an $L$-layer MPNN model, and call $\varphi^{(t)}$ update functions. For Lipschitz continuous $\psi : \mathbb{R}^{d_L} \mapsto \mathbb{R}^d$, we call the tuple $(\varphi, \psi)$ an MPNN model with readout, where $\psi$ is called a readout function. We call $L$ the depth of the MPNN, $p$ the input feature dimension, $d_0, \ldots, d_L$ the hidden feature dimensions, and $d$ the output feature dimension.*

Let $\varphi$ be an $L$-layer MPNN model. We recall that $\mathcal{N}(v)$ denotes the set of neighbors of node $v$. Each node $v \in V$ starts with an initial feature vector $\mathfrak{g}(\varphi)_v^{(0)} = \mathbf{f}(v)$, which are updated iteratively through the the layer of the message passing model. Each layer consists of two main steps: **aggregation** and **update**.

At each layer $0 < t \le L$, each node $v$ aggregates information from its neighbors by:

$$\mathfrak{m}(\varphi, G, \mathbf{f})_v^{(t)} := \text{Agg}\left(G, \mathfrak{g}(\varphi, G, \mathbf{f})_{-}^{(t-1)}\right),$$

where Agg is a differentiable aggregation function that process the graph and its hidden node representations on each layer. In this article we focus on normalized sum aggregation, which is defined as follows.

**Definition 19** (Normalized Sum Aggregation on Graph-Signals). *Let $(G, \mathbf{f})$ be a graph-signal, then normalized sum aggregation with respect to the $(G, \mathbf{f})$ and a node $v \in V(G)$ is defined as*

$$Agg(G, \mathfrak{g}(\varphi, G, \mathbf{f})_{-}^{(t)})(v) = \frac{1}{|V|} \sum_{u \in \mathcal{N}(v)} \mathfrak{g}(\varphi, G, \mathbf{f})_u^{(t)}$$

The node embeddings are then updated based on the message $\mathfrak{m}(\varphi, G, \mathbf{f})_v^{(k)}$ and the node feutures from the privious layer $\mathfrak{g}(\varphi, G, \mathbf{f})_v^{(k-1)}$:

$$\mathfrak{g}(\varphi, G, \mathbf{f})_v^{(k)} = \varphi^{(k)}\left(\mathfrak{g}(\varphi, G, \mathbf{f})_v^{(k-1)}, \mathfrak{m}(\varphi, G, \mathbf{f})_v^{(k)}\right), \tag{2}$$

where $\varphi^{(k)}$ is called the update function which is a learnable Lipschitz function like an MLP (Multi-Layer Perceptron). After $L$ layers, the last layer's hidden node representations $\mathfrak{g}(\varphi, G, \mathbf{f})_-^{(L)}$ are processed by an update layer. This layer is computed by applying a readout function $\psi : \mathbb{R}^{d_L} \mapsto \mathbb{R}^d$ and outputs a single graph-signal-level representation:

$$\mathfrak{G}(\varphi, \psi, G, \mathbf{f}) := \psi\left(\frac{1}{|V(G)|}\sum_{v \in V(G)} \mathfrak{g}(\varphi, G, \mathbf{f})_v^{(L)}\right) \tag{3}$$

$\psi$ is typically a permutation-invariant function, that ensures the graph features is independent of any node ordering.

### A.2.1 GRAPH ISOMORPHISM NETWORK (GIN)

Graph Isomorphism Network (GIN) Xu et al. (2019) is a popular MPNN model for graphs, which is as expressive as the Weisfeiler-Lehman graph isomorphism test. GIN is defined with sum aggregation but we use normalized sum aggregation. As an update function,it uses a multi layer preceptron (MLP) composed on addition of the massage $\mathfrak{m}(\varphi, G, \mathbf{f})_-^{(t)} := \mathfrak{m}_-^{(t)}$ and $\mathfrak{g}(\varphi, G, \mathbf{f})_-^{(t-1)} := \mathfrak{g}_-^{(t-1)}$ that depends on a contant $\epsilon$, that controls the weighting between the node's own embedding and its neighbors' embeddings, as follows:

$$\mathfrak{m}_v^{(t)} := \frac{1}{|V|}\sum_{u \in \mathcal{N}(v)} \mathfrak{g}_u^{(t-1)},$$

$$\mathfrak{g}_v^{(t)} := \mathrm{MLP}^{(t)}\left((1+\epsilon)\mathfrak{g}_v^{(t-1)} + \mathfrak{m}_v^{(t)}\right).$$

The readout after $L$ layers (which we also normalize in contrast to the original definition) is:

$$\mathfrak{G}(\varphi, \psi, G, \mathbf{f}) := \mathfrak{G} := \frac{1}{|V|}\sum_{v \in V} \mathfrak{g}_v^{(L)}.$$

GIN's ability to distinguish between non-isomorphic graphs makes it compatible for tasks, such as graph classification and node-level prediction.

### A.3 SUM AND MEAN AGGREGATION

We present here sum and mean aggregation schemes on graph-signals. This allows further discussion on the different aggregation schemes in Appendix D.

**Definition 20** (Sum Aggregation on Graph-Signals). *Let $\varphi$ be an L-layer MPNN model, $(G, \mathbf{f})$ be a graph-signal, and $t \in [L-1]$ then sum aggregation of $(G, \mathbf{f})$'s t-level features with respect to the node $v \in V(G)$ is defined as*

$$Agg(G, \mathfrak{g}(\varphi, G, \mathbf{f})_-^{(t)})(v) = \sum_{u \in \mathcal{N}(v)} \mathfrak{g}(\varphi, G, \mathbf{f})_u^{(t)}.$$

**Definition 21** (Mean Aggregation on Graph-Signals). *Let $\varphi$ be an L-layer MPNN model, $(G, \mathbf{f})$ be a graph-signal, and $t \in [L-1]$ then mean aggregation of $(G, \mathbf{f})$'s t-level features with respect to the node $v \in V(G)$ is defined as*

$$Agg(G, \mathfrak{g}(\varphi, G, \mathbf{f})_-^{(t)})(v) = \frac{1}{\mathcal{N}(v)}\sum_{u \in \mathcal{N}(v)} \mathfrak{g}(\varphi, G, \mathbf{f})_u^{(t)}.$$

## A.4 TOPOLOGY BASICS

Topology is a fundamental branch of mathematics, in which spatial properties that remain unchanged under continuous deformations are studied formally. Here, we introduce key concepts of topology.

**Definition 22** (Topological Space). *A topological space is a pair $(\mathcal{X}, \tau)$, where $\mathcal{X}$ is a set and $\tau$ is a collection of subsets of $\mathcal{X}$ satisfying:*

1. *Both $\emptyset$ and $\mathcal{X}$ are in $\tau$*

2. *$\tau$ is closed under finite intersections*

3. *$\tau$ is closed under arbitrary unions*

*The elements of $\tau$ are called open sets.*

### A.4.1 CONTINUITY

Continuity, in topology, generalizes the notion of continuity from calculus to any arbitrary topological space.

**Definition 23** (Continuous Function). *Given topological spaces $(\mathcal{X}, \tau_X)$ and $(\mathcal{Y}, \tau_Y)$, a function $f : \mathcal{X} \to \mathcal{Y}$ is continuous if the preimage of every open set in $\mathcal{Y}$ is open in $\mathcal{X}$. Formally, $f^{-1}(U) \in \tau_X$ for all $U \in \tau_Y$.*

### A.4.2 THE PRODUCT TOPOLOGY

The product topology is a topology which is naturally defined on a Cartesian product topological spaces. Although it can be defined for Cartesian products of any number of spaces, in our analysis, we are only interested in the finite case.

Let $\{(\mathcal{X}_i, \tau_i)\}_{i \in I}$ be an finite set of topological spaces, then the product space of the set $\{(\mathcal{X}_i, \tau_i)\}_{i \in I}$ is denoted by $\prod_{i \in I} \mathcal{X}_i$ and defined as the set of all vectors $(x_i)_{i \in I}$ where $x_i \in \mathcal{X}_i$ for each $i \in I$. The product topology is then generated by the basis $\{(U_i)_{i \in I} : U_i \in \tau_i\}$.

For each $i \in I$, the projection map $\pi_i : \prod_{j \in I} \mathcal{X}_j \to \mathcal{X}_i$ is defined by $\pi_i((x_j)_{j \in I}) = x_i$. In the product topology, all projection maps $\pi_i$ are continuous. Moreover, a sequence in the product space converges if and only if its projections onto each factor space converge.

### A.4.3 COMPACTNESS AND SEPARABILITY

Compactness and separability are both important properties in topology.

**Definition 24.** *[Compact Space] A topological space $(\mathcal{X}, \tau)$ is compact if every open cover of $\mathcal{X}$ has a finite subcover.*

**Definition 25** (Separable Space). *A topological space is separable if it contains a countable dense subset.*

**Theorem 26** (Tychonoff's Theorem). *Let $\{X_i\}_{i \in I}$ be a family of compact topological spaces. Then the product space $\prod_{i \in I} X_i$ is compact in the product topology.*

A metric space is a topological space, where the topology is induced by a distance function called a metric.

### A.4.4 NORM SPACES

A normed vector space is a vector space over the real or complex numbers on which a norm is defined.

**Definition 27.** *A* normed vector space *is a pair $(V, \|\cdot\|)$ where $V$ is a vector space over $\mathbb{R}$ or $\mathbb{C}$ and $\|\cdot\| : V \to \mathbb{R}$ is a function, called the* norm*, satisfying for all $x, y \in V$ and all scalars $\alpha$:*

1. *$\|x\| \geq 0$ and $\|x\| = 0$ if and only if $x = 0$.*

2. *$\|\alpha x\| = |\alpha| \|x\|$.*

3. *$\|x + y\| \leq \|x\| + \|y\|$.*

### A.4.5 METRIC AND PSEUDOMETRIC SPACES

A pseudometric space is a topological space, where the topology is induced by a distance function called a pseudometric.

**Definition 28** (Pseudometric Space). *A metric space is a pair $(\mathcal{X}, d)$ where $\mathcal{X}$ is a set and $d : \mathcal{X} \times \mathcal{X} \to \mathbb{R}$ is a function satisfying:*

1. *$d(x, y) \geq 0$, for all $x, y \in \mathcal{X}$.*

2. *$d(x, y) = d(y, x)$, for all $x, y \in \mathcal{X}$.*

3. *$d(x, z) \leq d(x, y) + d(y, z)$, for all $x, y, z \in \mathcal{X}$.*

Notice that it is possible that $d(x, y) = 0$ for some $x \neq y$, meaning distinct points can have zero distance. A metric is a pseudometric that satisfy the following additional requirement.

$$\forall x, y \in \mathcal{X} : d(x, y) = 0 \iff x = y.$$

The pair $(\mathcal{X}, d)$, where $\mathcal{X}$ is a set and $d : \mathcal{X} \times \mathcal{X} \to \mathbb{R}$ is a metric is called a metric space. The topology induced by a metric/pseudometric is generated by a base containing all open balls defined by $B(x, r) = \{y \in \mathcal{X} : d(x, y) < r\}$.

### A.4.6 COVERING NUMBER

**Definition 29.** *A metric space $\mathcal{X}$ is said to have a covering number $\kappa : (0, \infty) \to \mathbb{N}_0$ if, for any $\epsilon > 0$, $\mathcal{X}$ can be covered by $\kappa(\epsilon)$ balls of radius $\epsilon$ (see Appendix A.4.6).*

Note that if a metric space $\mathcal{X}$ is compact (see Definition 24), then it can be covered by a finite number of balls of radius $\epsilon$, for any $\epsilon > 0$.

### A.4.7 METRIC IDENTIFICATION

Let $(\mathcal{X}, d)$ be a pseudometric space. Consider the equivalence relation:

$$\forall x, y \in \mathcal{X} : x \sim_d y \quad \text{if} \quad d(x, y) = 0.$$

Any equivalence relation $\sim_d$ is part of a class of equivalence relations called *metric identifications*. A metric identification converts a pseudometric space into a metric space, while preserving the induced topologies. This identification is captured through the quotient map $\pi : \sim \mapsto \mathcal{X}/ \sim$ defined as the map $x \to [x]$. The open sets in the pseudometric space are exactly the sets of the form $\pi^{-1}(\mathcal{A})$, where $\mathcal{A} \in \mathcal{X}/ \sim$ is open. Namely, if an open set $\mathcal{A} \in \mathcal{X}$ contains $x$, it has to contain all of the other elements in $[x]$. The *pseudometric topology* is the topology generated by the open balls

$$B_r(p) = \{x \in \mathcal{X} : d(p, x) < r\},$$

which form a basis for the topology.

### A.4.8 COMPLETE METRIC SPACES

**Definition 30** (Complete Metric Space). *A metric space $(\mathcal{X}, d)$ is called* complete *if every Cauchy sequence in $\mathcal{X}$ converges to a point in $\mathcal{X}$. A sequence $\{x_n\} \subset \mathcal{X}$ is a* Cauchy sequence *if for every $\epsilon > 0$, there exists an integer $N \in \mathbb{N}_0$ such that for all $m, n \geq N$, the following holds:*

$$d(x_n, x_m) < \epsilon. \tag{4}$$

In other words, if the elements of a sequence get arbitrarily close to each other as the sequence progresses, then that sequence also converges to a point within the space. If a topological space is metrizable to a complete metric space, we say that it is *completely metrizable*.

### A.4.9 METRIZABLE SPACES

**Definition 31** (Metrizable Space). *A topological space $(\mathcal{X}, \tau)$ is called* metrizable *if there exists a metric $d : \mathcal{X} \times \mathcal{X} \to \mathbb{R}$ such that the topology induced by the metric $d$ coincides with the topology $\tau$ on $\mathcal{X}$.*

In other words, the open sets in the topology $\tau$ are exactly the open sets with respect to the metric $d$.

### A.4.10 MEASURE SPACES

Measure theory generalizes and formalizes geometrical measures such as length, area, and volume as well as other notions, such as magnitude, mass, and probability of events.

**Definition 32** ($\sigma$-algebra). *Let $\mathcal{X}$ be a set. A collection $\Sigma$ of subsets of $\mathcal{X}$ is called a $\sigma$-algebra if the following properties are satisfied:*

1. $\mathcal{X} \in \Sigma$.

2. $\Sigma$ *is closed under complement, i.e., if $\mathcal{A} \in \Sigma$, then $\mathcal{X} \setminus \mathcal{A} \in \Sigma$,*

3. $\Sigma$ *is closed under countable unions, i.e., if $\{\mathcal{A}_n\}_{n=1}^{\infty}$ is a countable collection of sets in $\Sigma$, then*

$$\bigcup_{n=1}^{\infty} \mathcal{A}_n \in \Sigma. \tag{5}$$

The pair $(\mathcal{X}, \Sigma)$ is called a *measurable space*. A $\sigma$-algebra provides the foundation for the construction of measures,

**Definition 33** (Measure). *Let $(\mathcal{X}, \Sigma)$ be a measurable space, where $\mathcal{X}$ is a set and $\Sigma$ is a $\sigma$-algebra on $\mathcal{X}$. A function $\mu : \Sigma \to [0, \infty]$ is called a* measure *if it satisfies the following properties:*

1. Non-negativity*: For all $\mathcal{A} \in \Sigma$, $\mu(\mathcal{A}) \geq 0$.*

2. Null empty set*: $\mu(\emptyset) = 0$.*

3. Countable additivity (or $\sigma$-additivity)*: For any countable collection of pairwise disjoint sets $\{\mathcal{A}_n\}_{n=1}^{\infty} \subset \Sigma$, the measure of their union is the sum of their measures:*

$$\mu\left(\bigcup_{n=1}^{\infty} \mathcal{A}_n\right) = \sum_{n=1}^{\infty} \mu(\mathcal{A}_n). \tag{6}$$

A measure $\mu$ assigns a non-negative extended real number to each set in the $\sigma$-algebra $\Sigma$. The triple $(\mathcal{X}, \Sigma, \mu)$ is called a *measure space*.

### A.4.11 POLISH SPACES

Polish spaces are topological spaces, that have a "nice" topology in the following way.

**Definition 34** (Polish Space). *A topological space is Polish if it is separable and completely metrizable, i.e., there exists a complete metric that generates its topology.*

### A.5 STONE–WEIERSTRASS THEOREM

Given a set $\mathcal{A}$, we denote by $1_{\mathcal{A}} : \mathcal{A} \to \mathbb{R}$ a non-zero constant function. We present a variation of the Stone-Weierstrass Theorem (see Rudin, 1976, Theorem 7.32) we use in Appendix I to prove a universal approximation theorem.

**Theorem 35** (Real Stone–Weierstrass). *Let $\mathcal{K}$ be a compact metric space and $A \subseteq C(\mathcal{K}, \mathbb{R})$ be a sub-algebra that contains $1_{\mathcal{K}}$ and separates points, i.e., for every $k \neq l \in \mathcal{K}$ there is $f \in A$ such that $f(k) \neq f(l)$. Then $A$ is uniformly dense in $C(\mathcal{K}, \mathbb{R})$.*

The following corollary (taken from Grebík & Rocha (2021)) is another useful description of the Stone-Weierstrass theorem, and is used as well in the universal approximation proof in Appendix I.

**Corollary 36** (Separating Measures). *Let $\mathcal{K}$ be a compact metric space and $\mathcal{E} \subseteq C(\mathcal{K}, \mathbb{R})$ be closed under multiplication, contain $1_{\mathcal{K}}$, and separate points. Then for every $\mu \neq \nu \in \mathscr{M}_{\leq 1}(\mathcal{K})$ there is $f \in \mathcal{E}$ such that*

$$\int_{\mathcal{K}} f d\mu \neq \int_{\mathcal{K}} f d\nu$$

*i.e., the linear functionals that correspond to elements of $\mathcal{E}$ separate points in $\mathcal{M}_{\leq 1}(\mathcal{K})$.*

## A.6 THE CUT NORM AND THE CUT METRIC

The *cut norm* was introduced by Frieze & Kannan (1999) and provides the central notion of convergence in the theory of dense graph limits. Levie (2023); Rauchwerger & Levie (2025) extended the cut norm to graphons with d-channel signals, i.e., graphon-signals in $\mathcal{WL}_r^d$.

**Definition 37** (Cut Norm and Cut Metric). *The* cut norm *of a measurable $W : [0,1]^2 \mapsto \mathbb{R}$, and the cut norm of a measurable function $f : [0,1] \mapsto \mathbb{R}^d$ is defined as*

$$\|W\|_\square := \sup_{A,B \in [0,1]} \left| \int_{A \times B} W(x,y)\, dxdy \right| \quad \text{and} \quad \|f\|_\square := \frac{1}{d} \sup_{S \subset [0,1]} \left\| \int_S f(x)\, d\lambda(x) \right\|_1,$$

*where the supremum is taken over the measurable subsets $A, B \subseteq [0,1]$ and $S \subseteq [0,1]$, respectively. We define the* graphon-signal cut norm, *for a graphon signal $(W,f) \in \mathcal{WL}_r^d$, by*

$$\|(W,f)\|_\square := \|W\|_\square + \|f\|_\square.$$

*We name the metric induced by the cut norm the* cut metric, *which is defined to be*

$$d_\square((W,f),(V,g)) := \|(W,f) - (V,g)\|_\square$$

*for any two graphon-signals $(W,f),(V,g) \in \mathcal{WL}_r^d$.*

## A.7 THE CUT DISTANCE

Lovász (2012) first defined the *cut distance* on graphons through the graphon cut norm. The cut norm was later extended to graphon-signals by Levie (2023); Rauchwerger & Levie (2025) using the graphon-signal cut norm.

Let $S'_{[0,1]}$ be the space of measurable bijections between co-null sets of $[0,1]$. Namely,

$$S'_{[0,1]} := \{\phi : A \to B \mid A, B \text{ co-null in } [0,1], \text{ and } \forall S \in A, \ \mu(S) = \mu(\phi(S))\},$$

where $\phi$ is a measurable bijection and $A, B, S$ are measurable. Let $\phi$ be a measurable bijection in $S'_{[0,1]}$, for any graphon, $W$, and a signal $f$, we define $W^\phi(x,y) := W(\phi(x),\phi(y))$ and $f^\phi(z) := f(\phi(z))$, respectively. We then define $(W,f)^\phi := (V^\phi, g^\phi)$. Note that $W^\phi$ and $f^\phi$ are only define up to a null-set, therefore, we (arbitrarily) set $W, W^\phi, f, f^\phi$ to 0 in this null-set. This does not affect our analysis, as the cut norm is not affected by changes to the values of functions on a null sets.

**Definition 38** (Cut Distance). *The* cut distance *is defined to be*

$$\delta_\square((W,f),(V,g)) := \inf_{\phi \in S'_{[0,1]}} d_\square((W,f),(V,g)^\phi).$$

## A.8 THE QUOTIENT METRIC SPACES OF GRAPHON-SIGNALS

The graphon-signal cut distance $\delta_\square$ is a pseudometric. Consider the equivalence relations: $(W,f) \sim (V,g)$ if $\delta_\square((W,f),(V,g)) = 0$. For any graphon-signals $(W,f) \in \mathcal{WL}^d$, denote the equivalent class of $(W,f)$, with respect to $\sim$, as $[(W,f)]$. Then, the quotient space

$$\widetilde{\mathcal{WL}_r^d} := \mathcal{WL}/\sim$$

of the equivalence classes $[(W,f)]$ is a metric space, with the metric $\delta_\square([(W,f)],[(V,g)]) := \delta_\square((W,f),(V,g))$. Notice that the equivalence relation $\sim$ is a metric identification (Appendix A.4.7).

## A.9 THE COMPACTNESS OF THE CUT DISTANCE

The space of graphon-signals is compact w.r.t. the cut distance $\delta_\square$ (Definition 38), as proven in Levie (2023); Rauchwerger & Levie (2025).

**Theorem 39** (Levie (2023), Theorem 3.6.; Rauchwerger & Levie (2025), Theorem 4.4.). *The Pseudometric space $(\widetilde{\mathcal{WL}_r^d}, \delta_\square)$ and the metric space $(\mathcal{WL}_r^d, \delta_\square)$ are compact. Moreover, given $r > 0$ and $c > 1$, for every sufficiently small $\epsilon > 0$, they can be covered by*

$$\kappa(\epsilon) = 2^{k^2} \tag{7}$$

*balls of radius $\epsilon$, where $k = \lceil 2^{\frac{9c}{4\epsilon^2}} \rceil$.*

## A.10 THE CUT METRIC REGULARITY LEMMA FOR GRAPHON-SIGNALS

To formulate our regularity lemma, we first define spaces of step functions, based on Levie (2023); Rauchwerger & Levie (2025).

**Definition 40** (Levie (2023), Definition 3.3.; Rauchwerger & Levie (2025), Definition 4.1.). *Given a partition $\mathcal{P}_k$, and $d \in \mathbb{N}_0$, we define the space $\mathcal{S}_{\mathcal{P}_k}^{p \to d}$ of* step functions $\mathbb{R}^p \mapsto \mathbb{R}^d$ *over the partition $\mathcal{P}_k$ to be the space of functions $F : [0,1]^p \to \mathbb{R}^d$ of the form*

$$F(x_1, \ldots, x_p) = \sum_{j=(j_1, \ldots, j_p) \in [k]^p} \left( \prod_{l=1}^{p} \mathbb{1}_{P_{j_l}}(x_l) \right) c_j, \tag{8}$$

*for any choice of $\{c_j \in \mathbb{R}^d\}_{j \in [k]^p}$.*

We call any graphon, that also belongs to the set $\mathcal{S}_{\mathcal{P}_k}^{2 \to 1}$ a *step graphon* (also called a *stochastic block model (SBM)*) with respect to $\mathcal{P}_k$. We call any signal, that also belongs to the set $\mathcal{S}_{\mathcal{P}_k}^{1 \to d}$ a *step signal*. We define the space of (graphon-signal) SBMs with respect to $\mathcal{P}_k$ as $\mathcal{WL}_r^d \cap (\mathcal{S}_{\mathcal{P}_k}^{2 \to 1} \times \mathcal{S}_{\mathcal{P}_k}^{1 \to d})$. Next, we define the projection of a graphon-signal upon a partition.

**Definition 41** (Levie (2023) Definition B.10; Rauchwerger & Levie (2025), Definition 4.2.). *Let $\mathcal{P}_n = \{P_1, \ldots, P_n\}$ be a partition of $[0,1]$, and $(W, f) \in \mathcal{WL}_r^d$. We define the projection of $(W, f)$ upon $\mathcal{WL}_r^d \cap (\mathcal{S}_{\mathcal{P}_k}^{2 \to 1} \times \mathcal{S}_{\mathcal{P}_k}^{1 \to d})$ to be the step graphon-signal $(W, f)_{\mathcal{P}_n} = (W_{\mathcal{P}_n}, f_{\mathcal{P}_n})$ that attains the value*

$$W_{\mathcal{P}_n}(x,y) = \int_{[0,1]^2} W(x,y) \mathbb{1}_{P_i \times P_j}(x,y) dx dy \,, \quad f_{\mathcal{P}_n}(x) = \int_{[0,1]} f(x) \mathbb{1}_{P_i}(x) dx$$

*for every $(x,y) \in P_i \times P_j$ and $1 \le i, j \le n$.*

Note that we use the notation $(W_{\mathcal{P}_n}, f_{\mathcal{P}_n})$ and the notation $([W]_{\mathcal{P}_n}, [f]_{\mathcal{P}_n})$ interchangeably. The following version of the regularity lemma states that any graphon-signal can be approximated using the average values of both the graphon and the signal within some partition.

**Theorem 42** (Levie (2023), Theorem 3.4 and Corollary B.11; Rauchwerger & Levie (2025), Theorem 4.3, Regularity Lemma for Graphon-Signals). *For any $c > 1$, $r > 0$, and any sufficiently small $\epsilon > 0$, for every $n \ge 2^{\lceil \frac{8c}{\epsilon^2} \rceil}$ and every $(W, f) \in \mathcal{WL}_r^d$, we have*

$$\delta_\square \Big( (W, f), (W, f)_{\mathcal{I}_n} \Big) \le \epsilon,$$

*where $\mathcal{I}_n$ is the equipartition of $[0,1]$ into $n$ intervals.*

## A.11 WEIGHED PRODUCT METRIC

We say that a product metric is weighed if there is a vector of weights $\vec{w} = (w_i)_{i=1}^n$ with positive entries $(w_i > 0)$ such that

$$d_1((x_1, \ldots, x_n), (y_1, \ldots, y_n)) := \|(w_1 \cdot d_{X_1}(x_1, y_1), \ldots, w_n \cdot d_{X_n}(x_n, y_n))\|_1.$$

## A.12 THE BOUNDED-LIPSCHITZ DISTANCE

**Definition 43.** *Let $\mathcal{X}$, $\mathcal{Y}$ be two metric spaces. Let $f \in \mathrm{Lip}(\mathcal{X}, \mathcal{Y})$, where $\mathrm{Lip}(\mathcal{X}, \mathcal{Y})$ is the space of Lipschitz continuous mappings $\mathcal{X} \mapsto \mathcal{Y}$. We define the function $\|\cdot\|_{\mathrm{L}} : \mathrm{Lip}(\mathcal{X}, \mathcal{Y}) \to \mathbb{R}_+$ by*

$$\|f\|_{\mathrm{L}} = \inf_{\mathrm{L}} \{L : L \text{ is a Lipschitz constant of } f\}.$$

Note that $\|\cdot\|_{\mathrm{L}}$ is a seminorm over $\mathrm{Lip}(\mathcal{X}, \mathcal{Y})$. Moreover, notice that $\|f\|_{\mathrm{L}}$ can also be expressed as

$$\|f\|_{\mathrm{L}} = \sup_{x \ne y} \frac{|f(x) - f(y)|}{d(x,y)}.$$

Since the set of all valid Lipschitz constants is closed from below in $\mathbb{R}$ and non-empty (for a Lipschitz function), the infimum is attainable in the sense that it still satisfies the Lipschitz condition:

$$|f(x) - f(y)| \leq \|f\|_{\mathrm{L}}\, d(x, y).$$

Thus, $\|f\|_{\mathrm{L}}$ is indeed a Lipschitz constant - it is the smallest one, also known as the *optimal Lipschitz constant* or the *best Lipschitz constant* of $f$.

**Definition 44.** *Let $(\mathcal{X}, \Sigma, \mu)$ be a measure space and $(\mathcal{Y}, \|\cdot\|_{\mathcal{Y}})$ be a complete normed vector space and let $f : \mathcal{X} \mapsto \mathcal{Y}$ be a measurable function. The $\ell_\infty$-norm of $f$ is defined as*

$$\|f\|_\infty = \operatorname*{ess\,sup}_{x \in \mathcal{X}} \|f(x)\|_{\mathcal{Y}}.$$

**Definition 45** (The Bounded-Lipschitz Function). *Let $(\mathcal{X}, \Sigma, \mu, d)$ be a measure space, with a metric $d$ and $(\mathcal{Y}, \|\cdot\|_{\mathcal{Y}})$ be complete normed space. Let $\mathrm{Lip}(\mathcal{X}, \mathcal{Y})$ be the space of Lipschitz continuous mappings $\mathcal{X} \mapsto \mathcal{Y}$. We define the Bounded-Lipschitz function over $\mathrm{Lip}(\mathcal{X}, \mathcal{Y})$ as*

$$\|\cdot\|_{\mathbf{BL}} := \|\cdot\|_\infty + \|\cdot\|_{\mathrm{L}},$$

*when $\|\cdot\|_{\mathrm{L}}$ and $\|\cdot\|_\infty$ are defined in Definition 43 and Definition 44, respectively.*

Note that as $\|\cdot\|_\infty$ is a norm and $\|\cdot\|_{\mathrm{L}}$ is a semi norm then $\|\cdot\|_{\mathbf{BL}}$ is a seminorm as well.

**Definition 46** (The Bounded-Lipschitz Distance). *Let $\mathcal{X}$ be a measurable metric space, $\mathcal{Y}$ be a metric space, $\mu, \nu \in \mathcal{M}_1(\mathcal{X})$. and $f : S \to \mathbb{R}$ be a $\mu$-measurable and $\nu$-mesurable function. Let*

$$\mathbf{BL}(\mu, \nu) := \sup_{\|f\|_{\mathbf{BL}} \leq 1} \left| \int f d\mu - \int f d\nu \right|.$$

*be the Bounded-Lipschitz distance of $\mu$ and $\nu$, respectively, where $\|f\|_{\mathbf{BL}}$ is the Bounded-Lipschitz function.*

### A.13 THE KANTOROVICH-RUBINSHTEIN DISTANCE

**Definition 47** (The Kantorovich-Rubinshtein Distance). *Let $\mathcal{X}$ be a measurable metric space, $\mathcal{Y}$ be a metric space, $\mu, \nu \in \mathcal{M}_1(\mathcal{X})$. and $f : S \to \mathbb{R}$ be a $\mu$-measurable and $\nu$-mesurable function. Let*

$$\mathbf{K}(\mu, \nu) := \sup_{\|f\|_{\mathrm{L}} \leq 1} \left| \int f d\mu - \int f d\nu \right|.$$

*be the Kantorovich-Rubinshtein distance of $\mu$ and $\nu$, respectively.*

### A.14 THE PROKHOROV METRIC

**Definition 48** (The Prokhorov Metric). *Let $(\mathcal{X}, \mathcal{B}(\mathcal{X}))$ be a measurable metric space with Borel $\sigma$-algebra. We define $A^\epsilon := \{y \in S \mid d(x, y) \leq \epsilon \text{ for some } x \in A\}$ for a subset $A \subseteq \mathcal{X}$ and $\epsilon \geq 0$. Then, the Prokhorov metric $\mathbf{P}$ on $\mathscr{M}_{\leq 1}(\mathcal{X}, \mathcal{B})$ is given by*

$$\mathbf{P}(\mu, \nu) := \inf\{\epsilon > 0 \mid \mu(A) \leq \nu(A^\epsilon) + \epsilon \text{ and } \nu(A) \leq \mu(A^\epsilon) + \epsilon \text{ for every } A \in \mathcal{B}\}.$$

## B ADDITIONAL RELATED WORK

We provide here an extended discussion on related work. Several recent works consider pseudo metrics on graphs, aiming to capture structural properties of the graph and the computational procedure of message passing. The latter is often described by *computation trees*, hierarchical structures resulting from unrolling message passing (Morris et al., 2019; Arvind et al., 2020; Garg et al., 2020; Xu et al., 2020; Chuang & Jegelka, 2022; Jegelka, 2022). Similarly, hierarchical structures of measures have been used for the analysis of MPNNs (Chen et al., 2022; 2023; Böker et al., 2023; Maskey et al., 2022). Although these structures may look different at first glance, they can describe the same iterative message passing mechanism. Other approaches include Titouan et al. (2019) graph metric defined a using both Wasserstein distance and Gromov-Wasserstein distance Mémoli (2011). This

approach, just like classic graph metrics Bunke & Shearer (1998); Sanfeliu & Fu (1983) requires using approximation. In addition, several graph kernels have been proposed Vishwanathan et al. (2010); Borgwardt et al. (2020). Here, we focus on the viewpoint of computation trees, as they closely align with MPNNs. A number of existing works study generalization for GNNs, e.g., via VC dimension, Rademacher complexity or PAC-Bayesian analysis (Oono & Suzuki, 2020; Tang & Liu, 2023; Li et al., 2022; Garg et al., 2020; Maskey et al., 2022; Morris et al., 2023; Maskey et al., 2022; Liao et al., 2021b). Most need assumptions on the data distribution, and often on the MPNN model too. Levie (2023); Rauchwerger & Levie (2025) use covering number (Definition 29), for a wide range of data distributions. We expand this result to a more general setting.

## C ADDITIONAL DISCUSSION

Here, we provide additional high level discussion on different aspects of our construction.

### C.1 THE LIMITATION OF OUR CONSTRUCTION TO DENSE GRAPHS

For sparse graphs, the number $E$ of edges is much smaller than the number $N^2$ of vertices squared. As a result, the induced graphon is supported on a set of small measure. Since graphon are bounded by 1, this means that induced graphons from sparse graphs are close in $L_1([0,1]^2)$ to the 0 graphon. DIDM mover's distance gives a courser topology than cut distance, which is courser than $L_1([0,1]^2)$. Hence, since all sparse graph sequences converge to 0 in $L_1([0,1]^2)$, they also converge to 0 in DIDM distance.

Therefore, we can only use our theory for datasets of graphs that roughly have the same sparsity level $S \in \mathbb{N}_0$, i.e., $N^2/E$ is on the order of some constant $S$ for most graphs in the dataset. For this, one can scale our distance by $S$, making it appropriate to graphs with $E \ll N^2$ edges, in the sense that the graphs will not all be trivially close to 0. Our theory does not solve the problem of sequences of graph asymptotically converging to 0.

In future work, one may develop a fine-grained expressivity theory based sparse graph limit theories. There are several graph limit theories that apply to sparse graphs, including Graphing theory Lovász (2020), Benjamini–Schramm limits Abért et al. (2014); Béla & Riordan (2011); Hatami et al. (2014), stretched graphons Jian et al. (2023); Ji et al. (2024), $L^p$ graphons Borgs et al. (2014a;b), and graphop theory Ágnes Backhausz & Szegedy (2020), which extends all of the aforementioned approaches . Future work may extend our theory to sparse graph limits.

### C.2 COMPARISON OF SUM MEAN, AND NORMALIZED SUM AGGREGATIONS

See Appendix A.3 for the definition of sum and mean aggregation on attributed graphs. First, (un-normalized) sum aggregation (Definition 20) does not work in the context of our analysis. Indeed, given an MPNN that simply sums the features of the neighbors and given the sequence of complete graphs of size $N \in \mathbb{N}_0$, then, the output of the MPNN on these graphs diverges to infinity as $n \to \infty$. As a result, equivalency of the metric at the output space of MPNNs with a compact metric on the space of graphs is not possible.

Another popular aggregation scheme is mean aggregation, defined canonically on graphon-signal as follows (Maskey et al., 2022; 2024).

**Definition 49** (Mean Aggregation on Graphon-Signals). *Let $\varphi$ be an L-layer MPNN model, $(W, f)$ be a graph-signal, and $t \in [L-1]$. Then mean aggregation of $(W, f)$'s t-level features with respect to the node $x \in V(W)$ is defined as*

$$Agg(W, \mathfrak{f}(\varphi, W, f)^{(t)}_-)(x) = \frac{1}{\int_{[0,1]} W(x,y)d\lambda(y)} \int_{[0,1]} W(x,y)\mathfrak{f}(\varphi, W, f)^{(t)}_y d\lambda(y).$$

The theory could potentially extend to mean aggregation using two avenues. One approach is to do this under a limiting assumption: restricting the space to graphs/graphons with minimum node degree bounded from below by a constant. This is like an idea outlined in Maskey et al. (2024).

A second option is to redefine $\delta_{\text{DIDM}}$ using balanced OT. In this paper, $\delta_{\text{DIDM}}$ highly relates to the type of computation MPNNs with normalized sum aggregation perform. We used unbalanced OT

(Definition 1) as the basis of $\delta_{\mathrm{DIDM}}$ due to the fact that MPNNs with normalized sum aggregation do not average incoming messages, which means they can separate nodes of different degrees within a graph. MPNNs with mean aggregation, in contrast, do average incoming messages. Hence, an appropriate version of optimal transport, in this case, could be based on averaging. I.e., using balanced OT on normalized measures could serve as a base for defining metrics in the analysis of MPNNs with mean aggregation.

### C.3    Conparison of Our Generalization Bound to Related Works

Both our work and Levie (2023); Rauchwerger & Levie (2025) do not make any assumptions on the graphs and allow a general MPL scheme. Our classification learning setting generalizes that of Levie (2023), which assumes a ground truth deterministic class per input, while we consider a joint distribution over the input and label as Rauchwerger & Levie (2025).

In comparison to other recent works on generalization, Garg et al. (2020); Liao et al. (2021a) assume bounded degree graphs, Morris et al. (2023) assumes graphs with bounded color complexity, and Maskey et al. (2022; 2024) assume the graphs are sampled from a small set of graphons. Moreover, Garg et al. (2020); Liao et al. (2021a); Morris et al. (2023) do not allow a general MPL s presented in this paper, so their dependence on $N$ is $N^{-1/2}$. This means their generalization bound decays faster as a function of the training set size $N$. Li et al. (2022) analysis is restricted to graph convolution networks which are a special case of MPNNs. Tang & Liu (2023) does not focus on graph classification tasks but on node classification. Oono & Suzuki (2020) focus on transductive learning in contrast to our inductive learning analysis.

## D    MPNN Architectures: Standard and Alternative Formulations

Levie (2023) suggest a different MPNN definition for the analysis of MPNNs on graphon-signals. This definition includes, in addition to update and readout functions, functions called message functions. In this section we show that, although our MPNNs definition is, in it essence, a simplified version of the MPNNs in Levie (2023), the two definition are equivalent in terms of expressivity. We start with a recap on our standard MPNNs definition.

### D.1    Standard MPNNs

An MPNN as defined in Definition 7 consists of an initial layer which updates the features, via a learnable Lipschitz continuous mapping $\varphi^{(0)} : \mathbb{R}^p \mapsto \mathbb{R}^{d_0}$, followed by $L$ layers, each of which consists of two steps: a *message passing layer* (MPL) that aggregates neighborhood information, followed by a node-wise *update layer*. Here, when we apply the MPNN on graph-signals or graphon-signals, we assume the MPL to be normalized sum pooling or integration pooling, respectively. The normalized sum we use can be defined for graphon-signals as follows.

**Definition 50** (Normalized Sum Aggregation on Graphon-Signals). *Let $\varphi$ be an $L$-layer MPNN model, $(W, f)$ be a graph-signal, and $t \in [L - 1]$ then normalized sum aggregation of $(W, f)$'s $t$-level features with respect to the node $x \in V(W)$ is defined as*

$$Agg(W, \mathfrak{f}(\varphi, W, f)_{-}^{(t)})(x) = \int_{[0,1]} W(x, y) \mathfrak{f}(\varphi, W, f)_y^{(t)} d\lambda(y).$$

The update layer consists of a learnable Lipschitz continuous mapping $\varphi^{(t)} : \mathbb{R}^{2d_{t-1}} \mapsto \mathbb{R}^{d_t}$ where $1 \leq t \leq L$ is the layer's index. Each layer computes a representation for each node. For predictions on the full graph, a *readout layer* aggregates the node representations into a single graph feature and transforms it by a learnable Lipschitz function $\psi : \mathbb{R}^{d_L} \mapsto \mathbb{R}^d$ for some $d \in \mathbb{N}_0$. For the readout on graph-signals and graphon-signals, we use average pooling. The readout layer aggregates representations across all nodes when a single graph representation is required (e.g. for graph classification).

## D.2 ALTERNATIVE MPNNS

We next show how to extend our normalized sum aggregation, used in this paper, to an aggregation scheme with a function called message function. This aggregation scheme is used in Levie (2023), for the analysis of MPNNs on graphon-signals. The idea is that general message functions $\phi$ depend both and the feature $b$ at the transmitting node and the feature $a$ at the receiving node of the message, i.e., $\phi(a, b)$. In Levie (2023), such a general $\phi(a, b)$ was approximated by a linear combination of simple tensors of the form $\xi_{\mathrm{rec}}(a)\xi_{\mathrm{trans}}(a)$ to accommodate the analysis.

**Definition 51** (Message Function). *Let $K \in N$. For every $1 \leq k \leq K$, let $\xi_{k,\mathrm{rec}}, \xi_{k,\mathrm{trans}} : \mathbb{R}^d \mapsto \mathbb{R}^p$ be Lipschitz continuous functions that we call the* receiver *and* transmitter message functions, *respectively. The corresponding* message function *$\phi : \mathbb{R}^{2d} \mapsto \mathbb{R}^p$ is the function*

$$\phi(a, b) = \sum_{k=1}^{K} \xi_{k,\mathrm{rec}}(a)\xi_{k,\mathrm{trans}}(b),$$

*where the multiplication is element-wise along the feature dimension.*

Given some signal $f$ over the domain $\mathcal{X}$, we see the point $x \in \mathcal{X}$ as the receiver of the message $\phi(f(x), f(y))$, and $y$ as the transmitter, and call $\phi(f(-), f(-)) : \mathcal{X}^2 \mapsto \mathbb{R}^p$ the message kernel.

Just like with MPNN models, for predictions on the full graph, a *readout layer* aggregates the node representations into a single graph feature and transforms it by a learnable Lipschitz function $\psi : \mathbb{R}^{d_L} \mapsto \mathbb{R}^d$ for some $d \in \mathbb{N}_0$. For the readout, we use average pooling. We now define the alternative MPNN model.

**Definition 52** (Alternative MPNN Model). *Let $L \in \mathbb{N}_0$ and $p, d_0, \ldots, d_L, p_0, \ldots, p_{L-1}, d \in \mathbb{N}_0$. We call the tuple $(\varphi, \phi)$ such that $\varphi$ is any collection $\varphi = (\varphi^{(t)})_{t=0}^{L}$ of Lipschitz continuous functions $\varphi^{(0)} : \mathbb{R}^p \mapsto \mathbb{R}^{d_0}$ and $\varphi^{(t)} : \mathbb{R}^{d_{t-1} \times p_{t-1}} \mapsto \mathbb{R}^{d_t}$, for $1 \leq t \leq L$, and $\phi$ is any collection $\phi = (\phi^{(t)})_{t=1}^{L}$ of (Lipschitz continuous) message functions $\phi^{(t)} : \mathbb{R}^{2d_{t-1}} \mapsto \mathbb{R}^{p_{t-1}}$, for $1 \leq t \leq L$, an $L$-*layer alternative MPNN model, and call $\varphi^{(t)}$ update functions. For Lipschitz continuous $\psi : \mathbb{R}^{d_L} \mapsto \mathbb{R}^d$, we call the tuple $(\varphi, \phi, \psi)$ an* alternative MPNN model with readout, *where $\psi$ is called a* readout function. *We call $L$ the* depth *of the MPNN, $p$ the* input feature dimension, *$d_0, \ldots, d_L$ the* hidden node feature dimensions, *$p_0, \ldots, p_{L-1}$ the* hidden edge feature dimensions, *and $d$ the* output feature dimension.

It is possible to define the application of alternative MPNN models not only on graphon-signals, but on graph-signals, IDMs and DIDMs as well. In this discussion our purpose is to show that our aggregation schemes are equivalent on graph-signals and graphon-signals.

For our purpose, the application of the alternative MPNN model on graph-signals and graphon-signals is enough.

**Definition 53** (Alternative MPNNs on Graph-Signals). *Let $(\varphi, \phi, \psi)$ be an $L$-layer alternative MPNN model with readout, and $(G, \mathbf{f})$ be a graphon-signal where $\mathbf{f} : V(G) \mapsto \mathbb{R}^p$. The application of the MPNN on $(G, \mathbf{f})$ is defined as follows: initialize $\hat{\mathfrak{g}}_{-}^{(0)} := \varphi^{(0)}(\mathbf{f}(-))$, and compute the hidden node representations $\hat{\mathfrak{g}}_{-}^{(t)} : V(G) \to \mathbb{R}^{d_t}$ at layer $t$, with $1 \leq t \leq L$ and the graphon-level output $\mathfrak{F} \in \mathbb{R}^d$ by*

$$\hat{\mathfrak{g}}_v^{(t)} := \varphi^{(t)}\Big(\hat{\mathfrak{g}}_v^{(t-1)}, \frac{1}{|V(G)|} \sum_{u \in \mathcal{N}(v)} \phi^{(t)}(\hat{\mathfrak{g}}_v^{(t-1)}, \hat{\mathfrak{g}}_u^{(t-1)})\Big) \quad and \quad \hat{\mathfrak{G}} := \psi\Big(\frac{1}{|V(G)|} \sum_{v \in V(G)} \hat{\mathfrak{g}}_x^{(L)}\Big).$$

**Definition 54** (Alternative MPNNs on Graphon-Signals). *Let $(\varphi, \phi, \psi)$ be an $L$-layer alternative MPNN model with readout, and $(W, f)$ be a graphon-signal where $f : V(W) \mapsto \mathbb{R}^p$. The application of the MPNN on $(W, f)$ is defined as follows: initialize $\hat{\mathfrak{f}}_{-}^{(0)} := \varphi^{(0)}(f(-))$, and compute the hidden node representations $\hat{\mathfrak{f}}_{-}^{(t)} : V(W) \to \mathbb{R}^{d_t}$ at layer $t$, with $1 \leq t \leq L$ and the graphon-level output $\mathfrak{F} \in \mathbb{R}^d$ by*

$$\hat{\mathfrak{f}}_x^{(t)} := \varphi^{(t)}\Big(\hat{\mathfrak{f}}_x^{(t-1)}, \int_{[0,1]} W(x, y)\phi^{(t)}(\hat{\mathfrak{f}}_x^{(t-1)}, \hat{\mathfrak{f}}_y^{(t-1)})d\lambda(y)\Big) \quad and \quad \hat{\mathfrak{F}} := \psi\Big(\int_{[0,1]} \hat{\mathfrak{f}}_x^{(L)}d\lambda(x)\Big).$$

As with the standard MPNN features, to clarify the dependence of $\hat{\mathfrak{f}}$ and $\hat{\mathfrak{F}}$ on $(\varphi, \phi)$ and $(w, f)$, we often denote $\hat{\mathfrak{f}}(\varphi, \phi)_v^{(t)}$ or $\hat{\mathfrak{f}}(\varphi, \phi, W, f)_v^{(t)}$, and $\hat{\mathfrak{F}}(\varphi, \phi, \psi)$ or $\hat{\mathfrak{F}}(\varphi, \phi, \psi, W, f)$.

**Definition 55** (Alternative Normalized Sum Aggregation on Graphon-Signals). *Let $\varphi$ be an $L$-layer MPNN model, $(W, f)$ be a graph-signal, and $t \in [L-1]$ then normalized sum aggregation of $(W, f)$'s $t$-level features with respect to the node $x \in V(W)$ is defined as*

$$Agg(W, \mathfrak{f}(\varphi, W, f)_{-}^{(t)})(x) = \int_{[0,1]} W(x, y) \phi^{(t)}(\hat{\mathfrak{f}}_x^{(t-1)}, \hat{\mathfrak{f}}_y^{(t-1)}) d\lambda(y).$$

Many well-known MPNNs architectures can be easily expressed as alternative MPNNs. We now present an examples taken from Levie (2023) of a spectral convolutional network.

**Definition 56** (Vector Concatenation). *Let $\mathbf{a} \in \mathbb{R}^m$ and $\mathbf{b} \in \mathbb{R}^n$ be two vectors. The* concatenation *of $\mathbf{a}$ and $\mathbf{b}$, denoted as $[\mathbf{a}; \mathbf{b}]$, is a vector in $\mathbb{R}^{m+n}$ defined as:*

$$[\mathbf{a}; \mathbf{b}] = \begin{bmatrix} a_1 \\ a_2 \\ \vdots \\ a_m \\ b_1 \\ b_2 \\ \vdots \\ b_n \end{bmatrix}.$$

Given a graph-signal $(G, \mathbf{f})$, with $\mathbf{f} \in \mathbb{R}^{n \times d}$ with adjacency matrix $A \in \mathbb{R}^{n \times n}$, a spectral convolutional layer based on a polynomial filter $\text{filter}(\lambda) = \sum_{j=0}^{J} \lambda^j C_j$, where $C_j \in \mathbb{R}^{d \times p}$, is defined to be

$$\text{filter}(A)\mathbf{f} = \sum_{j=0}^{J} \frac{1}{n^j} A^j \mathbf{f} C_j,$$

followed by a pointwise non-linearity like ReLU. Such a convolutional layer can be seen as $J+1$ MPLs, where each MPL is of the form

$$\mathbf{f} \to [\mathbf{f}; \frac{1}{n} A\mathbf{f}].$$

Notice that the action $\mathbf{f} \to \frac{1}{n} A\mathbf{f}$ is simply the action of a normalized sum aggregation. We first define $\varphi^{(0)}$ as the identity function, and then, we define $\varphi^{(t)} = [\cdot; \cdot]$ and $\phi(t)(a, b) = b$ for $0 < t \leq J$, to get the desired action. Lastly, we define

$$\varphi^{(t)}(\mathbf{f}) = \text{ReLu}(\mathbf{f}C)$$

for some $C \in \mathbb{R}^{(J+1)d \times p}$, where $\text{ReLu}(x) = \max(x, 0)$ is a pointwise non-linearity.

### D.3 Aggregation Schemes Expressivity Equivalency

We now show that alternative MPNN models have the same expressive power as MPNNs with our normalized sum aggregation. Denote $\mathcal{A}_L^{d_L} := \{\hat{\mathfrak{f}}(\varphi, \phi)_{-}^{(L)} : \mathcal{H}^L \mapsto \mathbb{R}^{d_L} | (\varphi, \phi) \text{ is an } L\text{-layer MPNN model}\}$ and $\mathcal{S}_L^{d_L} := \{\mathfrak{f}(\varphi)_{-}^{(L)} : \mathcal{H}^L \mapsto \mathbb{R}^{d_L} | \varphi \text{ is an } L\text{-layer MPNN model}\}$. It is clear that the alternative MPNN models are as expressive as our standard MPNN models. If we set the message function to be $\phi(a, b) := b$, the alternative normalized sum aggregation (Definition 55) is equal to the one in Definition 50. This means that $\mathcal{A}^{d_L} \subseteq \mathcal{S}_L^{d_L}$. We now prove Proposition 58, that shows $\mathcal{S}_L^{d_L} \subseteq \mathcal{A}_L^{d_L}$. It follows immediately that the sets $\{\hat{\mathfrak{F}}(\varphi, \phi, \psi, -, -) : \mathscr{P}(\mathcal{H}^L) \mapsto \mathbb{B}_r^d | (\varphi, \phi, \psi) \text{ is an } L\text{-layer alternative MPNN model with readout}\}$ and $\{\mathfrak{F}(\varphi, \psi, -, -) : \mathcal{H}^L \mapsto \mathbb{R}^{d_L} | (\varphi, \psi) \text{ is an } L\text{-layer MPNN model}\}$ are equal. We start by defining *function concatenation* and *function Cartesian product*, which we use in the proof of Proposition 58.

**Definition 57** (Function Concatenation). *Let $f : \mathcal{X} \mapsto \mathcal{Y}$ and $g : \mathcal{X} \mapsto \mathcal{Z}$ be two functions. We define* function concatenation *as the function $f \parallel g : \mathcal{X} \mapsto \mathcal{Y} \times \mathcal{Z}$ such that $p_{\mathcal{Y}} \circ (f \parallel g) = f$ and $p_{\mathcal{Z}} \circ (f \parallel g) = f$ where $p_{\mathcal{Y}}$, $p_{\mathcal{Z}}$ are the canonical projections from $\mathcal{Y} \times \mathcal{Z}$ to $\mathcal{Y}$ and $\mathcal{Z}$, respectively.*

Given $\{f_k\}_{k=1}^K$ a set of functions $f : \mathcal{X}_k \mapsto \mathcal{Y}_k$, we shortly denote $f_1 \parallel \ldots \parallel f_K$ as $\parallel_{k=1}^K f_k$.

**Proposition 58.** *Let $(\varphi, \phi)$ be an $L$-layer alternative MPNN model. Then, there exists an MPNN model $\varphi'$ such that,*

$$\hat{\mathfrak{f}}(\varphi, \phi, -, -)_{-}^{(L)} = \mathfrak{f}(\varphi', -, -)_{-}^{(L)}.$$

*Proof.* The proof is by induction.

*Induction Base.* For $L = 0$, $(\varphi, \psi) = (\varphi^{(0)})$, that is, the feature do not depend of the aggregation and therefore, the statement is trivial and we can jsut define $\varphi' = (\varphi'^{(0)})$ such that $\varphi'^{(0)} = \varphi^{(0)}$.

*Induction Assumption.* Presume that for any $L$-layer alternative MPNN model $(\varphi, \phi)$ there exists an $L$-layer MPNN model $\varphi'$ such that $\hat{\mathfrak{f}}(\varphi, \phi, -, -)_{-}^{(L)} = \mathfrak{f}(\varphi', -, -)_{-}^{(L)}$.

*Induction Step.* Let $(\varphi, \phi) = ((\varphi^{(t)})_{t \in [L+1]}, (\phi^{(t)})_{t=1}^{L+1})$ be an $L+1$-layers alternative MPNN model. Then, $((\varphi^{(t)})_{t \in [L]}, (\phi^{(t)})_{t=1}^L)$ is an $L$-layer MPNN model. By the induction assumption, there exists an $L$-layer MPNN model $\varphi'$ such that $\hat{\mathfrak{f}}(\varphi, \phi, -, -)_{-}^{(L)} = \mathfrak{f}(\varphi', -, -)_{-}^{(L)}$. Following the message function definition, we can write $\phi^{(L)}(a, b) = \sum_{k=1}^K \xi_{k,\text{rec}}(a) \xi_{k,\text{trans}}(b)$ for some Lipschitz continuous functions $\{\xi_{k,\text{rec}}\}_{k=1}^K$ and $\{\xi_{k,\text{trans}}\}_{k=1}^K$.

Define $\varphi''$, an $L+1$-layers MPNN model, as follows: for $0 \le t < L$ set $\varphi''^{(t)} := \varphi'^{(t)}$, and set $\varphi''^{(L)} := \zeta \circ \varphi'^{(L)}$, when $\zeta(x) := (x) \parallel (\parallel_{k=1}^K \xi_{k,\text{rec}}(x)) \parallel (\parallel_{k=1}^K \xi_{k,\text{trans}}(x)) \in \mathbb{R}^{(2K+1)p}$. Define $\varphi''^{(L+1)} = \varphi^{(L+1)} \circ \sigma$, where $\sigma : \mathbb{R}^{2(2K+1)p} \mapsto \mathbb{R}^{2p}$ is defines as follows; let $v = (v_i)_{i=1}^{2(K+1)p} \in \mathbb{R}^{2(2K+1)p}$ be a vector, then $\sigma(v) = ((v_i)_{i=1}^p, \sum_{j=1}^K v_{p+j} v_{(3K+2)p+j})$.

Let $(W, f)$ be a graphon-signal. In the following equations we use a shorten notation and do not write explicitly the dependence of the features on the graphon-signal, as all features depend on $(W, f)$. Then,

$$\mathfrak{f}(\varphi'')_x^{(L+1)} = \varphi'^{(L+1)}\left(\mathfrak{f}(\varphi'')_x^{(L)}, \int_{[0,1]} W(x, y) \mathfrak{f}(\varphi'')_y^{(L)} \mathrm{d}\lambda(y)\right)$$

$$= \varphi^{(L+1)} \circ \sigma\left(\zeta(\mathfrak{f}(\varphi')_x^{(L)}), \int_{[0,1]} W(x, y)(\zeta(\mathfrak{f}(\varphi')_y^{(L)}))dy)\right)$$

$$= \varphi^{(L+1)} \circ \sigma\left(\zeta(\hat{\mathfrak{f}}(\varphi, \phi)_x^{(L)}), \int_{[0,1]} W(x, y)(\zeta(\hat{\mathfrak{f}}(\varphi, \phi)_y^{(L)}))dy)\right)$$

$$= \varphi^{(L+1)}\left(\hat{\mathfrak{f}}(\varphi, \phi)_x^{(L)}, \sum_{k=1}^K \xi_{k,\text{rec}}(\hat{\mathfrak{f}}(\varphi, \phi)_x^{(L)}) \int_{[0,1]} W(x, y) \xi_{k,\text{trans}}(\hat{\mathfrak{f}}(\varphi, \phi)_y^{(L)} dy)\right)$$

$$= \varphi^{(L+1)}\left(\hat{\mathfrak{f}}(\varphi, \phi)_x^{(L)}, \int_{[0,1]} W(x, y) \sum_{k=1}^K \xi_{k,\text{rec}}(\hat{\mathfrak{f}}(\varphi, \phi)_x^{(L)}) \xi_{k,\text{trans}}(\hat{\mathfrak{f}}(\varphi, \phi)_y^{(L)}) dy)\right)$$

$$= \varphi^{(L+1)}\left(\hat{\mathfrak{f}}(\varphi, \phi)_x^{(L)}, \int_{[0,1]} W(x, y) \phi^{(L+1)}(\hat{\mathfrak{f}}(\varphi, \phi)_x^{(L)}, \hat{\mathfrak{f}}(\varphi, \phi)_y^{(L)}) dy\right)$$

$$= \hat{\mathfrak{f}}(\varphi, \phi)_x^{(L+1)}$$

$\square$

## D.4 LIPSCHITZ CONTINUITY WITH RESPECT TO THE CUT NORM

Levie, 2023, Theorem 4.1 (also Rauchwerger & Levie, 2025, Theorem 5.1.) states that alternative MPNNs (Definition 52) over the space of graphon-signal with respect to the cut distance $\delta_{\square}$ (Definition 38), without the Lipschitz continuous function $\varphi^{(0)}$, are Lipschitz continuous. Definition 52

without the first update function $\varphi^{(0)}$ is equivalent to the definition of MPNNs in Levie (2023); Rauchwerger & Levie (2025). The existence of the Lipschitz continuous function $\varphi^{(0)}$ in the model does affect the Lipschitz constant value, but does not affect its existence, since a composition of two Lipschitz continuous functions is a Lipschitz continuous function.

Since any standard MPNN (Definition 7) can be easily formulated as an alternative MPNN (Definition 52), Rauchwerger & Levie, 2025, Theorem 5.1 holds for standard MPNNs. Therefore, we rephrase Rauchwerger & Levie, 2025, Theorem 5.1 in terms of our definition of standard MPNNs. We slightly adjust the notations to be consistent with our own notations.

**Theorem 59** (MPNN Lipschitz Continuity with Respect to the Cut Distance). *Let $(\varphi, \psi)$ be an L-Layer MPNN with readout. Then, there exists a constant $C_{(\varphi,\psi)}$, that depends only on L, the number of layers, and the Lipschitz constants of model's update functions, such that*

$$\|\mathfrak{F}(\varphi, \psi, W, f) - \mathfrak{F}(\varphi, \psi, V, g)\|_2 \leq C_{(\varphi,\psi)} \cdot \delta_\square((W, f), (V, g)).$$

*for all $(W, f), (V, g) \in \mathcal{WL}_r^d$.*

# E  BASIC METRIC PROPERTIES

## E.1  THE WEAK* TOPOLOGY

Let $L \in \mathbb{N}_0$. we motivate the need to prove that $d_{\mathrm{IDM}}^L$ and $\mathbf{OT}_{d_{\mathrm{IDM}}^L}$ are well-defined, by the fact that the measures in $\mathcal{M}^L = \mathscr{M}_{\leq 1}(\mathcal{H}^L) = \mathscr{M}_{\leq 1}(\mathcal{H}^L, \mathcal{B}(\mathcal{H}^L))$ (and $\mathscr{P}(\mathcal{H}^L) = \mathscr{P}(\mathcal{H}^L, \mathcal{B}(\mathcal{H}^L)))$ are defined as functions $\mu : \mathcal{B}(\mathcal{H}^L) \to \mathbb{R}$. But for any topological space $\mathcal{X}$, its $\sigma$-algebra $\mathcal{B}(\mathcal{X})$ depends on the topology of $\mathcal{X}$. In the case of $\mathcal{H}^L$, the topology is the product topology, when $\mathcal{M}^i$, for $i \in [L]$ has the weak* topology, which makes it crucial for $\mathbf{OT}_{d_{\mathrm{IDM}}^i}$ to metrize the weak* topology on $\mathscr{M}_{\leq 1}(\mathcal{H}^L)$ and $\mathscr{P}(\mathcal{H}^L)$; otherwise the sets $\mathscr{M}_{\leq 1}(\mathcal{H}^L, \mathcal{B}(\mathcal{H}^L))$ and $\mathscr{M}_{\leq 1}(\mathcal{H}^L, d_{\mathrm{IDM}}^L)$ might be two different sets.

The well definiteness of the metric $d_{\mathrm{IDM}}^L$ on $\mathcal{H}^L$ and $\mathbf{OT}_{d_{\mathrm{IDM}}^L}$ on $\mathscr{P}(\mathcal{H}^L)$ follow similar arguments used in Böker et al. (2023). Let $(\mathcal{X}, d)$ be a complete separable metric space. Böker et al. (2023) showed that the unbalanced optimal transport, as we define it (see Definition 1) is indead a metric on $\mathscr{M}_{\leq 1}(S)$ by following the proof of Pele & Werman (2008).

**Lemma 60** (Böker et al. (2023), Corollary 21.). *Let $(S, d)$ be a separable metric space. Then, $\mathbf{OT}_d$ is a metric on $\mathscr{M}_{\leq 1}(S, d)$.*

Moreover, they show that this metric metrizes the weak* topology of $\mathscr{M}_{\leq 1}(S)$ by proving inequalities, that are known to hold for probability measures, for measures with total mass smaller then one.

**Lemma 61** (Böker et al. (2023), Lemma 22.). *Let $(\mathcal{X}, d)$ be a complete separable metric space. Then, for all $\mu, \nu \in \mathscr{M}_{\leq 1}(\mathcal{X})$,*

$$\mathbf{BL}(\mu, \nu) \leq \mathbf{K}(\mu, \nu) \leq \mathbf{OT}_d(\mu, \nu) \leq 2\mathbf{P}(\mu, \nu) \leq 4\sqrt{\mathbf{BL}(\mu, \nu)}.$$

Here $\mathbf{K}$ is the Kantorovich-Rubinshtein distance (Definition 47, $\mathbf{BL}$ is the Bounded-Lipschitz distance (Definition 46), $\mathbf{P}$ the Prokhorov metric (Definition 48). To prove Lemma 61 they follow proofs outlined in Schay (1974) and García-Palomares & Giné (1977), which use the duality of linear programming.

A direct result of Bogachev, 2007, Theorem 8.3.2 and Prokhorov, 1956, Theorem 1.11 is that if $(\mathcal{X}, d)$ is a complete separable space, then K and $\mathbf{P}$ metrize the weak* topology on $\mathscr{M}_{\leq 1}(\mathcal{X})$ and $\mathscr{P}(\mathcal{X})$. These facts together with Lemma 61 entail the following. If $(\mathcal{X}, d)$ is separable, then $\mathbf{OT}_d$ metrize the weak* topology on $\mathscr{M}_{\leq 1}(\mathcal{X})$ and $\mathscr{P}(\mathcal{X})$. We summarize this result in Lemma 62.

**Lemma 62.** *Let $(\mathcal{X}, d)$ be a complete separable metric space with Borel $\sigma$-algebra $\mathcal{B}$. Then $\mathbf{OT}_d$ is well defined and metrizes the weak* topology of $\mathscr{M}_{\leq 1}(\mathcal{X})$ and $\mathscr{P}(\mathcal{X})$.*

We can now follow the proof of Lemma 24. in Böker et al. (2023) and prove Theorem 4.

**Theorem 4.** *Let $L \in \mathbb{N}_0$. The metrics $d_{\mathrm{IDM}}^L$ on $\mathcal{H}^L$ and $\mathbf{OT}_{d_{\mathrm{IDM}}^L}$ on $\mathscr{P}(\mathcal{H}^L)$ and $\mathscr{M}_{\leq 1}(\mathcal{H}^L)$ are well-defined. Moreover, $\mathbf{OT}_{d_{\mathrm{IDM}}^L}$ metrizes the weak* topologies of $\mathscr{M}_{\leq 1}(\mathcal{H}^L)$ and $\mathscr{P}(\mathcal{H}^L)$.*

*Proof.* We start with the fact that $(\mathcal{H}^0, d_{\text{IDM}}^0) = (\mathbb{B}_r^p, \|\cdot\|_2)$ and therefore $d_{\text{IDM}}^0$ is well defined. As $\mathbb{B}_r^p$ is a complete separable metric space (a compact sub space of $\mathbb{R}^p$), by Lemma 62, convergence in $\mathbf{OT}_{d_{\text{IDM}}^0}$ is equivalent to weak* convergence on $\mathscr{M}_{\leq 1}(\mathcal{H}^0, d_{\text{IDM}}^0)$ and $\mathscr{P}(\mathcal{H}^0, d_{\text{IDM}}^0)$ which is just weak* convergence on $\mathscr{M}_{\leq 1}(\mathcal{H}^0, \mathcal{B}(\mathcal{H}^0))$ and $\mathscr{P}(\mathcal{H}^0, \mathcal{B}(\mathcal{H}^0))$, respectively. Hence, the topology induced by $\mathbf{OT}_{d_{\text{IDM}}^0}$ is equal to the weak* topology on $\mathscr{M}_{\leq 1}(\mathcal{H}^0, \mathcal{B}(\mathcal{H}^0))$ and $\mathscr{P}(\mathcal{H}^0, \mathcal{B}(\mathcal{H}^0))$ as both spaces are metrizable. The induction step follows the same arguments with the additional claim that $d_{\text{IDM}}^L$ is a product metric, which metrizes the product topology. $\square$

## E.2 THE COMPACTNESS OF THE SPACES OF IDMs AND DIDMs

Let $\mathcal{K}$ be a compact space. A well-established result (Kechris, 2012, Section 17) in measure theory states that the space, $\mathscr{M}_{\leq 1}(\mathcal{K})$, is equivalent to the set of non-negative Radon measures of total mass at most 1. The Riesz Representation Theorem (Rudin, 1986, Theorem 6.19) establishes that these measures correspond precisely to the non-negative linear functionals with norm at most 1 in the dual space of continuous real-valued functions on $\mathcal{K}$, $C(\mathcal{K}, \mathbb{R})$. The weak* topology on $\mathscr{M}_{\leq 1}(\mathcal{K})$ is defined as the minimal topology that ensures continuity of the mappings:

$$\int_{\mathcal{K}} f d\nu_n \mapsto \int_{\mathcal{K}} f d\nu$$

for all continuous real-valued functions $f$ on $\mathcal{K}$. A fundamental result asserts that $\mathscr{M}_{\leq 1}(\mathcal{K})$, when endowed with the weak* topology, forms a compact metrizable space. Moreover, the Borel $\sigma$-algebra generated by this weak* topology is identical to the conventional Borel structure on $\mathscr{M}_{\leq 1}(\mathcal{K})$ induced by the mappings:

$$\mathcal{A} \mapsto \nu(\mathcal{A}), \quad \mathcal{A} \in \mathcal{B}(\mathcal{K})$$

where $\mathcal{B}(\mathcal{K})$ denotes the Borel sets of $\mathcal{K}$ (Kechris, 2012, Section 17).

**Theorem 3.** *The spaces $\mathcal{H}^L$ and $\mathscr{P}(\mathcal{H}^L)$ are compact spaces for any $L \in \mathbb{N}_0$.*

*Proof.* The proof is done using induction.

*Induction Base.* Recall that $\mathcal{H}^0 = (\mathbb{B}_r^p, \|\cdot\|)$ is a compact metric space. As unbalanced optimal transport metrizes the weak* topology on $\mathscr{M}_{\leq 1}(\mathcal{H}^0)$ and $\mathscr{P}(\mathcal{H}^0)$, they both form compact metrizable spaces.

*Induction Assumption.* Presume that for any $0 < L$, the spaces $\mathcal{H}^i$, $\mathscr{M}_{\leq 1}(\mathcal{H}^i)$, and $\mathscr{P}(\mathcal{H}^i)$ are compact spaces for $i \in [L-1]$.

*Induction Step.* Let $0 < L$. Tychonoff's theorem (see Theorem 26) states that the product of any collection of compact topological spaces is compact with respect to the product topology. As $d_{\text{IDM}}^L$ metrize the product topology, we can combine Tychonoff's theorem with the induction assumption and conclude that $\mathcal{H}^L = \prod_{0 \leq i < L} \mathscr{M}_{\leq 1}(\mathcal{H}^i)$ is compact. We can now use the same argument as in the induction base, i.e., unbalanced optimal transport metrizes the weak* topology on $\mathscr{M}_{\leq 1}(\mathcal{H}^0)$ and $\mathscr{P}(\mathcal{H}^0)$, to conclude that both $\mathscr{M}_{\leq 1}(\mathcal{H}^L)$ and $\mathscr{P}(\mathcal{H}^L)$ with the topology induced by $\mathbf{OT}_{d_{\text{IDM}}^L}$ form compact spaces. $\square$

## E.3 THE COMPACTNESS OF THE SPACE OF GRAPHON-SIGNALS

We now show that the space of graphon-signals is compact under the DIDM mover's distance. The compactness of the graphon-signal space makes it possible to rephrase our generalization analysis directly on the graphon-signal space. More importantly, it allows us to approximate any real continuous function over the space of graphon-signals, rather than only functions over the space of graphon-signals that can be extended as real continuous functions over a space of DIDMs (see Appendix I).

To do so, we use results from Levie (2023); Rauchwerger & Levie (2025). Note that in our setting, signals are functions that map the interval $[0, 1]$ to a general compact subset of $\mathbb{R}^d$ rather than a sphere around 0, as in Levie (2023); Rauchwerger & Levie (2025). Nevertheless, their results are not affected by this change.

The DIDM mover's distance $\delta_{\mathrm{DIDM}}^L$ is a pseudometric on $\mathcal{WL}_r^d$. Consider the equivalence relation: $(W, f) \sim_L (V, g)$ if $\delta_{\mathrm{DIDM}}^L((W, f), (V, g)) = 0$. Then, the quotient space

$$\mathcal{WL}_r^d / \delta_{\mathrm{DIDM}}^L := \mathcal{WL} / \sim_L$$

of the equivalence classes $[(W, f)]_L$ is a metric space, with the metric $\delta_{\mathrm{DIDM}}^L([(W, f)], [(V, g)]) := \delta_{\mathrm{DIDM}}^L((W, f), (V, g))$. Notice that the equivalence relation $\sim$ is a metric identification (Appendix A.4.7). Notice that the equivalence relation $\sim_L$ is a metric identification (Appendix A.4.7). Recall that we similarly denote by $\widetilde{\mathcal{WL}_r^d}$ the graphon-signal space under metric identification $\sim$ (Appendix A.8) of the graphon-signal cut distance $\delta_\square$ (Definition 38).

The following theorem relates the convergence in $(\mathcal{WL}_r^d, \delta_{\mathrm{DIDM}}^L)$ to the convergence in $(\mathcal{WL}_r^d, \delta_\square)$.

**Theorem 63.** *For any fixed $L \in \mathbb{N}_0$, a sequence of graphon-signals $\{(W_i, f_i)\}_{i \in \mathbb{N}_0} \subset \mathcal{WL}_r^d$, and a graphon-signal $(W, f)$ the following holds,*

$$(W_i, f_i) \xrightarrow{\delta_\square} (W, f) \implies (W_i, f_i) \xrightarrow{\delta_{\mathrm{DIDM}}^L} (W, f)$$
$$\Updownarrow \qquad\qquad\qquad\qquad \Updownarrow$$
$$[(W_i, f_i)] \xrightarrow{\delta_\square} [(W, f)] \implies [(W_i, f_i)]_L \xrightarrow{\delta_{\mathrm{DIDM}}^L} [(W, f)]_L.$$

*Proof.* Let $(W_i, f_i) \xrightarrow{\delta_\square} (W, f)$, then, it follows from Theorem 59 that $\mathfrak{f}_{(W_i, f_i)} \to \mathfrak{f}_{(W, f)}$ for all MPNN model $\varphi$ with readout $\psi : \mathbb{R}^{d_L} \mapsto \mathbb{R}^d$, where $d > 0$, which entails, using Lemma 11, that $\mathfrak{h}_{\Gamma_{(W_i, f_i), L}} \to \mathfrak{h}_{\Gamma_{(W, f), L}}$ for all MPNN model $\varphi$ with readout $\psi : \mathbb{R}^{d_L} \mapsto \mathbb{R}^d$, where $d > 0$. By Corollary 12, we have that $\Gamma_{(W_i, f_i), L} \to \Gamma_{(W, f), L}$ (in the weak* topology). According to Theorem 4, $\mathbf{OT}_{d_{\mathrm{IDM}}^L}(\Gamma_{(W_i, f_i), L}, \Gamma_{(W, f), L}) \to 0$. By Definition 5, $(W_i, f_i) \xrightarrow{\delta_{\mathrm{DIDM}}^L} (W, f)$. The other implications are straightforward, $\qquad\square$

**Theorem 64.** *The pseudometric space $(\mathcal{WL}_r^d, \delta_{\mathrm{DIDM}}^L)$ and the metric space $(\mathcal{WL}_r^d / \delta_{\mathrm{DIDM}}^L, \delta_{\mathrm{DIDM}}^L)$ are compact.*

*Proof.* For any $L \in \mathbb{N}_0$, Theorem 63 implies that the pseudometric topology of $(\mathcal{WL}_r^d, \delta_{\mathrm{DIDM}}^L)$ is cooraser than pseudometric topology of $(\mathcal{WL}_r^d, \delta_\square)$. Thus, since $(\mathcal{WL}_r^d, \delta_\square)$ is compact (Theorem 39), $(\mathcal{WL}_r^d, \delta_{\mathrm{DIDM}}^L)$ must be compact.

Let $\{U_\alpha\}_\alpha$ be an open cover of $\mathcal{WL}_r^d / \delta_{\mathrm{DIDM}}^L$. The open sets in the pseudometric space are exactly the sets of the form $\pi^{-1}(\mathcal{A})$, where $\mathcal{A} \in \mathcal{X} / \sim$ is open, thus $\{\pi^{-1}(U_\alpha)\}_\alpha$ is an open cover of $\mathcal{WL}_r^d$ and thus has a finite cover $\{\pi^{-1}(U_{\alpha_i})\}_i$. Since the quotient map is surjective,

$$\mathcal{WL}_r^d / \delta_{\mathrm{DIDM}}^L = \pi(\pi^{-1}(\mathcal{WL}_r^d)) = \pi(\pi^{-1}(\cup_i U_{\alpha_i})) \subseteq \cup_i U_{\alpha_i}.$$

This means $\{U_{\alpha_i}\}_i$ is a finite cover of $\mathcal{WL}_r^d / \delta_{\mathrm{DIDM}}^L$. Thus, the quotient space $(\mathcal{WL}_r^d / \delta_{\mathrm{DIDM}}^L, \delta_{\mathrm{DIDM}}^L)$ is also compact. $\qquad\square$

**Corollary 65.** *The space of computation DIDMs $\Gamma_{(W, f), L}(\mathcal{WL}_r^d) \subsetneq \mathscr{P}(\mathcal{H}^L)$ is compact.*

*Proof.* By Theorem 64, the space of graphon-signals, $\mathcal{WL}_r^d$, is compact. By the definition of $\delta_{\mathrm{DIDM}}^L$ (Definition 5) the mapping $\Gamma_{(W, f), L} : \mathcal{WL}_r^d \mapsto \mathscr{P}(\mathcal{H}^L)$ is continuous. Thus $\Gamma_{(W, f), L}(\mathcal{WL}_r^d)$ is compact, since a continuous function between two metric spaces sends compact sets to compact sets. $\qquad\square$

## F COMPUTABILITY OF OUR METRICS

First, we extend Schmitzer & Schnörr, 2013, Proposition 4.5 for couplings between measures with unequal mass, by follows the steps in the original proof. We emphasize that in Proposition 66, $\Gamma$ and $\gamma$ do not refer to Definition 2.

**Proposition 66.** *For two discrete sets $\mathcal{A}$, $\mathcal{C}$ and two measurable maps $\phi_a : \mathcal{X} \to \mathcal{A}$, $\phi_b : \mathcal{Y} \to \mathcal{C}$ denote by $\phi$ the product map $\phi(x, y) = (\phi_a(x), \phi_b(y))$. Then one finds*

$$\phi_* \Gamma(\mu, \nu) = \Gamma(\phi_{a*} \mu, \phi_{b*} \nu)$$

*when $\phi_* \Gamma(\mu, \nu) := \{\gamma : \gamma = \phi_* \gamma' : \gamma' \in \Gamma(\mu, \nu)\}$.*

*Proof.* Assume $\|\mu\| \leq \|\nu\|$. For any $\gamma \in \Gamma(\mu, \nu)$ we get

$$(\phi_* \gamma)(\mathcal{S}) = \gamma(\phi^{-1}(\mathcal{S})) \geq 0.$$

when $\mathcal{S} \subseteq \mathcal{A} \times \mathcal{C}$ a measurable subset, and

$$(\phi)_* \gamma(\mathcal{S}_\mathcal{A} \times \mathcal{C}) = \gamma(\phi_a^{-1}(\mathcal{S}_\mathcal{A}) \times \mathcal{X}) = (p_1)_* \gamma(\phi_a^{-1}(\mathcal{S}_\mathcal{A}))$$
$$= \mu(\phi_a^{-1}(\mathcal{S}_\mathcal{A})) = (\phi_a)_* \mu(\mathcal{S}_\mathcal{A})$$

when $\mathcal{S}_\mathcal{A} \subseteq \mathcal{A}$ a measurable subset and analogous for $\mathcal{S}_\mathcal{C} \subseteq \mathcal{C}$ a measurable subset

$$(\phi)_* \gamma(\mathcal{A} \times \mathcal{S}_\mathcal{C}) = \gamma(\mathcal{X} \times \phi_b(\mathcal{S}_\mathcal{C}) = (p_1)_* \gamma(\phi_a^{-1}(\mathcal{S}_\mathcal{C}))$$
$$\leq \nu(\phi_a^{-1}(\mathcal{S}_\mathcal{C})) = (\phi_a)_* \nu(\mathcal{S}_\mathcal{C}).$$

Thus $(\phi)_* \Gamma(\mu, \nu) \subseteq \Gamma((\phi_a)_* \mu, (\phi_b)_* \nu)$.

We now show by construction for any $\rho \in \Gamma(\phi_{a*} \mu, \phi_{b*} \nu)$ the existence of some $\gamma \in \Gamma(\mu, \nu)$ such that $\rho = (\phi)_* \gamma$. For any element $(a, c) \in \mathcal{A} \times \mathcal{C}$ construct the pre-image measure

$$\gamma_{(a,c)}(x, y) = \begin{cases} 0 & \text{if } \rho(a, c) = 0 \vee (a, c) \neq \phi(x, y) \\ \frac{\mu(x)\nu(y)}{((\phi_a)_* \mu)(a)((\phi_b)_* \nu)(c)} \rho(a, c) & \text{else} \end{cases}$$

where this element wise definition for each $(x, y)$ is extended to all subsets of $\mathcal{X} \times \mathcal{Y}$ by

$$\gamma_{(a,c)}(\mathcal{S}) = \sum_{(x,y) \in \mathcal{S}} \gamma_{(a,c)}(x, y)$$

when $\mathcal{S}$ is any measurable subset of $\mathcal{X} \times \mathcal{Y}$. Now consider $\gamma = \sum_{(a,c) \in \mathcal{A} \times \mathcal{C}} \gamma_{(a,c)}$. First verify that it is indeed contained in $\Gamma(\mu, \nu)$:

$$\gamma(\mathcal{S}) \geq 0 : \forall \text{ measurable } \mathcal{S} \subseteq \mathcal{X} \times \mathcal{Y}.$$

since $\gamma(x, y) \geq 0$ for all $(x, y)$. Furthermore

$$\gamma(\mathcal{S}_{\mathcal{X}} \times \mathcal{Y}) = \sum_{\substack{x \in \mathcal{S}_{\mathcal{X}} \\ y \in \mathcal{Y}}} \sum_{\substack{(a,c) \in \mathcal{A} \times \mathcal{C}: \\ \phi(x,y)=(a,c), \\ \rho(a,c)>0}} \frac{\mu(x)\nu(y)}{((\phi_a)_*\mu)(a)((\phi_b)_*\nu)(c)}\rho(a,c)$$

$$= \sum_{x \in \mathcal{S}_{\mathcal{X}}} \sum_{\substack{c \in \mathcal{C}: \\ \rho(\phi_a(x),c)>0}} \frac{\mu(x)(\sum_{y:\phi_b(y)=c}\nu(y))}{((\phi_a)_*\mu)(\phi_a(x))((\phi_b)_*\nu)(c)}\rho(\phi_a(x),c)$$

$$= \sum_{x \in \mathcal{S}_{\mathcal{X}}} \sum_{\substack{c \in \mathcal{C}: \\ \rho(\phi_a(x),c)>0}} \frac{\mu(x)\nu(\phi_b^{-1}(c))}{((\phi_a)_*\mu)(\phi_a(x))((\phi_b)_*\nu)(c)}\rho(\phi_a(x),c)$$

$$= \sum_{x \in \mathcal{S}_{\mathcal{X}}} \frac{\mu(x)}{((\phi_a)_*\mu)(\phi_a(x))} \sum_{\substack{c \in \mathcal{C}: \\ \rho(\phi_a(x),c)>0}} \rho(\phi_a(x),c)$$

$$= \sum_{x \in \mathcal{S}_{\mathcal{X}}} \frac{\mu(x)}{((\phi_a)_*\mu)(\phi_a(x))}(p_1)_*\rho(\phi_a(x))$$

$$= \sum_{x \in \mathcal{S}_{\mathcal{X}}} \frac{\mu(x)}{((\phi_a)_*\mu)(\phi_a(x))}(\phi_a)_*\mu(\phi_a(x)) = \sum_{x \in \mathcal{S}_{\mathcal{X}}} \mu(x) = \mu(\mathcal{S}_{\mathcal{X}})$$

for all measurable subsets $\mathcal{S}_{\mathcal{X}} \subseteq \mathcal{X}$ and likewise

$$\gamma(X \times \mathcal{S}_{\mathcal{Y}}) \leq \nu(\mathcal{S}_{\mathcal{Y}}).$$

$$\gamma(\mathcal{X} \times \mathcal{S}_{\mathcal{Y}}) = \sum_{\substack{y \in \mathcal{S}_{\mathcal{Y}} \\ x \in \mathcal{X}}} \sum_{\substack{(a,c) \in \mathcal{A} \times \mathcal{C}: \\ \phi(x,y)=(a,c), \\ \rho(a,c)>0}} \frac{\mu(x)\nu(y)}{((\phi_a)_*\mu)(a)((\phi_b)_*\nu)(c)}\rho(a,c)$$

$$= \sum_{y \in \mathcal{S}_{\mathcal{Y}}} \sum_{\substack{a \in \mathcal{A}: \\ \rho(a,\phi_b(y))>0}} \frac{(\sum_{x:\phi_b(x)=a}\mu(x))\nu(y)}{((\phi_a)_*\mu)(a)((\phi_b)_*\nu)(\phi_b(y))}\rho(\phi_a(x),c)$$

$$= \sum_{y \in \mathcal{S}_{\mathcal{Y}}} \sum_{\substack{a \in \mathcal{A}: \\ \rho(a,\phi_b(y))>0}} \frac{\mu(\phi_a^{-1}(a))\nu(y)}{((\phi_a)_*\mu)(a)((\phi_b)_*\nu)(\phi_b(y))}\rho(a,\phi_b(y))$$

$$= \sum_{y \in \mathcal{S}_{\mathcal{Y}}} \frac{\nu(y)}{((\phi_b)_*\nu)(\phi_b(y))} \sum_{\substack{a \in \mathcal{A}: \\ \rho(a,\phi_b(y))>0}} \rho(a,\phi_b(y))$$

$$= \sum_{y \in \mathcal{S}_{\mathcal{Y}}} \frac{\nu(x)}{((\phi_b)_*\nu)(\phi_b(y))}(p_2)_*\rho(\phi_a(y))$$

$$\leq \sum_{y \in \mathcal{S}_{\mathcal{Y}}} \frac{\nu(y)}{((\phi_b)_*\nu)(\phi_b(y))}((\phi_b)_*\nu)(\phi_b(y)) = \sum_{y \in \mathcal{S}_{\mathcal{Y}}} \nu(x) = \nu(\mathcal{S}_{\mathcal{A}})$$

for all measurable subsets $\mathcal{S}_{\mathcal{Y}} \subseteq \mathcal{Y}$. Now check whether $\phi_*\gamma = \rho$:

$$
\begin{aligned}
(\phi)_*\gamma(\mathcal{S}) = \gamma(\phi^{-1}(\mathcal{S})) &= \sum_{(x,y)\in\phi^{-1}(\mathcal{S})} \gamma(x,y) \\
&= \sum_{\substack{(x,y)\in\phi^{-1}(\mathcal{S}): \\ \rho(\phi(x,y))>0}} \frac{\mu(x)\nu(y)}{((\phi_a)_*\mu)(\phi_a(x))((\phi_b)_*\nu)(\phi_b(y))}\rho(\phi(x,y)) \\
&= \sum_{\substack{(a,c)\in\mathcal{S} \\ \rho((a,c))>0}} \sum_{(x,y)\in\phi^{-1}((a,c))} \frac{\mu(x)\nu(y)}{((\phi_a)_*\mu)(a)((\phi_b)_*\nu)(c)}\rho(a,c) \\
&= \sum_{\substack{(a,c)\in\mathcal{S} \\ \rho((a,c))>0}} \frac{(\sum_{x\in\phi_a^{-1}(a)}\mu(x))(\sum_{y\in\phi_b^{-1}(c)}\nu(y))}{((\phi_a)_*\mu)(a)((\phi_b)_*\nu)(c)}\rho(a,c) \\
&= \sum_{\substack{(a,c)\in\mathcal{S} \\ \rho((a,c))>0}} \rho(a,c) = \rho(\mathcal{S})
\end{aligned}
$$

when $\mathcal{S}$ is a measurable subset of $\mathcal{X}\times\mathcal{Y}$. Consequently any $\rho\in\Gamma((\phi_a)_*\mu, (\phi_b)_*\nu)$ is also contained in $\phi_*\Gamma(\mu,\nu)$ and the two sets are equal. □

Chen et al., 2022, Lemma A.1 is now easily extended to measures with unequal mass. Although the proof is equivalent to the one in Chen et al. (2022), we will add it here for completeness.

**Lemma 67.** *Let $\mathcal{X}, \mathcal{Y}$ be finite metric spaces and let $(\mathcal{Z}, d_{\mathcal{Z}})$ be a complete and separable metric space. Let $\phi_{\mathcal{X}}: \mathcal{X} \to \mathcal{Z}$ and $\phi_{\mathcal{Y}}: \mathcal{Y} \to \mathcal{Z}$ be measurable maps. Consider any $\mu_{\mathcal{X}} \in \mathscr{M}_{\leq 1}(\mathcal{X})$ and $\mu_{\mathcal{Y}} \in \mathscr{M}_{\leq 1}(\mathcal{Y})$. Then, we have that*

$$
\mathbf{OT}_d((\phi_{\mathcal{X}})_*\mu_{\mathcal{X}}, (\phi_{\mathcal{Y}})_*\mu_{\mathcal{Y}}) = \inf_{\gamma\in\Gamma(\mu_{\mathcal{X}},\mu_{\mathcal{Y}})} \int_{\mathcal{X}\times\mathcal{Y}} d(\phi_{\mathcal{X}}(x), \phi_{\mathcal{Y}}(y))\gamma(dx\times dy).
$$

*Proof.* Since $\mathcal{X}$ and $\mathcal{Y}$ are finite, $\phi_{\mathcal{X}}(\mathcal{X})$ and $\phi_{\mathcal{Y}}(\mathcal{Y})$ are discrete sets. Then, if we let $\phi := \phi_{\mathcal{X}}\times\phi_{\mathcal{Y}}$,

$$
(\phi)_*\Gamma(\mu_{\mathcal{X}}, \mu_{\mathcal{Y}}) = \Gamma((\phi_{\mathcal{Y}})_*\mu_{\mathcal{X}}.(\phi_{\mathcal{Y}})_*\mu_{\mathcal{Y}})
$$

follows directly from Proposition 66.

Hence,

$$
\begin{aligned}
\mathbf{OT}_d((\phi_{\mathcal{X}})_*\mu_{\mathcal{X}}, (\phi_{\mathcal{Y}})_*\mu_{\mathcal{Y}}) &= \inf_{\gamma\in\Gamma((\phi_{\mathcal{X}})_*\mu_{\mathcal{X}},(\phi_{\mathcal{Y}})_*\mu_{\mathcal{Y}})} \int_{\mathcal{Z}\times\mathcal{Z}} d_Z(z_1,z_2)\gamma(dz_1\times dz_2) \\
&= \inf_{\gamma\in\Gamma(\mu,\nu)} \int_{\mathcal{Z}\times\mathcal{Z}} d_{\mathcal{Z}}(z_1,z_2)\phi_*\gamma(dz_1\times dz_2) \\
&= \inf_{\gamma\in\Gamma(\mu,\nu)} \int_{\mathcal{X}\times\mathcal{Y}} d_{\mathcal{Z}}(\phi_{\mathcal{X}}(x), \phi_{\mathcal{Y}}(y))\gamma(dx\times dy).
\end{aligned}
$$

□

Now, following the constraction of the algorithms used to compute the Weisfeiler-Lehman metric in Chen et al. (2022) and the metrics presented in Grebík & Rocha (2021), we phrase an algorithm for computing $\delta_{\mathrm{DIDM}}^L$ for $L \in \mathbb{N}_0$. We note that instead of using a min-cost-flow algorithm Pele & Werman (2009) to solve the unbalanced optimal transport problem, we use linear programming Flamary et al. (2021) as it is more convenient when working with real values (instead of integers). The unbalaced optimal transport problem can casted into a regular optimal transport problem by adding reservoir points in which the surplus mass is sent Chapel et al. (2020).

**Theorem 6.** *For any fixed $L \in \mathbb{N}_0$, $\delta_{\mathrm{DIDM}}^L$ between any two graph-signals $(G, \mathbf{f})$ and $(H, \mathbf{g})$ can be computed in time polynomial in $L$ and the size of $G$ and $H$, namely $O(L \cdot N^5 \log(N))$, where $N = \max(|V(G)|, |V(H)|)$.*

*Proof.* From Lemma 67 we have that

$$\delta^L_{\text{DIDM}}((G,\mathbf{f}),(H,\mathbf{g})) := \mathbf{OT}\left(\Gamma_{(G,\mathbf{f}),L},\Gamma_{(H,\mathbf{g}),L}\right) = \mathbf{OT}\left((\gamma_{(G,\mathbf{f}),L})_* \lambda_{V(G)},(\gamma_{(H,\mathbf{g}),L})_* \lambda_{V(H)}\right)$$

$$= \inf_{\gamma \in \Gamma(\lambda_{V(G)},\lambda_{V(H)})} \int_{V(G) \times V(H)} d^L_{\text{IDM}}\left(\gamma_{(G,\mathbf{f}),L}(x),\gamma_{(H,\mathbf{g}),L}(y)\right) \gamma(dx \times dy).$$

In order to compute $\delta^L_{\text{DIDM}}((G,\mathbf{f}),(H,\mathbf{g}))$, we must first compute $d^L_{\text{IDM}}\left(\gamma_{(G,\mathbf{f}),L}(x),\gamma_{(H,\mathbf{g}),L}(y)\right)$ for each $x \in V(G)$ and $y \in V(H)$. To do this, we introduce some notation. For each $i = 1, \ldots, L$, we let $C_i, D_i$ denote the $|V(G)| \times |V(H)|$ matrices such that for each $x \in V(G)$ and $y \in V(H)$,

$$C_i(x,y) := d^i_{\text{IDM}}\left(\gamma_{(G,\mathbf{f}),i}(x),\gamma_{(H,\mathbf{g}),i}(y)\right), D_i(x,y) := \mathbf{OT}^i\left(\gamma_{(G,\mathbf{f}),i}(x)(i),\gamma_{(H,\mathbf{g}),i}(y)(i)\right).$$

We also let $C_0$ denote the matrix such that $C_0(x,y) := \|\mathbf{f}(x) - \mathbf{g}(y)\|_2$ for each $x \in V(G)$ and $y \in V(H)$. Then, our task is to compute the matrix $C_L$. For this purpose, we consecutively compute the matrices $C_i$ and $D_i$ for $i = 1, \ldots, L$. Given matrix $C_{i-1}$, since $\gamma_{(G,\mathbf{f}),i}(x)(i) = \left(\gamma_{(G,\mathbf{f}),i-1}(x)\right)_* \nu_{(G,\mathbf{f})}$ and $\gamma_{(H,\mathbf{g}),i}(y)(i) = \left(\gamma_{(H,\mathbf{g}),i-1}(y)\right)_* \nu_{(H,\mathbf{g})}$, computing

$$\mathbf{OT}^i\left(\gamma_{(G,\mathbf{f}),i}(x)(i),\gamma_{(H,\mathbf{g}),i}(y)(i)\right)$$

$$= \inf_{\gamma \in \Gamma(\nu_{(G,\mathbf{f})},\nu_{(H,\mathbf{g})})} \int_{V(G) \times V(H)} d^{i-1}_{\text{IDM}}\left(\gamma_{(G,\mathbf{f}),i-1}(x),\gamma_{(H,\mathbf{g}),i-1}(y)\right) \gamma(dx \times dy).$$

is reduced to solving the optimal transport problem with $C_{i-1}$ as the cost matrix and $\nu_{(G,\mathbf{f})}$ and $\nu_{(H,\mathbf{g})}$ as the source and target distributions, which can be done in $O(N^3 \log(N))$ time (Flamary et al., 2021; Chapel et al., 2020). Thus, for each $i$, computing $D_i$ given that we know $C_{i-1}$ requires $O(N^2 \cdot N^3 \log(N))$. To get $C_i$, all that remains is to compute the sum $D_i + C_{i-1}$. Finally, we need $O(N^3 \log(N))$ time to compute $\delta^L_{\text{DIDM}}((G,\mathbf{f}),(H,\mathbf{g}))$ based on solving an optimal transport problem with cost matrix $C_L$ and with $\lambda_{V(G)}$ and $\lambda_{V(H)}$ being the source and target distributions.

Therefore, the total time needed to compute $\delta^L_{\text{DIDM}}((G,\mathbf{f}),(H,\mathbf{g}))$ is

$$L \cdot O(N^5 \log(N)) + O(N^3 \log(N)) = O(L \cdot N^5 \log(N)).$$

For any $N \in \mathbb{N}_0$, $\delta^L_{\text{DIDM}}$ generates a distance between two graph-signls with number of vertices bounded by $N$. □

## G  EQUIVALENCY OF MPNNS ON GRAPHS, GRAPHONS, AND DIDMS

Here, we show that, first, for a graph-signal $(G,\mathbf{f})$, the output of an MPNN on $G$ is equal to the output of the MPNN on the corresponding induced graphon-signal $(W_G, f_\mathbf{f})$, similarly to of Böker et al., 2023, Appendix C.1.

**Lemma 68.** *Let $(G,\mathbf{f})$ be a graph-signal and $\varphi$ be an L-layer MPNN model. Let $(I_v)_{v \in V(G)}$ be the partition of $[0,1]$ used in the construction of $(W_G, f_\mathbf{f})$ from $(G,\mathbf{f})$. Then, for all $t \in [L], v \in V(G)$, and $x \in I_v$,*

$$\mathfrak{f}^{(t)}_x := \mathfrak{f}(\varphi, W_G, f_\mathbf{f})^{(t)}_x = \mathfrak{g}(\varphi, G, \mathbf{f})^{(t)}_v =: \mathfrak{g}^{(t)}_v.$$

*Proof.* We prove the claim by induction on $t$.

*Base of the induction.* For all $v \in V(G)$ and $x \in I_v$,

$$\mathfrak{f}^{(0)}_x = \varphi^{(0)} \circ f_\mathbf{f}(x) = \varphi^{(0)} \circ \mathbf{f}(v) = \mathfrak{g}^{(0)}_v.$$

*Induction step.* The induction assumption is that $\mathfrak{f}_x^{(t-1)} = \mathfrak{g}_v^{(t-1)}$ for all $v \in V(G)$ and $x \in I_v$. Then, for all $v \in V(G)$ and $x \in I_v$, we have

$$\mathfrak{f}_x^{(t)} = \varphi^{(t)}\left(\mathfrak{f}_x^{(t-1)}, \int_{[0,1]} W_G(x,y)\mathfrak{f}_y^{(t-1)}dy\right) = \varphi^{(t)}\left(\mathfrak{f}_x^{(t-1)}, \sum_{u \in V(G)} \int_{I_u} W_G(x,y)\mathfrak{f}_y^{(t-1)}dy\right)$$

$$= \varphi^{(t)}\left(\mathfrak{g}_v^{(t-1)}, \sum_{u \in V(G)} \int_{I_u} W_G(x,y)\mathfrak{g}_u^{(t-1)}dy\right)$$

$$= \varphi^{(t)}\left(\mathfrak{g}_v^{(t-1)}, \frac{1}{|V(G)|} \sum_{u \in \mathcal{N}(v)} \mathfrak{g}_u^{(t-1)}\right) = \mathfrak{g}_v^{(t)}.$$

$\square$

**Lemma 69.** *Let $(G, \mathbf{f})$ be a graph, let $(\varphi, \psi)$ be an L-layer MPNN model with readout. Let $(W_G, f_{\mathbf{f}})$ be the induced graphon of $(G, \mathbf{f})$. Then,*

$$\mathfrak{G} := \mathfrak{G}(\varphi, \psi, G, \mathbf{f}) = \mathfrak{F}(\varphi, \psi, W_G, f_{\mathbf{f}}) =: \mathfrak{F}.$$

*Proof.* Let $(I_v)_{v \in V(G)}$ be the partition of $[0,1]$ used in the construction $(W_G, f_{\mathbf{f}})$ from $(G, \mathbf{f})$. With Lemma 68, we get

$$\mathfrak{F} = \psi\left(\int_{[0,1]} \mathfrak{f}_x^{(L)} d\lambda(x)\right) = \psi\left(\sum_{v \in V(G)} \int_{I_v} \mathfrak{f}_x^{(L)} d\lambda(x)\right)$$

$$= \psi\left(\sum_{v \in V(G)} \int_{I_v} \mathfrak{g}_v^{(L)} d\lambda(x)\right)$$

$$= \psi\left(\frac{1}{|V(G)|} \sum_{v \in V(G)} \mathfrak{g}_v^{(L)}\right) = \mathfrak{G}.$$

$\square$

**Theorem 70** (Change of Variable Formula). *Let $\mathcal{X}_1, \mathcal{X}_2$ be two measurable spaces and $\mu$ a measure on $\mathcal{X}_1$. A measurable function $g$ on $\mathcal{X}_2$ is integrable with respect to the pushforward measure $f_*(\mu)(\mathcal{A}) = \mu(f^{-1}(\mathcal{A})$ if and only if the composition $g \circ f$ is integrable with respect to the measure $\mu$. In that case, the integrals coincide, i.e.,*

$$\int_{\mathcal{X}_2} g \, d(f_* \mu) = \int_{\mathcal{X}_1} g \circ f \, d\mu.$$

*Note that in the previous formula $\mathcal{X}_1 = f^{-1}(\mathcal{X}_2)$.*

The following lemma is related to the *absolute continuity* of weighted Lebesgue measures with respect to the Lebesgue measure.

**Lemma 71** (Billingsley (1995), Theorem 16.11.). *Let $\delta : [0,1] \mapsto \mathbb{R}$ be a nonnegative measurable function and $\mathcal{A} \subseteq [0,1]$ be any measurable set. Denote the measure $\nu_\delta(\mathcal{A}) := \int_{\mathcal{A}} \delta d\lambda$. Then, a measurable function $f : [0,1] \mapsto \mathbb{R}$ is integrable with respect to $\nu_\delta(\mathcal{A})$ if and only if $f\delta$ is integrable with respect to $\lambda$, in which case $\int_{\mathcal{A}} f d\nu_\delta = \int_{\mathcal{A}} f\delta d\lambda$.*

Now, we use Lemma 71 to show that the output of an MPNN without readout on a graphon-signal $(W, f)$ equals the output of the MPNN on the corresponding distribution of computation IDMs $\Gamma_{(W,f)}$ of $(W, f)$.

**Lemma 72.** *Let $(W, f) \in \mathcal{WL}_r^d$ a graphon-signal and $\varphi$ be a L-layer MPNN model. Then, for every $t \in [L]$ and almost every $x \in [0,1]$,*

$$\mathfrak{f}_x^{(t)} = \mathfrak{f}(\varphi, W, f)_x^{(t)} = \mathfrak{h}^{(t)}(\varphi)_{\gamma_{(W,f),t}(x)} = \mathfrak{h}_{\gamma_{(W,f),t}(x)}^{(t)}$$

*Proof.* We prove by induction on $t$.

*Induction Base.* We have by definition

$$\mathfrak{f}_x^{(0)} = \varphi^{(0)} \circ f(x) = \mathfrak{h}_{\gamma_{(W,f)},0}^{(0)}(x).$$

*Induction Assumption.* Let $1 \leq t \leq L$. We assume that

$$\mathfrak{f}_x^{(t-1)} = \mathfrak{h}_{\gamma_{(W,f)},t-1}^{(t-1)}(x)$$

for almost every $x \in [0, 1]$.

*Induction Step.* We have

$$\mathfrak{f}_x^{(t)} = \varphi^{(t)} \left( \mathfrak{f}_x^{(t-1)}, \int_{[0,1]} W(x,y)\mathfrak{f}_y^{(t-1)}\mathrm{d}\lambda(y) \right)$$

Then, by [Lemma 71] $\forall x \in [0,1] : \mathfrak{f}_-^{(t-1)}$ is integrable with respect to $\nu_{W(x,-)}(\mathcal{A}) := \int_{\mathcal{A}} W(x,y)\mathrm{d}\lambda(y)$ if and only if $\mathfrak{f}_-^{(t-1)} W(x,y)$ is integrable with respect to $\lambda$, in which case $\int_{\mathcal{A}} \mathfrak{f}_y^{(t-1)} d\nu_{W(x,-)}(y) = \int_{\mathcal{A}} \mathfrak{f}_y^{(t-1)} W(x,y)d\lambda(y)$. So,

$$\mathfrak{f}_x^{(t)} = \varphi^{(t)} \left( \mathfrak{f}_x^{(t-1)}, \int_{[0,1]} \mathfrak{f}_y^{(t-1)} d\nu_{W(x,-)}(y) \right) = (*)$$

Hence, by the induction assumption, i.e., $\forall x \in [0,1] : \mathfrak{f}_x^{(t-1)} = \mathfrak{h}_{\gamma_{(W,f)},t-1}^{(t-1)}(x)$, we have

$$(*) = \varphi^{(t)} \left( \mathfrak{h}_{\gamma_{(W,f)},t-1}^{(t-1)}(x), \int_{[0,1]} \mathfrak{h}_{\gamma_{(W,f)},t-1}^{(t-1)}(y) d\nu_{W(x,-)}(y) \right)$$

$$= \varphi^{(t)} \left( \mathfrak{h}_{\gamma_{(W,f)},t-1}^{(t-1)}(x), \int_{[0,1]} \mathfrak{h}_-^{(t-1)} \circ \gamma_{(W,f),t-1}(y) d\nu_{(W,f)(x,-)}(y) \right)$$

$$= \varphi^{(t)} \left( \mathfrak{h}_{\gamma_{(W,f)},t-1}^{(t-1)}(x), \int_{\mathcal{H}^{t-1}} \mathfrak{h}_-^{(t-1)} d(\gamma_{(W,f),t-1})_* \nu_{(W,f)(x,-)} \right) = (**)$$

Once again, by [Lemma 71] $\forall x \in [0,1] : x \mapsto x$ is integrable with respect to $\nu_{W(x,-)}(\mathcal{A})$ if and only if $W(x,y)$ is integrable with respect to $\lambda$, in which case $\int_{\mathcal{A}} d\nu_{W(x,-)} = \int_{\mathcal{A}} W(x,y)d\lambda(y)$. So, for all $\mathcal{C} \in \mathcal{B}(\mathcal{H}^{t-1})$

$$(\gamma_{(W,f),t-1})_* \nu_{W(x,-)}(\mathcal{C}) = \int_{\gamma_{(W,f),t-1}^{-1}(\mathcal{C})} W(x,y)d\lambda(y)$$

$$= \int_{\gamma_{(W,f),t-1}^{-1}(\mathcal{C})} d\nu_{W(x,-)} = (\gamma_{(W,f),t-1})_* \nu_{W(x,-)}(\mathcal{C}).$$

where $\mathcal{C} \in \mathcal{H}^{t-1} : (\gamma_{(W,f),t-1})_* \nu_{(W,f)(\mathcal{C})(x,-)} = \nu_{(W,f)(x,-)}(\gamma_{(W,f),t-1}^{-1}(\mathcal{C}))$ is the push forward measure of $\nu_{(W,f)(x,-)}$. The third equality result from the change of variable formula ([Theorem 70]). Therefore,

$$(**) = \varphi^{(t)} \left( \mathfrak{h}_{\gamma_{(W,f)},t-1}^{(t-1)}(x), \int_{\mathcal{H}^{t-1}} \mathfrak{h}_-^{(t-1)} d\gamma_{(W,f),t}(x)(t) \right)$$

$$= \varphi^{(t)} \left( \mathfrak{h}_{p_{t,t-1}(\gamma_{(W,f),t}(x))}^{(t-1)}, \int_{\mathcal{H}^{t-1}} \mathfrak{h}_-^{(t-1)} dp_t(\gamma_{(W,f),t}(x)) \right) = \mathfrak{h}_{\gamma_{(W,f)},t}^{(t)}(x)$$

Hence, all in all, we have

$$\mathfrak{f}_x^{(t)} = \mathfrak{h}_{\gamma_{(W,f)},t}^{(t)}(x)$$

$\square$

Next, we use Lemma 72 to show that the output of an MPNN with readout on a graphon-signal $(W, f)$ equals the output of the MPNN on the corresponding distribution of computation IDMs $\Gamma_{(W,f)}$ of $(W, f)$.

**Lemma 73.** *Let* $(W, f) \in \mathcal{WL}_r^d$, *let* $(\varphi, \psi)$ *be an L-layer MPNN model with readout, then*

$$\mathfrak{F} := \mathfrak{F}(\varphi, \psi, W, f) = \mathfrak{H}(\varphi, \psi, \Gamma_{(W,f),L}) =: \mathfrak{H}$$

*Proof.* Recall that for any $A \in \mathcal{B}(\mathscr{M}_{t-1})$,

$$\Gamma_{(W,f),L}(A) = \int_{\gamma_{(W,f),L}^{-1}(A)} y.$$

So,

$$\Gamma_{(W,f),L} = (\gamma_{(W,f),L})_* \lambda.$$

Equality follows from the above remark and Lemma 72.

$$\begin{aligned}
\mathfrak{H} &= \psi \left( \int_{\mathcal{H}^L} \mathfrak{h}_-^{(L)} d\Gamma_{(W,f),L} \right) \\
&= \psi \left( \int_{\mathcal{H}^L} \mathfrak{h}_-^{(L)} d(\gamma_{(W,f),L})_* \lambda \right) \\
&= \psi \left( \int_{[0,1]} \mathfrak{h}_{\gamma_{(W,f),L}(x)}^{(L)} d\lambda(x) \right) \\
&= \psi \left( \int_{[0,1]} \mathfrak{f}_x^{(L)} d\lambda(x) \right) \\
&= \mathfrak{F}.
\end{aligned}$$

$\square$

Hence, it suffices to consider MPNNs on DIDMs. We can summerazie the results of this section in the following lemma.

**Lemma 11.** *Let* $(W, f)$ *be a graphon-signal and* $(\varphi, \psi)$ *an L-layer MPNN model with readout. Then, given the computation IDMs* $\{\gamma_{(W,f),t}\}_{t=0}^L$ *and DIDM* $\Gamma_{(W,f),L}$, *we have that* $\mathfrak{f}(\varphi, W, f)_x^{(t)} = \mathfrak{h}(\varphi)_{\gamma_{(W,f),t}(x)}^{(t)}$ *for any* $t \in [L]$, $x \in [0,1]$. *Similarly,* $\mathfrak{F}(\varphi, \psi, W, f) = \mathfrak{H}(\varphi, \psi, \Gamma_{(W,f),L})$.

## H   LIPSCHITZ CONTINUITY OF MPNNS

In this section, we prove that MPNNs are Lipschitz continuous. Note that here $\|f\|_\infty := \operatorname{ess\,sup}_{x \in \mathcal{X}} \|f(x)\|_2$, for a function $f : \mathcal{X} \mapsto \mathbb{R}^d$ (see Definition 44 in Appendix A.12 for a more general definition).

### H.1   UPPER BOUNDS OF HIDDEN REPRESENTATIONS AND OUTPUTS OF MPNNS

We start by showing that the features of MPNNs are bounded. A fact we relay on, in the prof of Theorem 13. Recall that we defined the formal bias of a function $f : \mathbb{R}^{d_1} \mapsto \mathbb{R}^{d_2}$ to be $\|f(0)\|_2$. Here, we use the notation $\mathrm{B}_\varphi^{(t)} := \|\varphi^{(t)}(0)\|_2$ to denote the formal bias of any update function $\varphi^{(t)}$ of an MPNN model $\varphi = (\varphi^{(t)})_{t=0}^L$.

**Theorem 74.** *Let* $\varphi = (\varphi^{(t)})_{t=0}^L$ *be an L-layer MPNN model. Then there exists a constant* $\mathrm{B}_\varphi$ *that depends only on L, the number of layers,* $\|\varphi^{(t)}\|_\mathrm{L}$, *the Lipschitz constants of the update functions and* $B_{\varphi^{(t)}}$, *the formal bias of the update functions, such that*

$$\|\mathfrak{h}(\varphi)_-^{(t)}\|_\infty \leq \mathrm{B}_\varphi.$$

*If* $\varphi$ *has a readout function* $\psi$, *then, there exists a constant* $\mathrm{B}_{(\varphi,\psi)}$ *that depends only on* $\mathrm{B}_\varphi$ *and* $\|\psi\|_\mathrm{L}$, *the Lipschitz constant of the model's readout function, such that*

$$\|\mathfrak{H}(\varphi, \psi, -)\|_\infty \leq \mathrm{B}_{(\varphi,\psi)}.$$

For an $L$-layer MPNN model $\varphi = (\varphi^{(t)})_{t=0}^{L}$, we define a constant, which (we later show) bounds $\mathfrak{h}(\varphi)^{(L)}_{-}$, by $\mathrm{B}_{\varphi} := \mathrm{B}_{\varphi}^{L}$, when $\mathrm{B}_{\varphi}^{t} \geq 0$ is inductively defined for $t \in \{0, \ldots, L\}$ by

$$\mathrm{B}_{\varphi}^{t} := \begin{cases} r \cdot \left\| \varphi^{(0)} \right\|_{\mathrm{L}} + \mathrm{B}_{\varphi^{(0)}} & : \text{if } t = 0, \\ 2 \left\| \varphi^{(t)} \right\|_{\mathrm{L}} \mathrm{B}_{\varphi}^{(t-1)} + \mathrm{B}_{\varphi^{(t)}} & : \text{if } 0 < t \leq L, \end{cases}$$

when $\| \cdot \|_{\mathrm{L}}$ is defined in Appendix A.12 (Definition 43) and $r > 0$ is the radius of $\mathcal{H}^{L} = \mathbb{B}_{r}^{d}$ (see Section 2). If additionally the MPNN model has a readout function $\psi$, then we define another constant, which (we later show) bounds $\mathfrak{H}(\varphi, \psi, -)^{(L)}$, by

$$\mathrm{B}_{(\varphi, \psi)} := \| \psi \|_{\mathrm{L}} \mathrm{B}_{\varphi}^{L} + \mathrm{B}_{\psi}.$$

*Proof.* Let us now prove the first inequality of Theorem 13 by induction. Let $L \in \mathbb{N}_{0}$ and $\varphi = (\varphi)_{t \in [L]}$ $L$-layer MPNN model.

*Induction Base.* For $t = 0$, let $\tau \in \mathcal{H}^{0} = \mathbb{B}_{r}^{d}$, the statement holds since

$$\left\| \mathfrak{h}_{\tau}^{(0)} \right\|_{2} = \left\| \varphi^{(0)}(\tau) \right\|_{2} = \left\| \varphi^{(0)}(\tau) - \varphi^{(0)}(0) \right\|_{2} + \left\| \varphi^{(0)}(0) \right\|_{2}$$
$$\leq \left\| \varphi^{(0)} \right\|_{\mathrm{L}} \| \tau \|_{2} + \left\| \varphi^{(0)} \right\|_{2} \leq r \cdot \left\| \varphi^{(0)} \right\|_{\mathrm{L}} + \mathrm{B}_{\varphi^{(0)}}.$$

*Induction Assumption.* We assume

$$\left\| \mathfrak{h}_{-}^{(t-1)} \right\|_{\infty} \leq \mathrm{B}_{\varphi}^{(t-1)},$$

for some $0 < t \leq L$.

*Induction Step.* Let $\tau \in \mathcal{H}^{t}$, we have

$$\left\| \mathfrak{h}_{\tau}^{(t)} \right\|_{2} = \left\| \varphi^{(t)} \left( \mathfrak{h}_{p_{t,t-1}(\tau)}^{(t-1)}, \int_{\mathcal{H}^{t-1}} \mathfrak{h}_{-}^{(t-1)} dp_{t}(\tau) \right) \right\|_{2}$$
$$\leq \left\| \varphi^{(t)} \left( \mathfrak{h}_{p_{t,t-1}(\tau)}^{(t-1)}, \int_{\mathcal{H}^{t-1}} \mathfrak{h}_{-}^{(t-1)} dp_{t}(\tau) \right) - \varphi^{(t)}(0, 0) \right\|_{2} + \left\| \varphi^{(t)}(0, 0) \right\|_{2}$$
$$\leq \left\| \varphi^{(t)} \right\|_{\mathrm{L}} \left( \left\| \mathfrak{h}_{p_{t,t-1}(\tau)}^{(t-1)} \right\|_{2} + \left\| \int_{\mathcal{H}^{t-1}} \mathfrak{h}_{-}^{(t-1)} dp_{t}(\tau) \right\|_{2} \right) + \mathrm{B}_{\varphi^{(t)}}$$
$$\leq \left\| \varphi^{(t)} \right\|_{\mathrm{L}} \left( \underset{\nu \in \mathcal{H}^{t-1}}{\mathrm{ess \, sup}} \left\| \mathfrak{h}_{\nu}^{(t-1)} \right\|_{2} + \int_{\mathcal{H}^{t-1}} \underset{\nu \in \mathcal{H}^{t-1}}{\mathrm{ess \, sup}} \left\| \mathfrak{h}_{\nu}^{(t-1)} \right\|_{2} dp_{t}(\tau) \right) + \mathrm{B}_{\varphi^{(t)}}$$
$$\leq 2 \left\| \varphi^{(t)} \right\|_{\mathrm{L}} \left\| \mathfrak{h}_{-}^{(t-1)} \right\|_{\infty} + \mathrm{B} \leq 2 \left\| \varphi^{(t)} \right\|_{\mathrm{L}} \mathrm{B}_{\varphi}^{(t-1)} + \mathrm{B}_{\varphi^{(t)}} = \mathrm{B}_{\varphi}^{(t)}.$$

As this is true for any $\tau$, we have

$$\| \mathfrak{h}_{\alpha}^{(t)} \|_{\infty} \leq \mathrm{B}_{\varphi}^{(t)}.$$

Notice that, for $t = L$, we get

$$\| \mathfrak{h}_{\alpha}^{(L)} \|_{\infty} \leq \mathrm{B}_{\varphi}.$$

The second part then follows from the first by a similar reasoning.

$$\| h(\varphi, \psi, \mu) \|_{2} = \left\| \psi \left( \int_{\mathcal{H}^{L}} \mathfrak{h}_{-}^{(L)} d\mu \right) \right\|_{2} \leq \left\| \psi \left( \int_{\mathcal{H}^{L}} \mathfrak{h}_{-}^{(L)} d\mu \right) - \psi(0) \right\|_{2} + \| \psi(0) \|_{2}$$
$$\leq \| \psi \|_{\mathrm{L}} \left\| \int_{\mathcal{H}^{L}} \mathfrak{h}_{-}^{(L)} d\mu \right\|_{2} + \mathrm{B}_{\psi} \leq \| \psi \|_{\mathrm{L}} \left\| \mathfrak{h}_{-}^{(L)} \right\|_{\infty} + \mathrm{B}_{\psi} \leq \| \psi \|_{\mathrm{L}} \mathrm{B}_{\varphi}^{L} + \mathrm{B}_{\psi}$$

$\square$

## H.2 Lipschitz Continuity of MPNNs with respect to Our Metrics

To prove that MPNNs are Lipschitz continuous, we follow the proofs of Grebík & Rocha, 2021, Appendix C.2.5.. The next claim is a trivial result of Claim 23. in Böker et al. (2023).

**Lemma 75.** *Let $f : \mathcal{S} \to \mathbb{R}^n$ be Lipschitz. Then,*

$$\left\| \int_S f d\mu - \int_S f d\nu \right\|_2 \leq \|f\|_{\mathbf{BL}} \cdot \left( (\|\mu\| - \|\nu\|) + \int_{S \times S} d(x, y) d\gamma(x, y) \right)$$

*for every $\gamma \in \Gamma(\mu, \nu)$, where $\gamma$ is a coupling as defined in Section 2 and $\| \cdot \|_{\mathbf{BL}}$ is the Bounded-Lipschitz seminorm defined in Appendix A.12 (Definition 45) over $\mathrm{Lip}(\mathcal{S}, \mathbb{R}^n)$, when $\mathrm{Lip}(\mathcal{S}, \mathbb{R}^n)$ is the space of Lipschitz continuous mappings $\mathcal{S} \mapsto \mathbb{R}^n$.*

**Theorem 13.** *Let $\varphi$ be an L-layer MPNN model. Then there exists a constant $C_\varphi$ that depends only on the number of layers L and the Lipschitz constants of the update functions, such that*

$$\|\mathfrak{h}(\varphi, \alpha)^{(L)} - \mathfrak{h}(\varphi, \beta)^{(L)}\|_2 \leq C_\varphi \cdot d_{\mathrm{IDM}}^L(\alpha, \beta)$$

*for all $\alpha, \beta \in \mathcal{H}^L$. If $\varphi$ has a readout function $\psi$, then, for all $\mu, \nu \in \mathscr{P}(\mathcal{H}^L)$, there exists a constant $C_{(\varphi, \psi)}$ that depends only on $C_\varphi$ and the Lipschitz constant of the model's readout function, such that*

$$\|\mathfrak{H}(\varphi, \psi, \mu) - \mathfrak{H}(\varphi, \psi, \nu)\|_2 \leq C_{(\varphi, \psi)} \cdot \mathbf{OT}_{d_{\mathrm{IDM}}^L}(\mu, \nu).$$

For an $L$-layer MPNN model $\varphi = (\varphi^{(t)})_{t=0}^L$, we define a constant, which we later show is a Lipschitz constant of $\mathfrak{h}(\varphi)_-^{(L)}$, by $C_\varphi := C_\varphi^L$, when $C_\varphi^t \geq 0$ is inductively defined for $t \in \{0, \ldots, L\}$ by

$$C_\varphi^t := \begin{cases} \|\varphi^{(0)}\|_{\mathrm{L}} & : \text{if } t = 0, \\ 2\left\|\varphi^{(t)}\right\|_{\mathrm{L}} (\|\mathfrak{h}_-^{(t-1)}\|_\infty + C_\varphi^{t-1}) & : \text{if } 0 < t \leq L, \end{cases}$$

when $\| \cdot \|_{\mathrm{L}}$ is defined in Appendix A.12 (Definition 43). If additionally the MPNN model has a readout function $\psi$, then we define another constant, which we later show is a Lipschitz constant of $\mathfrak{H}(\varphi, \psi, -)^{(L)}$, by

$$C_{(\varphi, \psi)} := \|\psi\|_{\mathrm{L}} (\left\|\mathfrak{h}_-^{(L)}\right\|_\infty + C_\varphi).$$

Since the features of MPNNs are bounded (Theorem 74), these constants are well defined. We now use Lemma 75 to prove the MPNN's Lipschitz property. We show that, given an L-layer MPNN model $\varphi$ and a readout function $\psi$, $C_\varphi$ and $C_{(\varphi, \psi)}$ are Lipschitz constants of $\mathfrak{h}(\varphi)_-^{(L)}$ and $\mathfrak{H}(\varphi, \psi, -)$, respectively.

*Proof.* Let us now prove the first inequality of Theorem 13 by induction. Let $L \in \mathbb{N}_0$ and $\varphi = (\varphi)_{t \in [L]}$ $L$-layer MPNN model.

*Induction Base.* For $t = 0$, $\varphi = (\varphi^{(0)})$, the statement holds trivially since $\varphi^{(0)}$ is Lipschitz and $\mathfrak{h}_\alpha^{(0)} = \varphi^{(0)}(\alpha)$.

*Induction Assumption.* We assume that the statement hold for $t - 1$ for $0 < t \leq L$.

*Induction Step.* For the inductive step, we have, by Lemma 75 and the induction hypothesis, for all $\alpha, \beta \in \mathcal{H}^t$.

$$\|\mathfrak{h}_\alpha^{(t)} - \mathfrak{h}_\beta^{(t)}\|_2$$
$$= \left\| \varphi^{(t)} \left( \mathfrak{h}_{p_{t,t-1}(\alpha)}^{(t-1)}, \int_{\mathcal{H}^{t-1}} \mathfrak{h}_-^{(t-1)} dp_t(\alpha) \right) - \varphi^{(t)} \left( \mathfrak{h}_{p_{t,t-1}(\beta)}^{(t-1)}, \int_{\mathcal{H}^{t-1}} \mathfrak{h}_-^{(t-1)} dp_t(\alpha) \right) \right\|_2$$
$$\leq \left\| \varphi^{(t)} \right\|_{\mathrm{L}} \left( \left\| \mathfrak{h}_{p_{t,t-1}(\alpha)}^{(t-1)} - \mathfrak{h}_{p_{t,t-1}(\beta)}^{(t-1)} \right\|_2 + \left\| \int_{\mathcal{H}^{t-1}} \mathfrak{h}_-^{(t-1)} dp_t(\alpha) - \int_{\mathcal{H}^{t-1}} \mathfrak{h}_-^{(t-1)} dp_t(\beta) \right\|_2 \right)$$
$$\leq \left\| \varphi^{(t)} \right\|_{\mathrm{L}} (\|\mathfrak{h}_-^{(t-1)}\|_{\mathrm{L}} d_{\mathrm{IDM}}^{t-1}(p_{t,t-1}(\alpha), p_{t,t-1}(\beta)) + \|\mathfrak{h}_-^{(t-1)}\|_{\mathbf{BL}} \mathbf{OT}_{d_{\mathrm{IDM}}^{t-1}}(p_t(\alpha), p_t(\beta))$$
$$\leq 2 \left\| \varphi^{(t)} \right\|_{\mathrm{L}} \|\mathfrak{h}_-^{(t-1)}\|_{\mathbf{BL}} d_{\mathrm{IDM}}^t(\alpha, \beta) = 2 \left\| \varphi^{(t)} \right\|_{\mathrm{L}} (\|\mathfrak{h}_-^{(t-1)}\|_\infty + \|\mathfrak{h}_-^{(t-1)}\|_{\mathrm{L}}) \cdot d_{\mathrm{IDM}}^t(\alpha, \beta)$$
$$= 2 \left\| \varphi^{(t)} \right\|_{\mathrm{L}} (\|\mathfrak{h}_-^{(t-1)}\|_\infty + C_\varphi^{t-1}) \cdot d_{\mathrm{IDM}}^t(\alpha, \beta) = C_\varphi^t d_{\mathrm{IDM}}^t(\alpha, \beta)$$

The second inequality results from combining induction with Lemma 75. Hence, we get the first part of Theorem 13. Notice that, for $t = L$, we get

$$\|\mathfrak{h}_\alpha^{(L)} - \mathfrak{h}_\beta^{(L)}\|_2 \leq C_\varphi d_{\text{IDM}}^L(\alpha, \beta).$$

The second part then follows from the first by a similar reasoning. For all $\mu, \nu \in \mathscr{P}(\mathcal{H}^L)$ we have

$$\begin{aligned}
\|h(\varphi, \psi, \mu) - h(\varphi, \psi, \nu)\|_2 &= \left\|\psi\left(\int_{\mathcal{H}^L} \mathfrak{h}_-^{(L)} d\mu\right) - \psi\left(\int_{\mathcal{H}^L} \mathfrak{h}_-^{(L)} d\nu\right)\right\|_2 \\
&\leq \|\psi\|_{\text{L}} \left\|\int_{\mathcal{H}^L} \mathfrak{h}_-^{(L)} d\mu - \int_{\mathcal{H}^L} \mathfrak{h}_-^{(L)} d\nu\right\|_2 \\
&\leq \|\psi\|_{\text{L}} \left\|\mathfrak{h}_-^{(L)}\right\|_{\textbf{BL}} \mathbf{OT}_{d_{\text{IDM}}^L}(\mu, \nu) \\
&= \|\psi\|_{\text{L}} \left(\left\|\mathfrak{h}_-^{(L)}\right\|_\infty + \left\|\mathfrak{h}_-^{(L)}\right\|_{\text{L}}\right) \mathbf{OT}_{d_{\text{IDM}}^L}(\mu, \nu) \\
&= \|\psi\|_{\text{L}} \left(\left\|\mathfrak{h}_-^{(L)}\right\|_\infty + C_\varphi^L\right) \mathbf{OT}_{d_{\text{IDM}}^L}(\mu, \nu) \\
&= \|\psi\|_{\text{L}} \left(\left\|\mathfrak{h}_-^{(L)}\right\|_\infty + C_\varphi\right) \mathbf{OT}_{d_{\text{IDM}}^L}(\mu, \nu) \\
&= C_{(\varphi, \psi)} \mathbf{OT}_{d_{\text{IDM}}^L}(\mu, \nu).
\end{aligned}$$

$\square$

The second inequality is a result of Lemma 75 and the Lipschitzness from the first part of Theorem 13. For the sake of completeness, we state Theorem 13 as an epsilon-delta statement.

**Theorem 76.** *Let $d > 0$ be fixed. For every $L \in \mathbb{N}_0$, $C > 0$, and $\varepsilon > 0$, there is a $\delta > 0$ such that, for all order-$t$ DIDMs $\mu$ and $\nu$, if $\mathbf{OT}_{d_{\text{IDM}}^t}(\mu, \nu) \leq \delta$, then $\|\mathfrak{H}(\varphi, \psi, \mu) - \mathfrak{H}(\varphi, \psi, \nu)\|_2 \leq \varepsilon$ for every $L$-layer MPNN model $\varphi$ with readout function $\psi : \mathbb{R}^{d_L} \to \mathbb{R}^d$ with $C_{(\varphi, \psi)} \leq C$.*

*Proof.* Follows immediately from Theorem 13. $\square$

# I  Universality and Fine-Grained Expressivity of Message-Passing Neural Networks

In this section we first prove our universal approximation theorem for MPNNs on IDMs and DIDMs, showing the sets $\mathcal{N}_t^1$ and $\mathcal{NN}_t^1$ are dense in $C(\mathcal{H}^t, \mathbb{R})$ and $C(\mathscr{P}(\mathcal{H}^t), \mathbb{R})$, respectively. We then conclude universal approximation theorems for MPNNs on graph-signals and graphon-signals.

## I.1  Universality of MPNNs over the Spaces of IDMs and DIDMs

The proofs of Lemma 78, Theorem 15, follow the proofs of Lemma 25, Theorem 4, and Theorem 6 in Böker et al. (2023), respectively. This follows by inductively applying Stone–Weierstrass theorem, cf. Appendix A.5, to the set $\mathcal{N}_t^1$. Given that $\mathcal{N}_t^1$ satisfies all requirements of the Stone–Weierstrass theorem, Corollary 12 yields that $\mathcal{N}_{t+1}^1$ separates points, which allows us to show that $\mathcal{N}_{t+1}^1$ again satisfies all requirements of the Stone-Weierstrass theorem. We recall the canonical projections were denoted by $p_{L,j} : \mathcal{H}^L \mapsto \mathcal{H}^j$ and $p_L : \mathcal{H}^L \to \mathcal{M}^L$, when $j \leq L < \infty$. We first introduce *function Cartesian product*.

**Definition 77** (Function Cartesian Product). *Let $f : \mathcal{X}_1 \mapsto \mathcal{Y}$ and $g : \mathcal{X}_2 \mapsto \mathcal{Z}$ be two functions. We define* function Cartesian product *as the function $f \times g : \mathcal{X}_1 \times \mathcal{X}_2 \mapsto \mathcal{Y} \times \mathcal{Z}$ such that $(f \times g)((x_1, x_2)) = (f(x_1), g(x_2))$ for $(x_1, x_2) \in \mathcal{X}_1 \times \mathcal{X}_2$.*

Given a set $\mathcal{A}$, recall that we denote by $1_\mathcal{A} : \mathcal{A} \to \mathbb{R}$ a non-zero constant function. Let $\varphi, \varphi'$ be two $L$-layer MPNN models. Define $\Xi_{\text{mul}}((x, y)^T) := x \cdot y$ and $\Xi_{\text{add}}((x, y)^T) := x + c \cdot y$ for all $x, y \in \mathbb{R}$, where $c \in \mathbb{R}$ is fixed. Then, define

$$\varphi_{\text{mul}} := (\varphi^{(0)} \times \varphi'^{(0)}, \ldots, \varphi^{(t)} \times \varphi'^{(t)}, \Xi_{\text{mul}} \circ (\varphi^{(t+1)} \times \varphi'^{(t+1)}))$$

and define $\varphi_{\text{add}}$ analogously via $\Xi_{\text{add}}$.

**Lemma 78.** *Let $0 \leq t < \infty$. The set $\mathcal{N}_t^1$ are closed under multiplication and linear combinations, contains $1_{\mathcal{H}^t}$ and separates points of $\mathcal{H}^t$.*

*Proof.* We will now prove the lemma inductively.

*Induction Base.* For $t = 0$, the claim trivially holds as $\mathcal{N}_0^1$ contains precisely the functions $f : \mathcal{H}^0 \mapsto \mathbb{R}$ that are Lipschitz continuous which contains $1_{\mathcal{H}^0}$, and closed to multiplication and addition.

*Induction Assumption.* We assume the sets $\mathcal{N}_t^1$ is closed under multiplication and linear combinations, contains $1_{\mathcal{H}^t}$ and separates points of $\mathcal{H}^t$.

*Induction Step.* Let $t + 1$. Clearly $\mathcal{N}_{t+1}^1$ contains the all-one function $1_{\mathcal{H}^{t+1}}$ since we can always choose $\varphi^{(t+1)}$ in an MPNN model to be the all-one function on any of the two inputs. Let $\varphi$, $\varphi'$ be two $(t + 1)$-layer MPNN models. Note that $\varphi_{\mathrm{mul}}$ and $\varphi_{\mathrm{add}}$ are in fact MPNN models since multiplication and addition on a compact, and hence, a bounded subset of $\mathbb{R}^2$ is Lipschitz continuous.

Let $\alpha, \beta \in \mathcal{H}^{t+1}$ with $\alpha \neq \beta$. We consider two cases: either $p_{t+1}(\alpha) \neq p_{t+1}(\beta)$ or $p_{t+1,t}(\alpha) \neq p_{t+1,t}(\beta)$. We start with the first case, i.e., $p_{t+1}(\alpha) \neq p_{t+1}(\beta)$. By the induction hypothesis, the set $\mathcal{N}_t^1$ is closed under multiplication and linear combinations, contains $1_{\mathcal{H}^t}$, and separates points of $\mathcal{H}^t$. Hence, it is a sub-algebra of $C(\mathcal{H}^t, \mathbb{R})$ that separates points and contains the constants. By the Stone–Weierstrass theorem, $\mathcal{N}_t^1$ is dense in $C(\mathcal{H}^t, \mathbb{R})$. Corollary 36 then entails that there is a $t$-layer MPNN model $\varphi$ with output dimension one such that

$$\int_{\mathcal{H}^t} \mathfrak{h}(\varphi)_-^{(t)} dp_{t+1}(\alpha) \neq \int_{\mathcal{H}^t} \mathfrak{h}(\varphi)_-^{(t)} dp_{t+1}(\beta).$$

Define the $(t + 1)$-layer MPNN model $\varphi' := (\varphi'^{(i)})_{i=0}^{t+1}$, where $\varphi'^{(i)} = \varphi^{(i)}$ for $i \in [t]$ and $\varphi'^{(t+1)}(x, y) := y$ for every $(x, y) \in \mathbb{R}^2$. Then, $\varphi' \in \mathcal{N}_t^1$, separates $\alpha$ and $\beta$ since

$$\begin{aligned}
\mathfrak{h}^{(t+1)}(\varphi', \alpha) &= \int_{\mathcal{H}^t} \mathfrak{h}(\varphi')_-^{(t)} dp_{t+1}(\alpha) \\
&= \mathfrak{h}(\varphi)_-^{(t)} dp_{t+1}(\alpha) \neq \int_{\mathcal{H}^t} \mathfrak{h}(\varphi)_-^{(t)} dp_{t+1}(\beta) \\
&= \int_{\mathcal{H}^t} \mathfrak{h}(\varphi')_-^{(t)} dp_{t+1}(\beta) = \mathfrak{h}^{(t+1)}(\varphi', \beta).
\end{aligned}$$

In the second case, where $p_{t+1,t}(\alpha) \neq p_{t+1,t}(\beta)$, we have that, from the induction assumption, there exists a $t$-layer MPNN model $\hat{\varphi}$ such that $\mathfrak{h}(\hat{\varphi})_{p_{t+1,t}(\alpha)}^{(t)} \neq \mathfrak{h}(\hat{\varphi})_{p_{t+1,t}(\beta)}^{(t)}$. Define the $(t + 1)$-layer MPNN model $\hat{\varphi}' := (\hat{\varphi}'^{(i)})_{i=0}^{t+1}$, where $\hat{\varphi}'^{(i)} = \hat{\varphi}^{(i)}$ for $i \in [t]$ and $\hat{\varphi}'^{(t+1)}(x, y) := x$ for every $(x, y) \in \mathbb{R}^2$. Then, $\hat{\varphi}' \in \mathcal{N}_t^1$ separates $\alpha$ and $\beta$ since

$$\begin{aligned}
\mathfrak{h}(\hat{\varphi}')_\alpha^{(t+1)} &= \mathfrak{h}(\hat{\varphi}')_{p_{t+1,t}(\alpha)}^{(t)} \\
&= \mathfrak{h}(\hat{\varphi})_{p_{t+1,t}(\alpha)}^{(t)} \neq \mathfrak{h}(\hat{\varphi})_{p_{t+1,t}(\beta)}^{(t)} \\
&= \mathfrak{h}(\hat{\varphi}')_{p_{t+1,t}(\beta)}^{(t)} = \mathfrak{h}(\hat{\varphi}')_\beta^{(t+1)}.
\end{aligned}$$

$\square$

With Lemma 78, we immediately obtain Theorem 15, which we restate here for better readability.

**Theorem 15** (Universal Approximation). *Let $L \in \mathbb{N}_0$. Then, the set $\mathcal{N}_L^1$ is uniformly dense in $C(\mathcal{H}^L, \mathbb{R})$ and the set $\mathcal{N}\mathcal{N}_L^1$ is uniformly dense in $C(\mathscr{P}(\mathcal{H}^L), \mathbb{R})$.*

*Proof.* By Lemma 78, the Stone–Weierstrass theorem is applicable to $\mathcal{N}_L^1$, and hence, $\mathcal{N}_L^1$ is dense in $C(\mathcal{H}^L, \mathbb{R})$. We can then use this to show that $\mathcal{N}\mathcal{N}_L^1$ is dense in $C(\mathscr{P}(\mathcal{H}^L), \mathbb{R})$. By the same arguments as in the first case of the inductive step in the proof of Lemma 78, $\mathcal{N}\mathcal{N}_L^1$ is closed under multiplication and linear combinations, contains the all-one function, and separates points of $\mathscr{P}(\mathcal{H}^L)$. Hence, an application Corollary 36 yields that $\mathcal{N}\mathcal{N}_L^1$ is dense in $C(\mathscr{P}(\mathcal{H}^L), \mathbb{R})$. $\square$

Theorem 15 then yields Corollary 12. To prove Corollary 12, we follow the proof of Corollary 5. in Böker et al. (2023).

**Corollary 12.** *Let $L \in \mathbb{N}_0$ and $d > 0$ be fixed. Let $\nu \in \mathscr{P}(\mathcal{H}^L)$ and $(\nu_i)_i$ be a sequence with $\nu_i \in \mathscr{P}(\mathcal{H}^L)$. Then, $\nu_i \to \nu$ if and only if $\mathfrak{H}(\varphi, \psi, \nu_i) \to \mathfrak{H}(\varphi, \psi, \nu)$ for all $L$-layer MPNN models $\varphi$ with a readout function $\psi : \mathbb{R}^{d_L} \to \mathbb{R}^d$.*

*Proof.* First, let $n = 1$. When restricted to functions $\mathfrak{H}(\varphi, \psi, -) \in \mathcal{NN}_L^n$ of the form $\mathfrak{H}(\varphi, \psi, \nu) = \int_{\mathbb{M}_L} \mathfrak{h}_-^{(L)} d\nu$, i.e., the readout $\psi$ is the identity, the claim follows since $\mathcal{N}_L^1$ is dense in $C(\mathbb{M}_L, \mathbb{R})$ by Theorem 15 and the definition of the weak* topology on $\mathscr{P}(\mathbb{M}_L)$, cf. Section 2. Since the readout function $\psi$ is continuous, the equivalence also holds when considering all functions in the set $\mathcal{NN}_L^d$. Finally, since one can always consider the projection to a single component and conversely map a single real number to a vector of these numbers, the equivalence also holds in the case $n > 1$. $\qquad \square$

## I.2 Proof of Fine-Grained Expressivity of MPNNs

Here, we present the proof of Theorem 14, which we copy here for the convenience of the reader.

**Theorem 14.** *Let $d > 0$ be fixed. For every $\varepsilon > 0$, there are $L \in \mathbb{N}_0$, $C > 0$, and $\delta > 0$ such that, for all DIDMs $\mu, \nu \in \mathscr{P}(\mathcal{H}^L)$, if $\|\mathfrak{H}(\varphi, \psi, \mu) - \mathfrak{H}(\varphi, \psi, \nu)\|_2 \le \delta$ holds for every $L$-layer MPNN model $\varphi$ with readout function $\psi : \mathbb{R}^{d_L} \to \mathbb{R}^d$ when $C_{(\varphi, \psi)} \le C$, then $\mathbf{OT}_{d_{\mathrm{IDM}}^L}(\mu, \nu) \le \varepsilon$.*

*Proof.* Assume that there is an $\varepsilon > 0$ such that such $L \in \mathbb{N}_0$, $C > 0$, and $\delta > 0$ do not exist. Then, for every $L \in \mathbf{N}$ and $C > 0$ $\delta_k := 1/k \ge 0$, there are $L$-layer DIDMs $\mu_k$ and $\nu_k$ such that $\|\mathfrak{H}(\varphi, \psi, \mu_k) - \mathfrak{H}(\varphi, \psi, \nu_k)\|_2 \le \delta_k$ for every $L$-layer MPNN model $(\varphi, \psi)$ with readout and output dimension $d$, and $C_{(\varphi, \psi)} \le C$ but also $\mathbf{OT}_{d_{\mathrm{IDM}}^L}(\mu_k, \nu_k) > \varepsilon$. By the compactness of $\mathscr{P}(\mathcal{H}^L)$, there are subsequences $(\mu_{k_i})_i$ and $(\nu_{k_i})_i$ converging to DIDMs $\widetilde{\mu}$ and $\widetilde{\nu}$, respectively, in the weak* topology. Let $\hat{\varphi}$ be an $L$-layer MPNN model and a readout fucntion $\hat{\psi} : \mathbb{R}^{d_L} \to \mathbb{R}^d$. Then, by Corollary 12, also $(\mathfrak{H}(\hat{\varphi}, \hat{\psi}, \mu_{k_i}))_i$ and $(\mathfrak{H}(\hat{\varphi}, \hat{\psi}, \nu_{k_i}))_i$ converge to $\mathfrak{H}(\hat{\varphi}, \hat{\psi}, \widetilde{\mu})$ and $\mathfrak{H}(\hat{\varphi}, \hat{\psi}, \widetilde{\nu})$, respectively. Hence,

$$
\begin{aligned}
\|\mathfrak{H}(\hat{\varphi}, \hat{\psi}, \widetilde{\nu}) - \mathfrak{H}(\hat{\varphi}, \hat{\psi}, \widetilde{\mu})\|_2 \le & \|\mathfrak{H}(\hat{\varphi}, \hat{\psi}, \widetilde{\nu}) - \mathfrak{H}(\hat{\varphi}, \hat{\psi}, \nu_{k_i})\|_2 \\
& + \|\mathfrak{H}(\hat{\varphi}, \hat{\psi}, \nu_{k_i}) - \mathfrak{H}(\hat{\varphi}, \hat{\psi}, \mu_{k_i})\|_2 \\
& + \|\mathfrak{H}(\hat{\varphi}, \hat{\psi}, \mu_{k_i}) - \mathfrak{H}(\hat{\varphi}, \hat{\psi}, \widetilde{\mu})\|_2 \xrightarrow{i \to \infty} 0
\end{aligned}
$$

by the assumption, i.e., $\mathfrak{H}(\hat{\varphi}, \hat{\psi}, \widetilde{\mu}) = \mathfrak{H}(\hat{\varphi}, \hat{\psi}, \widetilde{\nu})$. Since this holds for every MPNN model and Lipschitz $\psi$, we have $\mathbf{OT}_{d_{\mathrm{IDM}}^L}(\widetilde{\mu}, \widetilde{\nu}) = 0$ by Corollary 12 and Theorem 4 with the fact that $\mathscr{P}(\mathcal{H}^L)$ is Hausdorff. Then, however

$$
\mathbf{OT}_{d_{\mathrm{IDM}}^L}(\mu_{k_i}, \nu_{k_i}) \le \mathbf{OT}_{d_{\mathrm{IDM}}^L}(\mu_{k_i}, \widetilde{\mu}) + \mathbf{OT}_{d_{\mathrm{IDM}}^L}(\widetilde{\mu}, \widetilde{\nu}) + \mathbf{OT}_{d_{\mathrm{IDM}}^L}(\widetilde{\nu}, \nu_{k_i}) \xrightarrow{k \to \infty} 0
$$

since $(\nu_{k_i})_i$ and $(\mu_{k_i})_i$ converge to $\widetilde{\nu}$ and $\widetilde{\mu}$, respectively, also in $\mathbf{OT}_{d_{\mathrm{IDM}}^L}$ by by Corollary 12 and Theorem 4. This contradicts the assumption that $\mathbf{OT}_{d_{\mathrm{IDM}}^L}(\mu_{k_i}, \nu_{k_i}) > \varepsilon$ for every $k \ge 0$. $\qquad \square$

## I.3 Universality of MPNNs over the Spaces of Graph-Signals and Graphon-Signals

Note that Theorem 15 states that any continuous function from DIDMs to scalars can be approximated by an MPNN on DIDMs. To infer a universal approximation result for functions from graph-signals to vector we emphasize the following considerations. Recall we define the set $\mathcal{NN}_L^d(\mathcal{WL}_r^d) = \{\mathfrak{F}(\varphi, \psi, -, -) : \mathcal{WL}_r^d \mapsto \mathbb{R}^d | (\varphi, \psi) \text{ is an } L\text{-layer MPNN model with readout}\}$. First, note that the space of computation DIDMs, $\Gamma_{(W, f), L}(\mathcal{WL}_r^d)$, is a strict subset of $\mathscr{P}(\mathcal{H}^L)$, which is not dense (w.r.t $\delta_{\mathrm{DIDM}}^L$) in view of Theorem 64. Indeed, there are DIDMs that do not come from any graphon-signal, and a closed strict subset cannot be dense. Hence, the space of DIDMs of graph-signals is also not dense. Hence, Theorem 15 does not imply that any continuous function (w.r.t $\delta_{\mathrm{DIDM}}^L$) from graph-signal to vector can be approximated by $\mathcal{NN}_L^d$. Rather, any function on graph-signals that can be extended to a continuous function on DIDMs can be approximated by $\mathcal{NN}_L^d$, which is a weaker form of universality. Fortunately, we can directly prove a universal approximation theorem directly for the space $\mathcal{WL}_r^d$, which in terms gives a universal approximation

theorem for continuous functions from graph-signals to vectors by a density argument. For this, we prove Theorem 79 stating that graph-signals are dense in $\mathcal{WL}_r^d$ w.r.t. $\delta_{\mathrm{DIDM}}^L$.

**Theorem 16.** *Let $L \in \mathbb{N}_0$. Then, the set $\mathcal{NN}_L^1(\mathcal{WL}_r^d)$ is uniformly dense in $C(\mathcal{WL}_r^d, \mathbb{R})$.*

*Proof.* By Theorem 64, the space of graphon-signals is compact. With similar arguments presented in the proof of Lemma 78, $\mathcal{NN}_L^1(\mathcal{WL}_r^d)$ is a subalgebra of $C(\mathcal{WL}_r^d, \mathbb{R})$. Lemma 11 together with the fact that $\mathcal{NN}_L^1$ separate points (see the proof of Theorem 15) yields that $\mathcal{NN}_L^1(\mathcal{WL}_r^d)$ separate points. We can thus apply the Stone–Weierstrass theorem to $\mathcal{NN}_L^1(\mathcal{WL}_r^d)$, hence, $\mathcal{NN}_L^1(\mathcal{WL}_r^d)$ is uniformly dense in $C(\mathcal{WL}_r^d, \mathbb{R})$. $\qquad\square$

**Theorem 79.** *Let $L \in \mathbb{N}_0$. Graph-signals are dense in $\mathcal{WL}_r^d$ w.r.t. $\delta_{\mathrm{DIDM}}^L$.*

*Proof.* A direct conclusion of the graphon-signals regularity lemma (Theorem 42) w.r.t. the cut distance, $\delta_\square$, is that graph-signals are dense in the space of graphon-signals w.r.t. $\delta_\square$, i.e. $(\mathcal{WL}_r^d, \delta_\square)$. For any $L \in \mathbb{N}_0$, Theorem 63 implies that the pseudometric topology of $(\mathcal{WL}_r^d, \delta_{\mathrm{DIDM}}^L)$ is cooraser than pseudometric topology of $(\mathcal{WL}_r^d, \delta_\square)$. Thus, since graph-signals are dense in $(\mathcal{WL}_r^d, \delta_\square)$, graph-signals are dense in raph-signals are dense in the space of graphon-signals w.r.t. $\delta_{\mathrm{DIDM}}^L$, i.e., $(\mathcal{WL}_r^d, \delta_{\mathrm{DIDM}}^L)$. $\qquad\square$

## J    PROXIMITY RELATIONS OF MPNNS

Here, we summarize how proximity of any MPNN's outputs on any two different DIDMs is related to the proximity of the two DIDMs.

**Theorem 80.** *Let $L \in \mathbb{N}_0$ and $(\mu_i)_i$ be a sequence of order-L DIDMs, and let $\mu \in \mathscr{P}(\mathcal{H}^L)$ be a DIDM. Then, the following are equivalent:*

1. $\mathbf{OT}_{d_{\mathrm{IDM}}^L}(\mu_i, \mu) \to 0$.

2. $\mathbf{h}_{\mu_i} \to \mathbf{h}_\mu$ *for any MPNN model $\varphi$ with readout $\psi: \mathbb{R}^{d_L} \mapsto \mathbb{R}^d$, where $d > 0$.*

3. $\mu_i \to \mu$.

*Proof.* The implication $(1) \Rightarrow (2)$ is just a result of Theorem 13, and its converse is Theorem 14. Properties (1) and (2) are equivalent to (3) by Theorem 4 and Corollary 12. $\qquad\square$

We further note that the following variant of Theorem 80 holds as well.

**Theorem 81.** *Let $\mu, \nu \in \mathscr{P}(\mathcal{H}^L)$. Then, the following are equivalent:*

1. $\mathbf{OT}_{d_{\mathrm{IDM}}^L}(\mu, \nu) = 0$.

2. $\mathfrak{h}_\mu = \mathfrak{h}_\nu$ *for every MPNN model $\varphi$ and a readout function $\psi: \mathbb{R}^d \to \mathbb{R}^n$, where $n > 0$.*

3. $\mu = \nu$.

*Proof.* The equivalences follow as in Theorem 80 since $\mathscr{P}(\mathcal{H}^L)$ is Hausdorff. $\qquad\square$

## K    GENERALIZATION THEOREM FOR MPNNS

We expand the generalization analysis in Levie (2023) to a more general setting as done in Rauchwerger & Levie (2025) and adjust it to meet our definitions.

## K.1    Statistical Learning and Generalization Analysis

In statistical learning theory, usually we consider a product probability space $\mathcal{P} = \mathcal{X} \times \mathcal{Y}$, which represents all possible data. We call any arbitrary probability measure on $(\mathcal{P}, \mathcal{B}(\mathcal{P}))$ a *data distribution*. We presume we have a fixed and unknown data distribution $\tau$. As the completeness of our measure space does not affect our construction, we may assume that we complete $\mathcal{B}(\mathcal{P})$ with respect to $\tau$ to a complete $\sigma$-algebra $\Sigma$ or just denote $\Sigma = \mathcal{B}(\mathcal{P})$. Additionally, let $\mathbf{X} \subseteq \mathcal{P}$ be a dataset of independent random samples from $(\mathcal{P}, \tau)$. Additionally, we presume $\mathcal{Y}$ contains values that relate to every point in $\mathcal{X}$, according to a fixed and unknown conditional distribution function $\tau_{\mathcal{Y}|\mathcal{X}} \in \mathscr{P}(\mathcal{Y})$. In its essence, the problem of learning is choosing from some set of function, the one that best approximate the relation between the points in $\mathcal{Y}$ and the points in $\mathcal{X}$.

Let $\mathcal{E}$ be a Lipschitz loss function with a Lipschitz constant denote by $\mathrm{C}_{\mathcal{E}}$. Note that the loss $\mathcal{E}$ can have a learnable component that depends on the dataset $\mathbf{X}$ as long as it is Lipschitz with a constant $\mathrm{C}_{\mathcal{E}}$. Our objective is to find the optimal model $\mathfrak{M}$ from some *hypothesis space* $\mathcal{Z}$ that has a low *statistical risk*

$$\mathcal{R}(\mathfrak{M}) = \mathbb{E}_{(\nu,y)\sim\tau}[\mathcal{E}(\mathfrak{M}(\nu), y)] = \int \mathcal{E}(\mathfrak{M}(\nu), y)d\tau(\nu, y), \quad \mathfrak{M} \in \mathcal{Z}$$

However, as stated before, the true distribution $\tau$ is not directly observable. Instead, we have access to a set of independent, identically distributed (i.i.d) samples $\mathbf{X} = (X_1, \dots, X_N)$ from the data distribution $(\mathcal{P}, \tau)$. Instead of minimizing the statistical risk with an unknown data distribution $\tau$, we try to approximation the optimal model by minimizing the *empirical risk*:

$$\hat{\mathcal{R}}_{\mathbf{X}}(\mathfrak{M}_{\mathbf{X}}) = \frac{1}{N} \sum_{i=1}^{N} \mathcal{E}(\mathfrak{M}_{\mathbf{X}}(\nu_i), Y_i),$$

where $0 < i \leq N : X_i = (\nu_i, Y_i)$ and $\mathfrak{M}_{\mathbf{X}}$ is a model with some possible dependence on the sampled data, e.g., through training.

Generalization analysis goal is to show that low empirical risk of a a network entails low statistical risk as well. One approach to bounding the statistical risk involves using the inequality:

$$\mathcal{R}(\mathfrak{M}) \leq \hat{\mathcal{R}}(\mathfrak{M}) + E$$

where $E$ is called the generalization error, defined as:

$$E = \sup_{\Theta \in \mathcal{H}} |\mathcal{R}(\Theta) - \hat{\mathcal{R}}(\Theta)|$$

It is important to note that the trained network $\mathfrak{M} := \mathfrak{M}_{\mathbf{X}}$ depends on the dataset $\mathbf{X}$. This essantially means that the empirical risk is not truly a Monte Carlo approximation of the statistical risk in the learning context, as the network is not constant when varying the dataset. If the model $\mathfrak{M}$ was fixed, Monte Carlo theory would provide us an order $\mathcal{O}(\sqrt{\zeta(p)/N})$ bound for $E$ with probability $1 - p$, where $\zeta(p)$ depends on the specific inequality used (e.g., $\zeta(p) = \log(2/p)$ in Hoeffding's inequality).

Such events are called *good sampling events* and depend on the model $\mathfrak{M}$. This dependence, result in the requirement of intersecting all good sampling events in $\mathcal{Z}$, in order to compute a naive bound to the generalization error.

Uniform convergence bounds are employed to intersect appropriate sampling events, allowing for more efficient bounding of the generalization error. This intersection introduces a term in the generalization bound called the *complexity* or *capacity*. This concept describes the richness of the hypothesis space $\mathcal{Z}$ and underlies approaches such as VC-dimension, Rademacher dimension, fat-shattering dimension, pseudo-dimension, and uniform covering number (see, e.g., Shalev-Shwartz & Ben-David (2014)).

## K.2    Uniform Monte Carlo Estimation for Lipschitz Continuous Functions

Recall that the covering number of a metric space $(\mathcal{X}, d)$ is the smallest number of open balls of radius $\epsilon$ needed to cover $\mathcal{X}$ (see Appendix A.4.6). We call any metric space with a probability Borel

measure (where we either take the completion of the measure space with respect to $\mu$, i.e. we add all subsets of null-sets to the $\sigma$-lgebra, or not) a probability metric spaces. The proof of Theorem 17 relies on Theorem 82, which examines uniform Monte Carlo estimations of Lipschitz continuous functions over probability metric spaces with finite covering. Theorem 82 is an extended version of Maskey et al., 2022, Lemma B.3 taken from Levie (2023).

**Theorem 82** (Levie (2023), Theorem G.3, Uniform Monte Carlo Estimation for Lipschitz Continuous Functions). *Let $\mathcal{P}$ be a probability metric space with probability measure $\mu$ and covering number $\kappa(\epsilon)$. Let $X_1, \ldots, X_N$ be drawn i.i.d. from $\mathcal{P}$. Then, for any $p > 0$, there exists an event $\mathcal{E}_{Lip}^p \subset \mathcal{P}^N$ (regarding the choice of $(X_1, \ldots, X_N)$), with probability*

$$\mu^N(\mathcal{E}_{Lip}^p) \geq 1 - p$$

*such that for every $(X_1, \ldots, X_N) \in \mathcal{E}_{Lip}^p$, for every bounded Lipschitz continuous function $f : \mathcal{P} \to \mathbb{R}^d$ with Lipschitz constant $L_f$, we have*

$$\left\| \int f(x) d\mu(x) - \frac{1}{N} \sum_{i=1}^{N} f(X_i) \right\|_\infty \leq 2\xi^{-1}(N)L_f + \frac{1}{\sqrt{2}}\xi^{-1}(N)\|f\|_\infty \left(1 + \sqrt{\log(2/p)}\right),$$

*where $\xi(r) = \frac{\kappa(r)^2 \log(\kappa(r))}{r^2}$, $\xi^{-1}$ is the inverse function of $\xi$ and $\kappa(\epsilon)$ is the covering number of $\mathcal{P}$.*

### K.3 A GLOBAL LIPSCHITZ CONSTANT AND UPPER BOUND OF FEATURES OF MPNNs

Recall that we consider the space of IDMs of order-0 to be $\mathcal{H}^0 = \mathbb{B}_r^d = \{x \in \mathbb{R}^d : \|x\|_2 \leq r\} \subset \mathbb{R}^d$ (for a fixed $r > 0$). Moreover, recall that we defined the formal bias of a function $f : \mathbb{R}^{d_1} \mapsto \mathbb{R}^{d_2}$ to be $\|f(0)\|_2$ (Maskey et al., 2022; Levie, 2023) and the smallest Lipschitz constants of $f$ as $\|f\|_L$ (see Appendix A.12). To apply Theorem 82 in our setting, we phrase the following theorems, which show that, under some assumptions, features of MPNNs are bounded Lipschitz Continuous Functions with a bound and Lipschitz constant that do not depend on the specific MPNN model. Theorem 74 and Theorem 13 straightforwardly lead to Corollary 83 and Corollary 84, respectively.

**Corollary 83.** *Let $r > 0$ (such that $\mathcal{H}^0 = \mathbb{B}_r^d$). Assume there exist constants $A_1 > 0$ and $A_2 > 0$, such that the Lipschitz constants $\|\varphi^{(t)}\|_L$ of $\varphi^{(t)}$ satisfy $\|\varphi^{(t)}\|_L \leq A_1$ and the formal biases $\|\varphi^{(t)}(0)\|_2$ of $\varphi^{(t)}$ satisfy $\|\varphi^{(t)}(0)\|_2 \leq A_2$, for any $t \in [L]$ and any $L$-layer MPNN model $\varphi = (\varphi^{(t)})_{t=0}^L$. Then there exists a constant $B_1$ that depends only on $L, A_1$ and $A_2$, such that*

$$\|\mathfrak{h}(\varphi)_-^{(t)}\|_\infty \leq B_1.$$

*If, in addition, the Lipschitz constant $\|\psi\|_L$ of $\psi$ satisfies $\|\psi\|_L \leq A_1$ and the formal bias $\|\psi(0)\|_2$ of $\psi$ satisfies $\|\psi(0)\|_2 \leq A_2$, for any readout function $\psi$, then, there exists a constant $B_2$ that depends only on $L, A_1$ and $A_2$, such that*

$$\|\mathfrak{H}(\varphi, \psi, -)\|_\infty \leq B_2.$$

Let $A_1, A_2$. Define by $B_1 := B^L$ and $B_2 := A_1 B_1 + A_2$, when $B^t \geq 0$ is inductively defined for $t \in \{0, \ldots, L\}$ by

$$B^t := \begin{cases} rA_1 + A_2 & : \text{if } t = 0, \\ 2A_1 B^{(t-1)} + A_2 & : \text{if } 0 < t \leq L, \end{cases} \tag{9}$$

Based on Theorem 74, it easily follows that $B_1$ and $B_2$ provide upper bounds for $\mathfrak{h}(\varphi)_-^{(L)}$ and $\mathfrak{H}(\varphi, \psi, -)$ for all $(\varphi, \psi)$ that satisfy the above assumptions, respectively.

**Corollary 84.** *Assume there exist constants $A_1 > 0$ and $A_2 > 0$, such that the Lipschitz constants $\|\varphi^{(t)}\|_L$ of $\varphi^{(t)}$ satisfy $\|\varphi^{(t)}\|_L \leq A_1$ and the formal biases $\|\varphi^{(t)}(0)\|_2$ of $\varphi^{(t)}$ satisfy $\|\varphi^{(t)}(0)\|_2 \leq A_2$, for any $t \in [L]$ and any $L$-layer MPNN model $\varphi = (\varphi^{(t)})_{t=0}^L$. Then there exists a constant $C_1$ that depends only on $L$ and $A_1$ and $A_2$, such that*

$$\|\mathfrak{h}(\varphi, \alpha)^{(L)} - \mathfrak{h}(\varphi, \beta)^{(L)}\|_2 \leq C_1 \cdot d_{\text{IDM}}^L(\alpha, \beta),$$

*for all $\alpha, \beta \in \mathcal{H}^L$. If, in addition, the Lipschitz constant $\|\psi\|_L$ of $\psi$ satisfies $\|\psi\|_L \leq A_1$ and the formal bias $\|\psi(0)\|_2$ of $\psi$ satisfies $\|\psi(0)\|_2 \leq A_2$, for any readout function $\psi$, then, for all $\mu, \nu \in \mathscr{P}(\mathcal{H}^L)$, there exists a constant $C_2$ that depends only on $L, A$ and $B$, such that*

$$\|\mathfrak{H}(\varphi, \psi, \mu) - \mathfrak{H}(\varphi, \psi, \nu)\|_2 \leq C_2 \cdot \mathbf{OT}_{d_{\text{IDM}}^L}(\mu, \nu).$$

Let $A_1, A_2 > 0$. Define by $C_1 := C^L$ and $C_2 := A_1(B_1 + C_1)$, when $C^t \geq 0$ is inductively defined for $t \in \{0, \dots, L\}$ by

$$C^t := \begin{cases} A_1 & : \text{if } t = 0, \\ 2A_1(B^{t-1} + C^{t-1}) & : \text{if } 0 < t \leq L, \end{cases}$$

$B^t$ is defined for $t \in \{0, \cdot, L\}$ in Equation (9), and $B_1 := B^L$. Based on Theorem 13, it easily follows that $C_1$ and $C_2$ are Lipschitz constants of $\mathfrak{h}(\varphi)_{-}^{(L)}$ and $\mathfrak{H}(\varphi, \psi, -)$ for all $(\varphi, \psi)$ that satisfy the above assumptions, respectively.

### K.4 A GENERALIZATION THEOREM FOR MPNNs

In classification tasks our goal is to classify the input space into K classes. We look at the product probability metric space $\mathcal{P} = \mathcal{X} \times \mathbb{R}^K$, when the metric space $(\mathcal{X}, d)$ is either $(\mathscr{P}(\mathcal{H}^L), \mathbf{OT}_{d_{\mathrm{IDM}}^L})$ or $(\mathcal{WL}_r^d, \delta_{\mathrm{DIDM}}^L)$, and use $L$-layer MPNNs with readout. Our loss $\mathcal{E}$ is a Lipschitz loss function with a Lipschitz constant $C_{\mathcal{E}}$ and our output vectors are vectors $\vec{v} \in \mathbb{R}^K$. Each entry $(\vec{v})_k$ of an output vector $\vec{v}$, depicts the probability that the input belongs to class $0 < k \leq K$. Although loss functions like cross-entropy are not Lipschitz continuous, composing cross-entropy on softmax is Lipschitz continuous, which is usually how cross-entropy is being used. Recall that we defined the formal bias of a function $f : \mathbb{R}^{d_1} \mapsto \mathbb{R}^{d_2}$ to be $\|f(0)\|_2$ (Maskey et al., 2022; Levie, 2023) and the smallest Lipschitz constants of $f$ as $\|f\|_{\mathrm{L}}$ (see Appendix A.12).

Fix $L \in \mathbb{N}$ and $A_2, A_1 > 0$. Let $\Theta$ be the set of all $L$-layer MPNN models with readout $\left((\varphi^{(t)})_{t \in [L]}, \psi\right)$ such that the Lipschitz constants $\|\varphi^{(t)}\|_{\mathrm{L}}$ and $\|\psi\|_{\mathrm{L}}$ are bounded by $A_1$ and the formal biases $\|\varphi^{(t)}(0)\|_2$ and $\|\psi(0)\|_2$ are bounded by $A_2$. Consider a Lipschtiz continuous loss $\mathcal{E}$ with Lipschtiz constant $\tilde{C}_{\mathcal{E}}$. In Appendix K.3, We show that there exist $C_{\Theta}, B_{\Theta} > 0$ that depend on $L, A_1, A_2$ such that $\Theta \subseteq \mathrm{Lip}(\mathcal{X}, C_{\Theta}, B_{\Theta})$. Here, $\mathrm{Lip}(\mathcal{X}, C_{\Theta}, B_{\Theta})$ is the set of all bounded Lipschitz continuous functions $f : \mathcal{X} \mapsto \mathbb{R}^{d_L}$ with bounded Lipschitz constants $\|f\|_{\mathrm{L}} \leq C_{\Theta}$ and with bounded norms $\|f\|_{\infty} \leq B_{\Theta}$. As a result, if we prove a generalization bound using $\mathrm{Lip}(\mathcal{X}, C_{\Theta}, B_{\Theta})$ as the hypothesis class, the bound would also be satisfied for the hypothesis class $\Theta$.

**Lemma 85.** *Let* $\mathfrak{M} \in \mathrm{Lip}(\mathcal{X}, C_{\Theta}, B_{\Theta})$ *and* $\mathcal{E}$ *a loss function with a Lipschitz constant* $C_{\mathcal{E}}$. *Then*

$$\|\mathcal{E}(\mathfrak{M}(\cdot), \cdot)\|_{\infty} \leq C_{\mathcal{E}}(B_{\Theta} + 1) + |\mathcal{E}(0,0)| \quad \text{and} \quad \|\mathcal{E}(\mathfrak{M}(\cdot), \cdot)\|_{\mathrm{L}} \leq C_{\mathcal{E}} \max(C_{\Theta}, 1).$$

*Proof.* For the first inequality,

$$\begin{aligned} |\mathcal{E}(\mathfrak{M}(x), y)| &\leq |\mathcal{E}(\mathfrak{M}(x), y) - \mathcal{E}(0,0)| + |\mathcal{E}(0,0)| \\ &\leq C_{\mathcal{E}}(\|\mathfrak{M}(x)\|_2 + \|y\|_2) + |\mathcal{E}(0,0)| \\ &\leq C_{\mathcal{E}}(\|\mathfrak{M}(x)\|_2 + 1) + |\mathcal{E}(0,0)|. \end{aligned}$$

Thus,

$$\|\mathcal{E}(\mathfrak{M}(\cdot), \cdot)\|_{\infty} \leq C_{\mathcal{E}}(\|\mathfrak{M}(\cdot)\|_{\infty} + 1) + |\mathcal{E}(0,0)| \leq C_{\mathcal{E}}(B_{\Theta} + 1) + |\mathcal{E}(0,0)|.$$

For the second inequality,

$$\begin{aligned} |\mathcal{E}(\mathfrak{M}(x), y) - \mathcal{E}(\mathfrak{M}(x'), y')| &\leq C_{\mathcal{E}}(\|\mathfrak{M}(x) - \mathfrak{M}(x')\|_2 + \|y - y'\|_2) \\ &\leq C_{\mathcal{E}}(C_{\Theta} d(x, x') + \|y - y'\|_2) \\ &\leq C_{\mathcal{E}} \max(C_{\Theta}, 1)(d(x, x') + \|y - y'\|_2). \end{aligned}$$

$\square$

Although the covering numbers of the DIDMs' spaces and the graphon-signals space are currently unknown, the graphon-signal space might have a smaller covering number. This can potentially improve the generalization bound, thus stating the theorem for the graphon-signal space is indeed a meaningful fact. Thus, we express our generalization bounds on the space of DIDMs, which is mathematically more general than a formulation on the space of graphon-signals and on the space of graphon-signals. Following the proof of Theorem G.4 in Levie (2023), we prove the next theorem via Theorem 82.

**Theorem 17** (MPNN generalization theorem). *Consider the above classification setting with $\mathcal{X}$ being either $(\mathscr{P}(\mathcal{H}^L), \mathbf{OT}_{d_{\mathrm{IDM}}^L})$ or $(\mathcal{WL}_r^d, \delta_{\mathrm{DIDM}}^L)$. Let $\mathrm{C} := \mathrm{C}_\mathcal{E} \max(\mathrm{C}_\Theta, 1)$, $\mathrm{B} := \mathrm{C}_\mathcal{E}(\mathrm{B}_\Theta + 1) + |\mathcal{E}(0,0)|$, and $\{X_i\}_{i=1}^N$ be independent random samples from the data distribution $(\mathcal{X} \times \{0,1\}^K, \Sigma, \tau)$. Then, for every $p > 0$, there exists an event $\mathcal{U}^p \subset (\mathcal{X} \times \{0,1\}^K)^N$ regarding the choice of $\mathbf{X} = (X_1, \ldots, X_N)$, with probability $\tau^N(\mathcal{U}^p) \geq 1 - p$, in which for every function $\mathfrak{M}_\mathbf{X}$ in the hypothesis class $Lip(\mathcal{X}, \mathrm{C}_\Theta, \mathrm{B}_\Theta)$, we have*

$$\left| \mathcal{R}(\mathfrak{M}_\mathbf{X}) - \hat{\mathcal{R}}_\mathbf{X}(\mathfrak{M}_\mathbf{X}) \right| \leq \xi^{-1}(N) \left( 2\mathrm{C} + \frac{1}{\sqrt{2}} \mathrm{B} \left( 1 + \sqrt{\log(2/p)} \right) \right), \tag{1}$$

*where $\xi(\epsilon) = \frac{\kappa(\epsilon)^2 \log(\kappa(\epsilon))}{\epsilon^2}$, $\kappa$ is the covering number of the compact space $\mathcal{X} \times \{0,1\}^K$ and $\xi^{-1}$ is the inverse function of $\xi$.*

*Proof.* From Theorem 82 we get the following. For every $p > 0$, there exists an event $\mathcal{E}_i^p \subset \mathcal{X}^N$ regarding the choice of $(X_1, \ldots, X_N) \subseteq (\mathcal{X} \times \{0,1\}^K)^N$, where $X_i = (\nu_i, Y_i)$ for $\nu_i \in \mathcal{X}$, $Y_i \in \{0,1\}^K$, and $0 < i \leq N$, with probability

$$\tau^N(\mathcal{E}_{\mathrm{Lip}}^p) \geq 1 - p,$$

such that for every function $\mathfrak{M}$ in the hypothesis class $\mathrm{Lip}(\mathcal{X}, C_\Theta, B_\Theta)$, we have

$$\left| \int \mathcal{E}(\mathfrak{M}(\nu), y) d\tau(\nu, y) - \frac{1}{N} \sum_{i=1}^N \mathcal{E}(\mathfrak{M}(\nu_i), Y_i) \right| \tag{10}$$

$$\leq \xi^{-1}(N) \left( 2 \|\mathcal{E}(\mathfrak{M}'(\cdot), \cdot)\|_{\mathrm{L}} + \frac{1}{\sqrt{2}} (\|\mathcal{E}(\mathfrak{M}'(\cdot), \cdot)\|_\infty \left( 1 + \sqrt{\log(2/p)} \right) \right) \tag{11}$$

$$\leq \xi^{-1}(N) \left( 2\mathrm{C} + \frac{1}{\sqrt{2}} \mathrm{B} \left( 1 + \sqrt{\log(2/p)} \right) \right), \tag{12}$$

where $\xi(N) = \frac{\kappa(N)^2 \log(\kappa(N))}{N^2}$, $\kappa(\epsilon)$ is the covering number of $\mathcal{X} \times \{0,1\}^K$, and $\xi^{-1}$ is the inverse function of $\xi$. In the last inequality, we used Lemma 85.

Since Equation (10) is true for any $\mathfrak{M} \in \mathrm{Lip}(\mathcal{X}, C_\Theta, B_\Theta)$, it is also true for $\mathfrak{M}_\mathbf{X}$ for any realization of $\mathbf{X}$, so we have

$$\left| \mathcal{R}(\mathfrak{M}_\mathbf{X}) - \hat{\mathcal{R}}_\mathbf{X}(\mathfrak{M}_\mathbf{X}) \right| \leq \xi^{-1}(N) \left( 2\mathrm{C} + \frac{1}{\sqrt{2}} \mathrm{B} \left( 1 + \sqrt{\log(2/p)} \right) \right)$$

$\square$

## L  PROKHOROV'S DISTANCE FOR DIDM METRICS

For completeness, we show an alternative approach to define a metric on graphons through IDMs and DIDMs using Prokhorv metric.

### L.1  DEFINITION AND BASIC PROPERTIES OF PROKHOROV'S DISTANCE

Let $\mathcal{X}$ be a complete separable metric space with Borel $\sigma$-algebra $\mathcal{B}$. We define $A^\epsilon := \{y \in S \mid d(x,y) \leq \epsilon \text{ for some } x \in A\}$ for a subset $A \subseteq \mathcal{X}$ and $\epsilon \geq 0$. Then, the Prokhorov metric $\mathbf{P}$ on $\mathcal{M}_{\leq 1}(\mathcal{X}, \mathcal{B})$ is given by

$$\mathbf{P}_d(\mu, \nu) := \inf\{\epsilon > 0 \mid \mu(A) \leq \nu(A^\epsilon) + \epsilon \text{ and } \nu(A) \leq \mu(A^\epsilon) + \epsilon \text{ for every } A \in \mathcal{B}\}.$$

The following theorem shows that Prokhorov metric is topologically equivalent to $\mathbf{OT}_d$ on any complete separable metric space $(\mathcal{X}, d)$.

**Lemma 86** ( Prokhorov (1956), Theorem 1.11). *Let $(\mathcal{X}, d)$ be a complete separable metric space. Then, $(\mathcal{M}(\mathcal{X}), \mathbf{P}_d)$ is a complete separable metric space, and convergence in $\mathbf{P}_d$ is equivalent to weak* convergence of measures.*

The following definition presents an alternative metric to $\delta_{\mathrm{DIDM}}^L$ (see Section 3).

**Definition 87** (DIDM Prokhorov's Distance). *Given two graphon-signals $(W_a, f_a)$, $(W_b, f_b)$ and $L \geq 1$, the tree prokhorov's distance between $(W_a, f_a)$ and $(W_b, f_b)$ is defined as*

$$\rho_{\mathrm{DIDM}}^L((W_a, f_a), (W_b, f_b)) = \mathbf{P}_{p_{\mathrm{IDM}}^L}(\Gamma_{(W_a, f_a), L}, \Gamma_{(W_b, f_b), L}),$$

By following the proof of Theorem 4, with Lemma 86, the following result is obtained.

**Theorem 88.** *Let $L \in \mathbb{N}_0$. The metrics $p_{\mathrm{IDM}}^L$ on $\mathcal{H}^L$, $\mathbf{P}_{p_{\mathrm{IDM}}^L}$ on $\mathscr{P}(\mathcal{H}^L)$ and $\mathscr{M}_{\leq 1}(\mathcal{H}^L)$ are well-defined. Moreover, $\mathbf{P}_{p_{\mathrm{IDM}}^L}$ metrizes the weak\* topology of $\mathscr{M}_{\leq 1}(\mathcal{H}^L)$ and $\mathscr{P}(\mathcal{H}^L)$.*

Except for the computability (Theorem 6), all the results in this paper, can be rephrased using $p_{\mathrm{IDM}}^L$, $\mathbf{P}_{p_{\mathrm{IDM}}^L}$, and $\rho_{\mathrm{DIDM}}^L$ instead of $d_{\mathrm{IDM}}^L$, $\mathbf{OT}_{d_{\mathrm{IDM}}^L}$, and $\delta_{\mathrm{DIDM}}^L$. The only that remains, is to discuss $\rho_{\mathrm{DIDM}}^L$ compatibility.

### L.2 Computability

The metric $\rho_{\mathrm{DIDM}}^L$ is polynomial-time computable. Böker et al. (2023) prove the Lemma 89 by generalizing an observation in Theorem 1 Schay (1974) and Lemma García-Palomares & Giné (1977) to finite measures to finite measures and show that value of $\rho(\varepsilon) := \inf\{\eta > 0 \mid \mu(A) \leq \nu(A^\varepsilon) + \eta$ for every $A \subseteq S\}$ can be computed through a linear program. Additionally, they based their conclusions on Garel & Massé (2009), which deals with the computation of $\mathbf{P}_d$ on (possibly non-discrete) probability distributions.

**Lemma 89** (Böker et al. (2023), Theorem 16.). *Let $\mu, \nu \in \mathscr{M}(\mathcal{X})$, where $(\mathcal{X}, d)$ is a finite metric space with $\mathcal{X} = \{x_1, \ldots, x_n\}$. Then, the Prokhorov metric $\mathbf{P}_d(\mu, \nu)$ can be computed in time polynomial in $n$ and the number of bits needed to encode $d$, $\mu$ and $\nu$.*

By following Theorem 6, with Lemma 89, the following result is obtained.

**Theorem 90.** *For any fixed $L \in \mathbb{N}_0$, $\delta_{\mathrm{DIDM}}^L$ between any two graph-signals $(G, \mathbf{f})$ and $(H, \mathbf{g})$ can be computed in time polynomial in $h$ and the size of $G$ and $H$, namely $O(L \cdot N^7)$ where $N = \max(|V(G)|, |V(H)|)$.*

The computational advantage of using unbalanced optimal transport, tipped the scales in favor it, making it the main focus of this paper.

## M Additional Experiments and Details

We present here additional experimental results. We evaluate $\delta_{\mathrm{DIDM}}^2$ in graph classification tasks, i.e., graphs separation tasks. We follow the same set up in Chen et al. (2022); Böker et al. (2023) for comparison. The goal of our experiments is to support the theoretical results which formulate a form of equivalence between GNN outputs and DIDM mover's distance. We emphasize that our proposed DIDM mover's distance metric is mainly a tool for theoretical analysis and the proposed experiments are not designed to compete with state-of-the-art methods. Although our metric is not intended to be used directly as a computational tool, our results suggest that we can roughly approximate the DIDM mover's distance between two graphs by the Euclidean distance between their outputs under random MPNNs. This Euclidean distance can be used in practice as it is less computationally expensive than DIDM mover's distance.

### M.1 1-Nearest-Neighbor classifier

The goal of the experiment in this section is to show that the geometry underlying the metric $\delta_{\mathrm{DIDM}}$ captures in some sense the underlying data-driven similarity related to the classification task. We consider the problem of classifying attributed graphs, and a solution based on the 1-nearest neighbor.

The *1-Nearest Neighbor* (1-NN) *classifier* is a non-parametric, instance-based machine learning method. Given a dataset $\mathcal{D} = \{((G_i, \mathbf{f}_i), y_i)\}_{i=1}^n$, where $(G_i, \mathbf{f}_i)$ represents graph-signals and $y_i \in \mathcal{C}$ denotes class labels from a finite set $\mathcal{C}$, the goal is to classify a new input $(G_i, \mathbf{f}_i)$. The classification process of classifying the input $(G, \mathbf{f})$ involves:

1. Computing the distance between the input $(G, \mathbf{f})$ and every point $(G_i, \mathbf{f}_i)$ in the dataset using a distance metric $d$.

2. Identifying the nearest neighbor $(G_k, \mathbf{f}_k)$ such that:

$$(G_k, \mathbf{f}_k) = \arg \min_{i \in n} d((G, \mathbf{f}), (G_i, \mathbf{f}_i)).$$

3. Assigning the label $y_k$ of the nearest neighbor as the label $y$ of $(G, \mathbf{f})$:

$$y \leftarrow y_k.$$

Here we chose to compare $\delta_{\text{DIDM}}^2$ with other optimal-transport-based iteratively defined metrics. Tree Mover's Distance $\mathbf{TMD}^L$ from Chuang & Jegelka (2022) is defined via optimal transport between finite attributed graphs through so called computation trees. The Weisfeiler-Lehman (WL) distance $d_{\text{WL}}^{(L)}$ and its lower bound distance $d_{\text{WLLB}, \geq L}^{(L)}$ from Chen et al. (2022; 2023) are defined via optimal transport between finite attributed graphs through hierarchies of probability measures. The metric $\delta_{W,L}$ from Böker et al. (2023) is define via optimal transport on a variant of IDMs and DIDMs where the IDMs are not concatenated. The metric $\delta_{W, \geq L}$, from Böker et al. (2023), is a variation of $\delta_{W,L}$ where the maximum number of iterations performed after having obtained a stable coloring is bounded by 3. Unlike $\delta_{\text{DIDM}}$, all the above metrics cannot be used to unify expressivity, uniform approximation and generalization for attributed graphs. Namely, the above pseudometrics are either restricted to graphs without attributes or are not compact.

Table 1 compares the mean classification accuracy of $\delta_{\text{W},3}$, $\delta_{\text{W}, \geq 3}$ (Böker et al., 2023), $d_{\text{WL}}^{(3)}$, $d_{\text{WLLB}, \geq 3}^{(3)}$ (Chen et al., 2022; 2023), $\mathbf{TMD}^3$ (Chuang & Jegelka, 2022), and $\delta_{\text{DIDM}}^2$ in a 1-NN classification task using node degrees as initial labels. We used the MUTAG dataset (Morris et al., 2020) and followed the same random data split as in Chen et al. (2022); Böker et al. (2023): 90 percent of the data for training and 10 percent of the data for testing. We repeat the random split ten times. We started by computing the pairwise distances for all the graphs in the dataset. We continued by performing graph classification using a 1-nearest-neighbor classifier (1-NN).

We note that the 1-NN classification experiment "softly" supports our theory, in the sense that this experiment shows that our metric clusters the graphs quite well with respect to their task-driven classes. We stress that this experiment does not directly evaluates any rigorous theoretical claim. We moreover note that while the metric in Chen et al. (2022) achieves better accuracy, the space of all graphs under their metric is not compact, so this metric does not satisfy our theoretical requirements: a compact metric which clusters graphs well.

## M.2 MPNNs' Input and Output Distance Correlation

As a proof of concept, we empirically test the correlation between $\delta_{\text{DIDM}}^L$ and distance in the output of MPNNs. We hence chose well-known and simple MPNN architectures, varying the hidden dimensions and number of layers. We do not claim that GIN Xu et al. (2019) and GraphConv Morris et al. (2019) are representative of the variety of all types of MPNNs. Nevertheless, they are proper choices for demonstrating our theory in practice.

### M.2.1 MPNN Architectures

The `GIN_meanpool` model is a variant of the Graph Isomorphism Network (GIN) (see Appendix A.2.1) designed for graph-level representation learning. Each layer consists of normalized sum aggregation and a multi layer perceptron (MLP). The first MLP consists of two linear transformations, ReLU activations, and batch normalization. Each MLP that follows has additionally a skip connection and summation of the input features and output features. The readout after $L$ layers is mean pooling (with no readout function).

The `GC_meanpool` model is a realization of graph convolution network (GCN) for graph-level representation learning. Each layer consists of normalized sum aggregation with a linear message and update functions (see Appendix D for the definition of message function and for the equivalency between MPNNs that use message functions and MPNNs with no message functions). All layers except the first layer have additionally a skip connection and a summation of the input features and output features. The readout after $L$ layers is mean pooling (with no readout function).

### M.2.2 CORRELATION EXPERIMENTS ON GRAPHS GENERATED FROM A STOCHASTIC BLOCK MODEL

We extend here the experiments presented in Section 4, with the same experimental procedure and also offer an extended discussion and description of the experiments. We empirically test the correlation between $\delta_{\mathrm{DIDM}}^L$ and the distance in the output distance of an MPNN. We use stochastic block models (SBMs), which are generative models for graphs, to generate random graph sequences. We generated a sequence of 50 random graphs $\{G_i\}_{i=0}^{49}$, each with 30 vertices. Each graph is generated from an SBM with two blocks (communities) of size 15 with $p = 0.5$ and $q_i = 0.1 + 0.4i/49$ probabilities of having an edge between each pair of nodes from the same block different blocks, respectively. We denote $G := G_{49}$, which is an Erdős–Rényi model. We plot $\delta_{\mathrm{DIDM}}^2(G_i, G)$ against distance in the output of randomly initialized MPNNs, i.e., once against $\|\mathfrak{H}(\texttt{GIN\_meanpool}, G_i) - \mathfrak{H}(\texttt{GC\_meanpool}, G)\|_2$ and once against $\|\mathfrak{H}(\texttt{GIN\_meanpool}, G_i) - \mathfrak{H}(\texttt{GC\_meanpool}, G)\|_2$. Note that in each experiment, we initialize `GIN_meanpool` and `GC_meanpool` only once with random weights and then compute the hidden representations of all graphs.

We conducted the entire procedure twice, once with a constant feature attached to all nodes and once with a signal which has a different constant value on each community of the graph. Each value is randomly sampled from a uniform distribution over $[0, 1]$. In Section 4 We present the results of the experiments when varying hidden dimension (see Figure 2). Figure 6 and Figure 7 show the results when varying the number of layers when the signal is constant and when the signal has a different randomly generated constant value on each community, respectively. The results still show a strong correlation between input distance and GNN outputs. When increasing the number of layer, the correlation slightly weakens.

We conducted the experiment one more time with signal values sampled from a normal distribution $\mathcal{N}(\mu, \sigma_i)$ with mean $\mu = 1$ and variance $\sigma_i = \frac{49-i}{49}$. Figure 8 and Figure 9 show the results when varying the number of dimensions and the number of layers, respectively. The results still show a correlation between input distance and MPNN outputs, but with a higher variance. As we interpret this result, the increased variance could be an artifact of using random noise as signal in our experiments. The specific MPNNs we used, have either a linear activation or a ReLU activation function. Thus, they have a "linear" averaging effect on the signal, which cancels in a sense the contribution of the noise signal to the output of the MPNN, while the metric takes the signal into full consideration. This leads to a noisy correlation.

### M.2.3 REAL DATASETS CORRELATION EXPERIMENTS

We empirically test the correlation between $\delta_{\mathrm{DIDM}}^L$ and distance in the output of MPNNs on MUTAG and PROTEINS databases. In the following, we present the results that showcase insightful relations.

**Correlation experiments using a single randomized MPNN.** Denote by $\mathcal{D}$ a generic dataset. For the entire dataset we randomly initialized one MPNN with random weights. We randomly picked an attributed graph from the dataset $(\hat{G}, \hat{\mathbf{f}}) \in \mathcal{D}$. For each $(G, \mathbf{f}) \in \mathcal{D}$ we computed $\delta_{\mathrm{DIDM}}^2((\hat{G}, \hat{\mathbf{f}}), (G, \mathbf{f}))$. We plotted the distance in the output of the randomly initialized MPNNs on each $(G, \mathbf{f}) \in \mathcal{D}$ against $\delta_{\mathrm{DIDM}}^2((\hat{G}, \hat{\mathbf{f}}), (G, \mathbf{f}))$. We conducted the experiment multiple times. Figure 4 and Figure 5 show the results on MUTAG when varying the number of dimensions and the number of layers, respectively. Figure 11 and Figure 12 show the results on PROTEINS when varying the number of dimensions and the number of layers, respectively.

**Correlation between DIDM model's distance and maximal MPNN distance.**

Corollary 12 states that "convergence in DIDM mover's distance" is equivalent to "convergence in the MPNN's output *for all MPNNs*". The previous experiment depicts Corollary 12 only vaguely, since the experiment uses a single MPNN, and does not check the output distance for all MPNNs. Instead, in this experiment we would like to depict the "for all" part of Corollary 12 more closely. Since one cannot experimentally apply all MPNNs on a graph, we instead randomly choose 100 MPNNs for the whole dataset. Denote the set containing the 100 MPNNs by $\mathcal{N}'$. To verify the "for all" part, given each pair of graphs, we evaluate the distance between the MPNN's outputs on the two graphs for each MPNN and return the maximal distance. We plot this maximal distance against the DIDM mover's

distance. Namely, we plot $\max_{\mathfrak{H} \in \mathcal{N}'} \|\mathfrak{H}(\texttt{GIN\_meanpool}, G_i) - \mathfrak{H}(\texttt{GC\_meanpool}, G)\|_2$ and $\max_{\mathfrak{H} \in \mathcal{N}'} \|\mathfrak{H}(\texttt{GIN\_meanpool}, G_i) - \mathfrak{H}(\texttt{GC\_meanpool}, G)\|_2$ against $\delta^2_{\mathrm{DIDM}}(G_i, G)$. Note that in each experiment, we initialize $\texttt{GIN\_meanpool}$ and $\texttt{GC\_meanpool}$ only once with random weights and then compute the hidden representations of all graphs.

In more details, we checked the extent to which $\delta^2_{\mathrm{DIDM}}$ correlates with the maximal distance of 100 MPNNs' vectorial representation distances on MUTAG dataset and marked the Lipschitz relation. Here, we randomly generated 100 MPNNs for the entire dataset. Figure 3 showcase different Lipschitz relation. Note that the results are normalized.

From this, one can estimate a bound on the Lipschitz constants of all MPNNs from the family.

**The Random MPNN Distance conjecture.** Observe that in our experiments we plotted the MPNN's output distance for random MPNNs, not for "all MPNNs," and still got a nice correlation akin to Corollary 12. This leads us to the hypothesis that randomly initialized MPNNs have a fine-grained expressivity property: for some distribution over the space of MPNNs, a sequence of graph-signals converges in DIDM mover's distance if and only if the output of the sequence under a random MPNN converges in high probability. See Figure 10 for a comparison of $\delta^2_{\mathrm{DIDM}}$'s correlation with the maximal distance of 100 MPNNs' vectorial representation distances, with $\delta^2_{\mathrm{DIDM}}$'s correlation with the mean distance of 100 MPNNs' vectorial representation distances, and with $\delta^2_{\mathrm{DIDM}}$'s correlation with a single MPNN's vectorial representation distances on MUTAG dataset.

Table 1: Graph distances classification accuracy of 1-NN. $\delta_{W,3}$ and $\delta_{W,\geq 3}$ results are taken from Böker et al. (2023). $d_{WL}^{(3)}$ and $d_{WLLB,\geq 3}^{(3)}$ results are taken from Chen et al. (2022) using node degrees as initial labels. The table shows the mean classification accuracy of 1-NN using different graph distances using node degrees as initial labels.

| **Accuracy** $\uparrow$ | MUTAG |
|---|---|
| Chen et al. (2022) $d_{WL}^{(3)}$ | $91.1 \pm 4.3$ |
| Chen et al. (2022) $d_{WLLB,\geq 3}^{(3)}$ | $85.2 \pm 3.5$ |
| Böker et al. (2023) $\delta_{W,3}$ | $87.89 \pm 4.11$ |
| Böker et al. (2023) $\delta_{W,\geq 3}$ | $86.32 \pm 4.21$ |
| Chuang & Jegelka (2022) $\mathbf{TMD}^3$ | $89.47 \pm 7.81$ |
| $\delta_{DIDM}^2$ | $89.47 \pm 7.81$ |

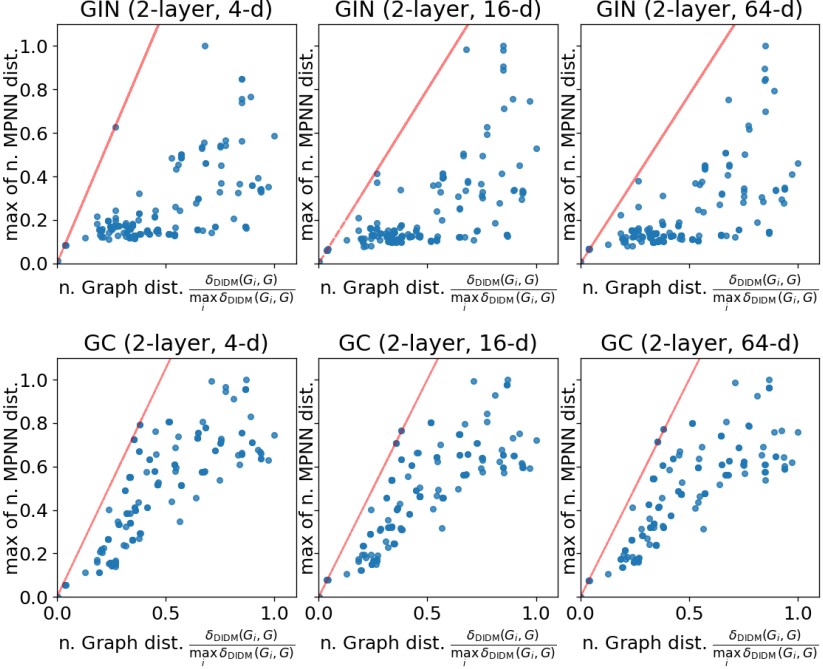

Figure 3: Correlation between $\delta_{DIDM}^2$ and the maximum over distances in the outputs of 100 randomly initialized MPNNs with a varying number of hidden dimensions. The Lipschitz bound is marked by the red line.

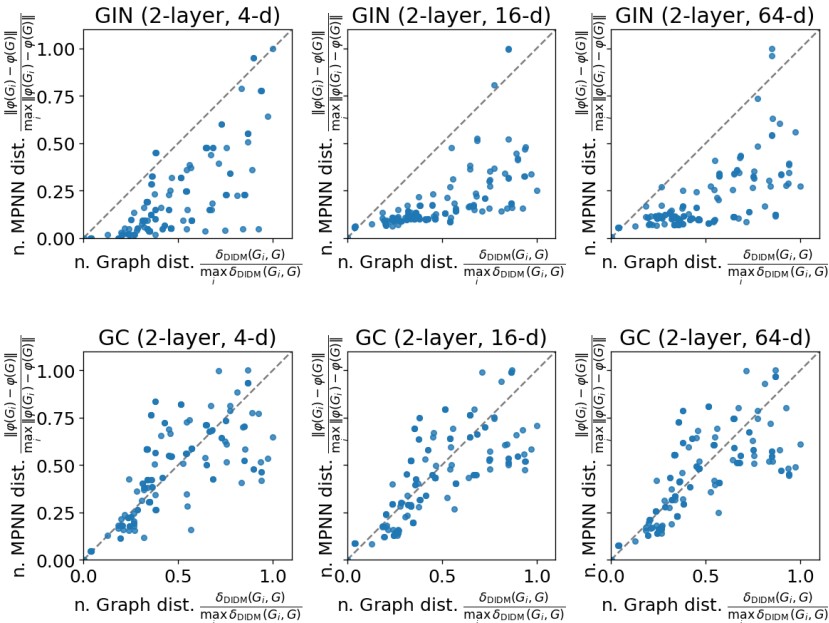

Figure 4: Correlation between $\delta^2_{\mathrm{DIDM}}$ and distance in the output of a randomly initialized MPNN with a varying number of hidden dimensions. GraphConv embeddings preserve graph distance better than GIN on MUTAG dataset.

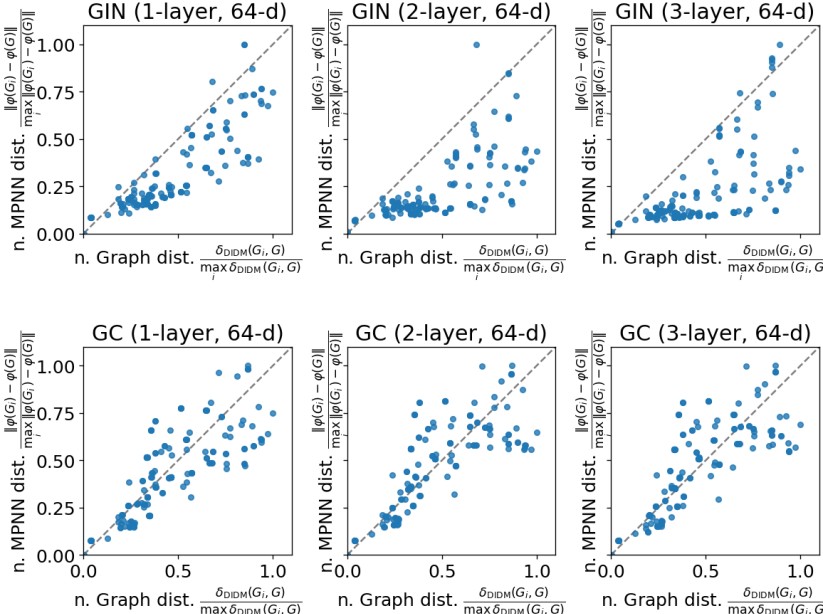

Figure 5: Correlation between $\delta^2_{\mathrm{DIDM}}$ and distance in the output of a randomly initialized MPNN with a varying number of layers. GraphConv embeddings preserve graph distance better than GIN on MUTAG dataset.

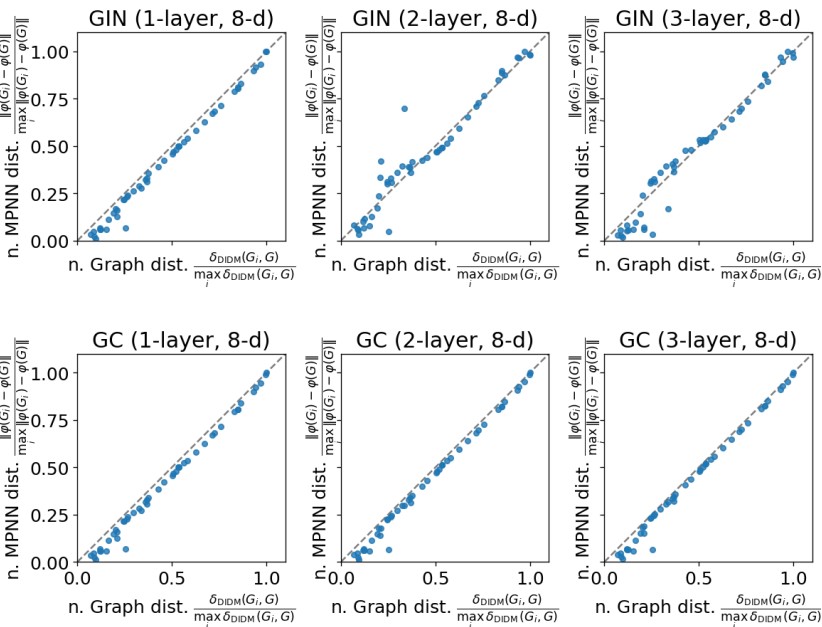

Figure 6: Correlation between $\delta^2_{\mathrm{DIDM}}$ and distance in the output of a randomly initialized MPNN with a varying number of layers. A convergent sequence of graphs with a constant signal. The graphs are generated by a stochastic block models.

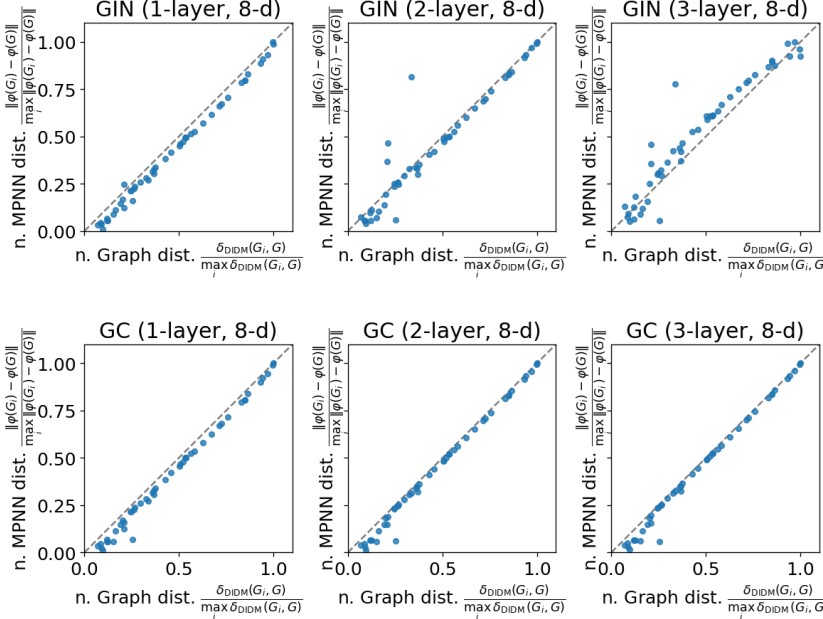

Figure 7: Correlation between $\delta^2_{\mathrm{DIDM}}$ and distance in the output of a randomly initialized MPNN with a varying number of layers. A convergent sequence of graphs with a signal, such that the signal values are constant each graph's community. Each signal value is sampled from a uniform distribution over $[0, 1]$. The graphs are generated by a stochastic block models.

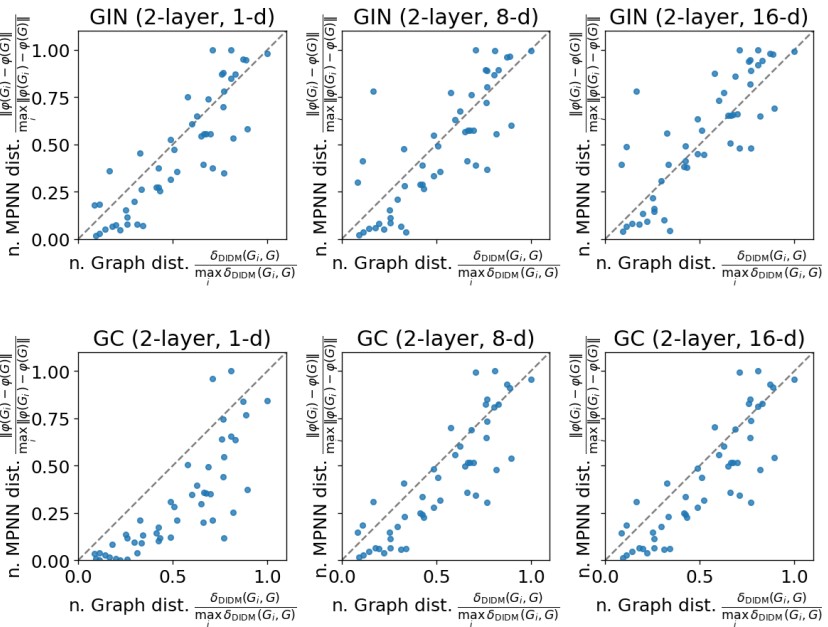

Figure 8: Correlation between $\delta^2_{\mathrm{DIDM}}$ and distance in the output of a randomly initialized MPNN with a varying number of hidden dimensions. A convergent sequence of graphs with a signal, such that the signal values are sampled from a normal distribution $\mathcal{N}(\mu, \sigma_i)$ with mean $\mu = 1$ and linearly decreasing variance. The graphs are generated by a stochastic block models.

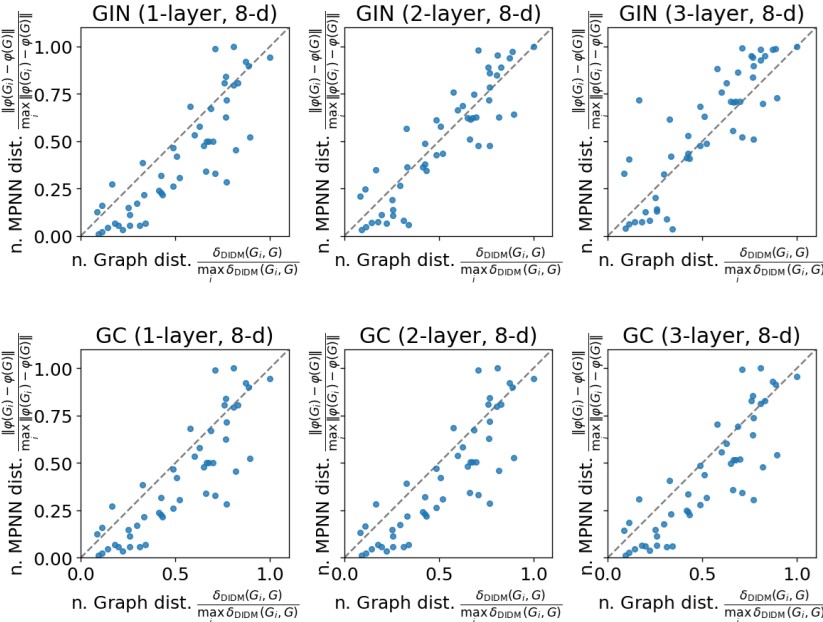

Figure 9: Correlation between $\delta^2_{\mathrm{DIDM}}$ and distance in the output of a randomly initialized MPNN with a varying number of layers. A convergent sequence of graphs with a signal, such that the signal values are sampled from a normal distribution $\mathcal{N}(\mu, \sigma_i)$ with mean $\mu = 1$ and linearly decreasing variance. The graphs are generated by a stochastic block models.

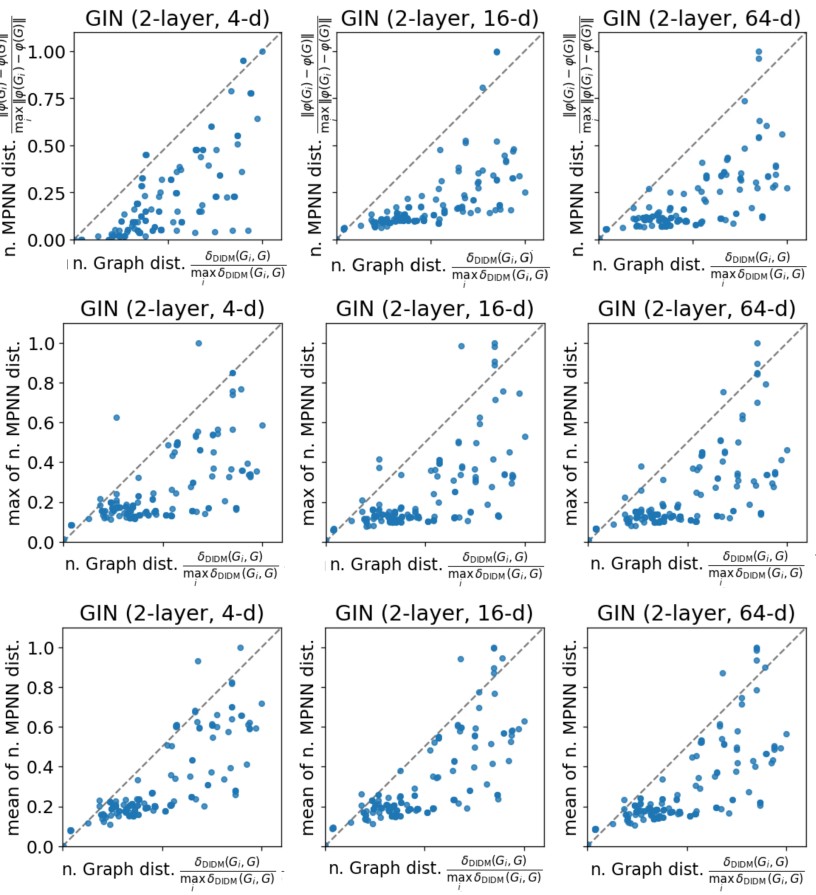

Figure 10: Correlation between $\delta^2_{\mathrm{DIDM}}$ and distance in the output of randomly initialized MPNNs with a varying number of hidden dimensions. On the top row of figures, the correlation between $\delta^2_{\mathrm{DIDM}}$ and distance in the output of a single randomly initialized MPNNs is presented. On the middle row of figures, the correlation between $\delta^2_{\mathrm{DIDM}}$ and the maximum over distances in the outputs of 100 randomly initialized MPNNs is presented. On the bottom row of figures, the correlation between $\delta^2_{\mathrm{DIDM}}$ and the mean over distances in the outputs of 100 randomly initialized MPNNs is presented.

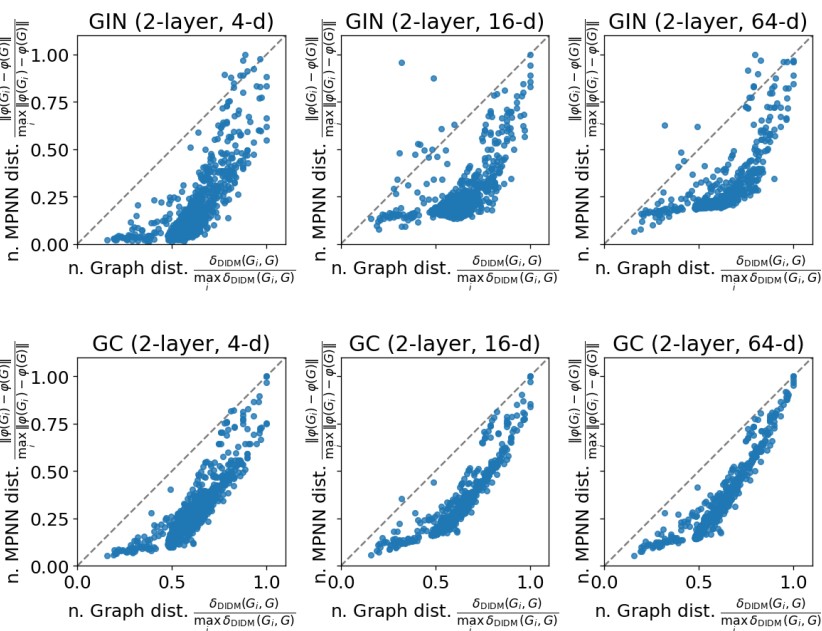

Figure 11: Correlation between $\delta^2_{\mathrm{DIDM}}$ and distance in the output of a randomly initialized MPNN with a varying number of dimensions. GraphConv embeddings preserve graph distance better than GIN on PROTEINS dataset.

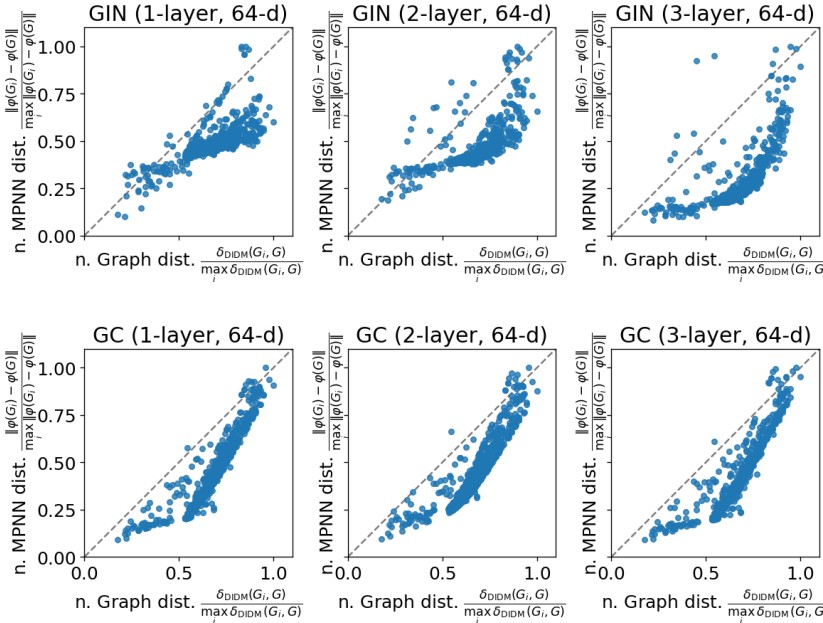

Figure 12: Correlation between $\delta^2_{\mathrm{DIDM}}$ and distance in the output of a randomly initialized MPNN with a varying number of layers. GraphConv embeddings preserve graph distance better than GIN on PROTEINS dataset.

## N    LIST OF NOTATIONS

### Sets and Graphs

| | |
|---|---|
| $\mathcal{A} \times \mathcal{C}$ | The cartesian product of two sets $\mathcal{A}$ and $\mathcal{C}$ |
| $\mathbb{N}_0$ | The set of natural numbers including 0. |
| $\mathbb{R}$ | The set of real numbers |
| $\mathbb{R}^d$ | The set of real vectors of length d |
| $\mathbb{B}_r^d$ | A fixed compact sub-set of $\mathbb{R}^d$, (Page 3) |
| $C_b(\mathcal{X})$ | The set of all bounded continuous real-valued functions on $\mathcal{X}$, (Page 3) |
| $C(\mathcal{X}, \mathcal{Y})$ | The set of all continuous functions from $\mathcal{X}$ to $\mathcal{Y}$ |
| $\mathcal{K}$ | A compact space |
| $\{0, 1\}$ | The set containing 0 and 1 |
| $\{0, 1, \ldots, n\}$ | The set of all integers between (and including) 0 and $n$ |
| $[n]$ | The set of all integers between (and including) 0 and $n$ |
| $[a, b]$ | The real interval including $a$ and $b$ |
| $I$ | A real interval |
| $(a, b]$ | The real interval excluding $a$ but including $b$ |
| $(G, \mathbf{f})$ | A graph-signal (Page 4) |
| $(H, , \mathbf{f})$ | A graph-signal |
| $(W, f)$ | A graphon-signal (Page 4) |
| $(Q, g)$ | A graphon-signal |
| $V(G)$ | The set of nodes of the graph G |
| $V(W)$ | The set of nodes of the graphon W |
| $E(W)$ | The set of edges of the graphon W |
| $\mathcal{N}(v)$ | The set of all neighboring nodes of a graph node $v$ |

### Calculus

| | |
|---|---|
| $\int f(\boldsymbol{x})d\boldsymbol{x}$ | Definite integral over the entire domain of $\boldsymbol{x}$ |
| $\int_{\mathcal{S}} f(\boldsymbol{x})d\boldsymbol{x}$ | Definite integral with respect to $\boldsymbol{x}$ over the set $\mathcal{S}$ |

### Numbers and Arrays

| | |
|---|---|
| $x$ | Element of a set, can be both a scalar or a vector |
| $a_i$ | Element $i$ of vector or a sequence |
| $C$ | A matrix |
| $D$ | A matrix |
| $\boldsymbol{X}$ | A random (either multi or single) variable |

### Measure Theory and Iterated degree Measures

| | |
|---|---|
| $\gamma_{(W,f),L}$ | Computation iterated degree measure (Page 5) |
| $\Gamma_{(W,f),L}$ | Computation distribution of iterated degree measure (Page 5) |
| $f_*\mu$ | the push-forward of a measure $\mu \in \mathscr{M}(\mathcal{X})$ via a measurable map $f : \mathcal{X} \to \mathcal{Y}$ |
| $\mu_i$ | The $i$'th element of an IDM $\mu$ |
| $\mu(i)$ | The $i$'th element of an IDM $\mu$ |
| $\mathcal{F}$ | A $\sigma$-algebra |
| $\Sigma$ | A $\sigma$-algebra |
| $\mathcal{B}(\mathcal{X})$ | The standard Borel $\sigma$-algebra of a measurable space $\mathcal{X}$ |
| $(\mathcal{X}, \Sigma)$ | a mesurable space |
| $(\mathcal{X}, \Sigma, \mu)$ | a measure space |
| $\mathscr{M}_{\leq 1}(\mathcal{X})$ | The space of all non negative Borel measures with total mass at most one on $(\mathcal{X}, \mathcal{B}(\mathcal{X}))$ (Page 3) |
| $\mathscr{P}(\mathcal{X})$ | The space of all Borel probability measures on $(\mathcal{X}, \mathcal{B}(\mathcal{X}))$ (Page 3) |
| $\mathcal{H}^d$ | The space of iterated degree measures (IDMs) of order $d$ (Page 5) |
| $\mathscr{P}(\mathcal{H}^d)$ | The space of distributions of iterated degree measures (DIDMs) of order $d$ (Page 5) |



**Metrics**



| | |
|---|---|
| $d$ | A metric |
| $\|\boldsymbol{x}\|_p$ | $\ell_p$ norm of $\boldsymbol{x}$ |
| $\|\boldsymbol{x}\|_\infty$ | the infinity norm of $\boldsymbol{x}$ |
| $\mathbf{OT}_d$ | Optimal transport with respect to the distance function $d$ (Page 3) |
| $d_{\mathrm{IDM}}$ | IDM distance (Page 6) |
| $\delta_{\mathrm{DIDM}}$ | DIDM mover's distance (Page 6) |
| $\mathbf{P}_d$ | Prokhorov metric with respect to the distance function $d$ (Page 48) |
| $p_{\mathrm{IDM}}$ | IDM Prokhorov distance (Page 48) |
| $\rho_{\mathrm{DIDM}}$ | DIDM Prokhorov's distance (Page 48) |



**Probability and Information Theory**



| | |
|---|---|
| $\mathcal{N}(\boldsymbol{\mu}, \boldsymbol{\Sigma})$ | Gaussian distribution with mean $\boldsymbol{\mu}$ and covariance $\boldsymbol{\Sigma}$ |



**Functions**



| | |
|---|---|
| $f : \mathcal{A} \to \mathcal{C}$ | The function $f$ with domain $\mathcal{A}$ and range $\mathcal{C}$ |
| $f(x)$ | The function $f$ evaluated at a point $x$ |
| $f(-)$ | The function $f$ evaluated at some point |
| $f_-$ | The function $f$ evaluated at some point |
| $f \circ g$ | Composition of the functions $f$ and $g$ |
| $\log x$ | Natural logarithm of $x$ |
| $\mathbb{1}_{\text{condition}}$ | is 1 if the condition is true, 0 otherwise |
| $\hat{\mathcal{R}}$ | the empirical risk (Page 9) |
| $\mathcal{R}$ | the statistical risk (Page 9) |

**Message Passing Neural Networks (Page 7)**

| | |
|---|---|
| $\varphi$ | A $L$-layer MPNN model |
| $(\varphi, \psi)$ | A $L$-layer MPNN model with $\psi$ a readout function |
| $\varphi^{(t)}$ | An update function |
| $\psi$ | An readout function |
| $\mathfrak{g}_-^{(t)}$ | $L$-layer MPNN model graph-signal feutures for $t \in [L]$ |
| $\mathfrak{G}$ | $L$-layer MPNN model with readout graph-signal feutures |
| $\mathfrak{f}_-^{(t)}$ | $L$-layer MPNN model graphon-signal feutures for $t \in [L]$ |
| $\mathfrak{F}$ | $L$-layer MPNN model with readout graphon-signal feutures |
| $\mathfrak{h}_-^{(t)}$ | $L$-layer MPNN model IDM feutures for $t \in [L]$ |
| $\mathfrak{H}$ | $L$-layer MPNN model with readout DIDM feutures |

