# OpenReview forum: "Generalization, Expressivity, and Universality of Graph Neural Networks on Attributed Graphs"
_ICLR.cc/2025/Conference — ICLR 2025 Poster_

### Official Review · Reviewer_xDjn · 2024-10-28

**Soundness:** 4
**Presentation:** 3
**Contribution:** 4
**Rating:** 8
**Confidence:** 4

**Summary:**

The present manuscript studies the approximation and generalization capabilities of message passing GNNs on attributed graphs under the lens of  proposed pseudometrics, based on the Wasserstein distance between distributions of computation tree analogs over graphons. It is demonstrated that such a pseudo metric leads to the characterization of a topology that allows to prove that message passing GNNs are universal approximations over attributed graphons, which are the fine-grained version of attributed graphs. Furthermore,  bounds on the generalization capability are provided exploiting the compactness and the lipschitzness of the retrieved topology. The theoretical findings are supported by an extensive experimental framework.

**Strengths:**

This manuscript provides a solid theoretical analysis of message passing GNNs under the framework of attributed graphons. The framework studies many properties, from universal approximation to generalization capabilities, therefore yielding a comprehensive overview of message passing GNNs that integrates in several ways the contribution already present in literature.

**Weaknesses:**

It is not explained why GINs and GraphConvs where chosen for the empirical evaluation, among the large class of message passing GNNs present in literature. I would suggest to discuss whether GIN and GraphConv are representative of different types of MPNNs, or if they have properties that make them particularly suitable for evaluating the theoretical results.

A (short) discussion of the results of the empirical evaluation over the generalization bound is missing. I would suggest to discuss how closely the empirical results align with the theoretical bound, also considering the statistical nature of the task, or if there are any unexpected patterns or discrepancies that needs some careful consideration and warrant further investigation.

**Questions:**

Can the authors provide a motivation for the choice of GINs and GraphConvs for the empirical evaluation?
Can the authors provide a short discussion of the results of the empirical evaluation over the generalization bound?

---

> ### Author Response · Authors · 2024-11-24
> **Rebuttal Revisions**
>
> We thank the reviewer for their positive assessment of our paper and their insightful comments.
> *For the convenience of the reviewer, we merge related comments. In our responses, we quote any relevant revision we made in the paper. This led to a seemingly longer rebuttal, but we believe that it will streamline the work for the reviewer.*
>
> >>**Weaknesses and Questions:**
>
> >> Weakness 1. It is not explained why GINs and GraphConvs where chosen for the empirical evaluation, among the large class of message passing GNNs present in literature. I would suggest to discuss whether GIN and GraphConv are representative of different types of MPNNs, or if they have properties that make them particularly suitable for evaluating the theoretical results.
>
> >>Question 1. Can the authors provide a motivation for the choice of GINs and GraphConvs for the empirical evaluation?
>
> **Response.** Thank you for this comment. Our work is theoretical, and the target application for our theory is theorem proving, not practical computational methods, therefore we focused mostly on proof-of-concept experiments. We hence chose very well-known and simple MPNN architectures. To clarify this, we added the following discussions.
>
>  - We emphasized the goal of the experiments.
>
> "The goal of our experiments is to support the theoretical results which formulate a form of equivalence between GNN outputs and DIDM mover's distance. We emphasize that our proposed DIDM mover's distance metric is mainly a tool for theoretical analysis and the proposed experiments are not designed to compete with state-of-the-art methods. Although our metric is not intended to be used directly as a computational tool, our results suggest that we can roughly approximate the DIDM mover's distance between two graphs by the Euclidean distance between their outputs under random MPNNs. This Euclidean distance can be used in practice as it is less computationally expensive than DIDM mover's distance." (Page 41 Lines 2192-2199)
>
> "As a proof of concept, we empirically test the correlation between $\delta_{\rm DIDM}^L$ and distance in the output of MPNNs. We hence chose well-known and simple MPNN architectures, varying the hidden dimensions and number of layers. We do not claim that GIN [1] and GraphConv [2] are representative of the variety of all types of MPNNs. Nevertheless, they are proper choices for demonstrating our theory in practice."(Page 42, Lines 2247-2251)
>
> >>Weakness 2. A (short) discussion of the results of the empirical evaluation over the generalization bound is missing. I would suggest to discuss how closely the empirical results align with the theoretical bound, also considering the statistical nature of the task, or if there are any unexpected patterns or discrepancies that needs some careful consideration and warrant further investigation.
>
> >> Question 2. Can the authors provide a short discussion of the results of the empirical evaluation over the generalization bound?
>
> **Response.**  You are right, the experiments we presented in Section 5 of the old version of the paper are not directly related to generalization. They merely show that the Lipschitz continuity of MPNNs with respect to our metric can be empirically observed. As this is an essential property to have for robustness-type generalization theorems, such as the generalization theorem we presented, we initially decided to present this experiment in Section 5.
>
> However, in the revised paper, we refer the reader to experiments that are more relevant for the generalization theorem and moved our previous presented experiments from the end of Section 5 to appendix M.
>
> "The only nonstandard part in our construction is the choice of  normalized sum aggregation in our MPNN architecture, while most MPNNs use sum, mean, or max aggregation. We justify this choice as follows. First, experimentally, in [4: Tables 1, 3] and [3: Table 2], it is shown that MPNNs with normalized sum aggregation generalize well, comparably to MPNNs with sum aggregation. We believe that this is the case since in most datasets most graphs have roughly the same order of vertices. Hence, the normalization by $N^2$ is of a constant order of magnitude and can be swallowed by the weights of the MPNN, mimicking the behavior of an MPNN with sum aggregation. Future work may explore extensions of our theory to other aggregation schemes (see Appendix A.3 and Appendix B.2 for more details)." (Page 10 Lines: 523-531)
>
> Moreover, we significantly extended the discussion about the empirical evaluation in Appendix M, pages 42-51. For more details about the changes we made, see the response for Questions 1 and 2 of reviewer uw7j.

---

> > ### Author Response · Authors · 2024-11-24
> > **Rebuttal Revisions**
> >
> > **References.**
> >
> > [1] Xu, et al., 2019. How Powerful are Graph Neural Networks? ICLR.
> >
> > [2] Morris, et al., 2019. Weisfeiler and Leman Go Neural: Higher-order Graph Neural Networks. AAAI Conference on Artificial Intelligence.
> >
> > [3] Levie, 2023. A Graphon-signal Analysis of Graph Neural Networks. NeurIPS.
> >
> > [4] Böker, et al., 2023. Fine-Grained Expressivity of Graph Neural Networks. NeurIPS.

---

> > ### Comment · Reviewer_xDjn · 2024-11-25
> >
> > The authors addressed my (few) concerns in an exhaustive manner, therefore I recommend this paper for acceptance at ICLR 2025.

---

> > > ### Author Response · Authors · 2024-11-29
> > >
> > > We thank the reviewer again for their constructive comments, which improved our paper.

---

### Official Review · Reviewer_9Cqp · 2024-10-28

**Soundness:** 3
**Presentation:** 2
**Contribution:** 2
**Rating:** 6
**Confidence:** 3

**Summary:**

This paper theoretically shows that the closeness of two graphs by MPNNs can be represented by the proposed metrics based on Wasserstein Distance. This work also shows the existence of MPNNs that can separate two graphs with non-zero distance and can approximate any functions over the space of graphs. A bound of generalization gap is also provided for MPNNs with milder assumptions.

**Strengths:**

1. The theoretical analysis is solid and strong. Especially, this work tries to provide an analysis that is as universal as possible with the fewest assumptions.
2. The discussion of MPNNs and tree mover's distance in Section 4 is interesting and novel to me.

**Weaknesses:**

1. The presentation is a bit strange. The main results in Section 5 appear too late on almost page 9 in the main body of the paper. Moreover, the universal approximation and the generalization bound seem independent of the previous analysis of graph/graphon distances. It is better to move Section 5 earlier and elaborate more about the connections with previous sections.

2. Section 5 is not insightful enough. For example, although Theorem 7 makes contributions by requiring fewer assumptions, experiments in Figure 3 show that the generalization error is still related to the covering number and the Lipschitz constant, which is unsurprising and known in generalization theory.

3. Some references on the generalization error of MPNNs are missing.

Oono, K. and Suzuki, T. Optimization and generalization analysis of transduction through gradient boosting and application to multi-scale graph neural networks. Neurips 2020.

Li et al., 2022. Generalization guarantee of training graph convolutional networks with graph topology sampling. ICML 2022.

Tang et al., 2023. Towards understanding the generalization of graph neural networks. ICML 2023.

**Questions:**

1. Can your theoretical analysis extend to the node classification problem?

2. Is Theorem 7 related to any distance metric proposed in Sections 3 and 4? If not, can the analysis in Sections 3 and 4 be used to derive a generalization bound?

---

> ### Author Response · Authors · 2024-11-24
> **Rebuttal Revisions**
>
> We thank the reviewer for their positive assessment of our paper and their insightful comments.
> *For the convenience of the reviewer, we merge related comments. In our responses, we quote any relevant revision we made in the paper. This led to a seemingly longer rebuttal, but we believe that it will streamline the work for the reviewer.*
>
> >**Weaknesses and Questions**
> >>Weakness 1. The presentation is a bit strange. The main results in Section 5 appear too late on almost page 9 in the main body of the paper. Moreover, the universal approximation and the generalization bound seem independent of the previous analysis of graph/graphon distances. It is better to move Section 5 earlier and elaborate more about the connections with previous sections.
>
> >> Weakness 2. Section 5 is not insightful enough. For example, although Theorem 7 makes contributions by requiring fewer assumptions, experiments in Figure 3 show that the generalization error is still related to the covering number and the Lipschitz constant, which is unsurprising and known in generalization theory.
>
> >>Question 2. Is Theorem 7 related to any distance metric proposed in Sections 3 and 4? If not, can the analysis in Sections 3 and 4 be used to derive a generalization bound?
>
> **Response.** **We totally agree that the structure and clarity of the old version of the paper can be significantly improved.** We hence extensively revised the paper to enhance readability.
>
> Our answer consists of two parts. First, we clarify our contributions and the connections between the different sections in the article. We Then list some of the main changes we have made to improve clarity.
>
> While it is well known how to bound the generalization error of Lipschitz continuous models over compact space (e.g., robustness analysis [4] and [6]), this is not the contribution of our paper, but rather a direct application of our main contribution. Both the universal approximation (Theorem 15) and generalization analysis (Theorem 7) are not independent of the compactness of the space of DIDMs and the fine-grained expressivity of MPNNs (Lipschitz continuity and equivalence of convergence of graph sequences to convergence of MPNN outputs). The universal approximation and generalization are direct applications of these results, which are derived from the definition of DIDM mover's distance.
>
> Moreover, Theorem 7 uses the covering number, which is directly determined by the metric. Other than the covering number, the theorem does not explicitly use the metric. However, the metric also affects the Lipschitz constant of the MPNNs and hence affects the generalization bound. Moreover, the metric is used in the proof.
>
> To summarize, the contribution of our paper is to show how to build a compact space of all graph-signals under which GNNs are Lipschitz continuous, where GNN separate points with respect to this metric. We are the first to propose such a construction. Once we constructed and proved the above, that the generalization bound is a direct application. With this said, our main results are in Section 3 and Section 4, whereas Section 5 contains direct applications of our construction. We edited the text in the paper to make this clear.
>
> We made the following changes to emphasize the above and to make the article contributions more transparent.
>
> - we added the following sentences to the introduction to emphasize our main goals and the type of generalization bound we discuss.
>
> "In this paper, we define a pseudometric for attributed graphs which is highly related to the type of computation message passing GNNs (MPNNs) perform. Our  construction enables a unified analysis of expressivity, universal approximation and generalization of MPNNs on graphs with node attributes." (Page 1 Lines 44-47)
>
> "In this work, we close these gaps via a unified approach that allows for an analysis of expressivity, universal approximation, and generalization." (Page 2 Lines 76-77)
>
> "Moreover, a robustness-type generalization theorem over the entire space, which depends on the space having a finite covering number,..." (Page 1 Lines 63-64)

---

> ### Author Response · Authors · 2024-11-24
> **Rebuttal Revisions**
>
> - We extended and modified the Contribution subsection, and clarify our contributions and to better communicate what our main results and focus to the reader.
>
> "**Contributions.** We propose the first metric for attributed graphs under which the space of attributed graphs is compact and MPNNs are Lipschitz continuous and separate points. Our construction leads to the first theory of MPNNs that unifies expressivity, universality, and generalization on any data distribution of attributed graphs. In detail:
>
>  - We show a *fine-grained* metric version of the separation power of MPNNs, extending the results of [5] to attributed graphs}: two graph-signals are close in our metric if and only if the outputs of all MPNNs on the two graphs are close. Hence, the geometry of graphs (with respect to our metric) is equivalent to the geometry of the graphs' representations via MPNNs (with respect to the Euclidean metric).
>
>  - We prove that the space of attributed graphs with our metric is compact and MPNNs are Lipschitz continuous (and separate points). This leads to two theoretical applications: (1) A universal approximation theorem, i.e., MPNNs can approximate any continuous function over the space of attributed graphs. (2) A generalization bound for MPNNs, akin to robustness bounds [4], requiring no assumptions on the data distribution or the number of GNN weights." (Page 2 Lines 89-102)
>
> - We changed the name of section 5 to "Theoretical Applications", as the previous title "main results" was misleading. The results in Section 5 are better interpreted as applications of our main results, where the main results are in Section 3,4. The main results are the compactness of the space of attributed graphs/DIDMs with respect to the DIDM mover's distance, and the Lipschitz continuity of MPNNs with respect to the DIDM mover's distance, and the equivalency between convergence in our metrics and convergence in MPNN's outputs.
>
> "**5 Theoretical Applications**
> Next, we state the main theoretical applications of the compactness of the space of attributed graphs and the Lipschitz and separation power of MPNNs with respect to DIDM mover's distance." (Page 9, Lines 463 -466)
>
> - We emphasized and added sentences in the revised version of the paper so it is clear that the generalization bound is an application, and that we are not claiming that the idea of using Lipschitz continuous functions over compact spaces to derive generalization bounds is new.
>
> "The following generalization theorem uses the same techniques as [6]." (Page 10, Line 494)
>
> "These two properties together lead to a uniform generalization bound, akin to robustness-type bounds [4]." (Page 10, Lines 496-497)
>
> - We modified some of the terminology and changed many notations to improve readability. For example, we changed the notation of computation IDMs and DIDMs from $\mathscr{H}$ and $\mathscr{G}$ to $\gamma$ and $\Gamma$ and updated the MPNN features' notations, to make it easier to follow the dependencies of each feature on its corresponding MPNN model and to clarify the dependency of the features on the input object. Now, overall, we have fewer versions of H in different fonts denoting different things, and we give more meaningful names to the different terms. For more details see the response to Weakness 5 mentioned by Reviewer Ku8V.
>
> >>Weakness 3. Some references on the generalization error of MPNNs are missing.
>
> >>Oono, K. and Suzuki, T. Optimization and generalization analysis of transduction through gradient boosting and application to multi-scale graph neural networks. Neurips 2020.
>
> >>Li et al., 2022. Generalization guarantee of training graph convolutional networks with graph topology sampling. ICML 2022.
>
> >>Tang et al., 2023. Towards understanding the generalization of graph neural networks. ICML 2023.
>
> **Response.**  Thank you for the additional references. We added them in the following locations both in the main paper and in the appendix (Appendix B.3).
>
> "...and generalization [1,2,3]..."(Page 1, Lines 35-36)
>
> "A number of existing works study generalization for GNNs, e.g., via VC dimension, Rademacher complexity or PAC-Bayesian analysis [1,2,3]..."(Page 2, Lines 115-118)
>
> "In comparison to other recent works on generalization...[2] analysis is restricted to graph convolution networks which are a special case of MPNNs. [3] does not focus on graph classification tasks but on node classification. [1] focus on transductive learning in contrast to our inductive learning analysis."(Page 22, lines 1145-1153)
>
> - We reference Appendix B.3 right after first introducing the generalization theorem (Theorem 7).
>
> "For a comparison of our generalization bound to other bounds in the literature, see Appendix B.3." (Page 9, Lines 515-516)

---

> ### Author Response · Authors · 2024-11-24
> **Rebuttal Revisions**
>
> >>Question 1. Can your theoretical analysis extend to the node classification problem?
>
> **Response.** We added the following comment on this in Section 6 (Discussion).
>
> "Moreover, the current theory is designed around graph level tasks. However, potentially, one can also use IDMs to study node level tasks, as IDMs represent the computational structure corresponding to each node. For example, one can potentially use our construction for analysing the stability of node-level GNNs to perturbations to the structure of the graph and its features, where the magnitude of the perturbation is modeled via IDM distance." (Page 10, Lines 535-538)
>
> **References**
>
> [1] Oono, et al., 2020. T. Optimization and generalization analysis of transduction through gradient boosting and application to multi-scale graph neural networks. NeurIPS.
>
> [2] Li et al., 2022. Generalization guarantee of training graph convolutional networks with graph topology sampling. ICML.
>
> [3] Tang et al., 2023. Towards understanding the generalization of graph neural networks. ICML 2023.
>
> [4] Xu, et al., 2012. Robustness and generalization. COLT.
>
> [5] Böker, et al., 2023. Fine-Grained Expressivity of Graph Neural Networks. NeurIPS.
>
> [6] Levie, 2023. A Graphon-signal Analysis of Graph Neural Networks. NeurIPS.

---

> > ### Comment · Reviewer_9Cqp · 2024-11-27
> >
> > Thank you for the response. I have increased the rating to 6.

---

> > > ### Author Response · Authors · 2024-11-29
> > >
> > > We thank the reviewer again for their constructive comments. Their suggestions have significantly improved our paper.

---

### Official Review · Reviewer_uw7j · 2024-11-01

**Soundness:** 4
**Presentation:** 2
**Contribution:** 3
**Rating:** 6
**Confidence:** 2

**Summary:**

This paper develops a comprehensive theoretical framework that enhances the understanding of generalization, expressivity, and universality in Graph Neural Networks (GNNs) when applied to attributed graphs (graphs with node features). The central contribution is the introduction of a new metric, the Tree Mover’s Distance (TMD), designed to precisely measure the similarity between graphs by capturing both their structural and attribute-based characteristics. Key findings of this paper include the design of TMD,  the universal approximation of GNNs, and the analysis of the expressivity of GNNs.

**Strengths:**

1. The theoretical analysis is solid and thorough.
2. This work enhances the understanding of GNNs. The work introduces a new framework that unifies generalization and expressivity theories for attributed graphs, a notable achievement in GNN research.
3. The idea of introducing OT to this problem is quite novel and fancy.

**Weaknesses:**

See the questions below.

**Questions:**

1.  While the paper provides some empirical results, the evaluation could be expanded to offer deeper insights into practical applications and better demonstrate the efficacy of the proposed method. Have the authors considered evaluating TMD on additional graph classification benchmarks beyond MUTAG? Comparing TMD against other graph similarity metrics on tasks like graph matching or clustering could also strengthen the empirical evaluation. Furthermore, could the authors provide more detailed explanations of the experimental settings and parameter choices？

2. In Table 1, it would be helpful to clarify the organization and naming of the columns, as the label “Accuracy” appears to represent results across multiple methods. Providing a clearer breakdown of the results, along with an analysis that contextualizes each method’s performance, would enhance understanding. For example, if the statistics below MUTAG represent accuracy scores, they suggest that TMD does not significantly outperform some existing methods, such as Chen et al. (2022)’s  d^{(3)}_{WL} . A discussion of how TMD compares to these methods and whether additional datasets were considered may add further depth to the results.

3.  Structuring the manuscript to more prominently summarize the major findings and their impact would improve readability. Perhaps revise the contribution part in the introduction to better summarize the major findings. Also, perhaps adding a section to state the broader impact of these findings more clearly and how each finding advances the field of GNN theory to better illustrate the importance of this work.

4. The methods primarily focus on dense graphs, as noted in the conclusion that “sparse graphs are always considered close to the empty graph under our metrics.” A further explanation of this limitation—specifically, why sparse graphs exhibit this behavior under TMD—would benefit the reader. Additionally, a discussion of whether this limitation affects the method’s generality and its potential applications to sparse graph domains would strengthen the paper.

---

> ### Author Response · Authors · 2024-11-24
> **Rebuttal Revisions**
>
> We thank the reviewer for their positive assessment of our paper and their insightful comments.
> *For the convenience of the reviewer, we merge related comments. In our responses, we quote any relevant revision we made in the paper. This led to a seemingly longer rebuttal, but we believe that it will streamline the work for the reviewer.*
>
> >> **Questions:**
>
> >>Question 1. While the paper provides some empirical results, the evaluation could be expanded to offer deeper insights into practical applications and better demonstrate the efficacy of the proposed method. Have the authors considered evaluating TMD on additional graph classification benchmarks beyond MUTAG? Comparing TMD against other graph similarity metrics on tasks like graph matching or clustering could also strengthen the empirical evaluation. Furthermore, could the authors provide more detailed explanations of the experimental settings and parameter choices？
>
> >>Questions 2. In Table 1, it would be helpful to clarify the organization and naming of the columns, as the label “Accuracy” appears to represent results across multiple methods. Providing a clearer breakdown of the results, along with an analysis that contextualizes each method’s performance, would enhance understanding. For example, if the statistics below MUTAG represent accuracy scores, they suggest that TMD does not significantly outperform some existing methods, such as Chen et al. (2022)’s d^{(3)}_{WL} . A discussion of how TMD compares to these methods and whether additional datasets were considered may add further depth to the results.
>
>
> **Response.**
>
> Thank you for the comments. We included new experiments and a much more thorough description of all our experiments in the revised paper. Moreover, we agree that the previous versions of Appendix M and Table 1 were indeed unclear.
>
> The classifier based on our metric (In Table 1) is not meant to compete with state-of-the-art. Rather, the experiment is designed to showcase that our DIDM mover's distance is a meaningful graph similarity measure that captures well the "geometry" of real-life data. The competition between our metric and past metrics is not measured by the performance of this specific example. Rather, our metric "beats" all other metrics in its theoretical properties: it is the only metric that makes the space of attributed graphs compact along with making MPNNs Lipschitz continuous and separating points. These three results make our DIDM mover's distance the only metric that leads to universal approximation and generalization out of the competing metrics. While the metric of [6] leads to a better accuracy in this specific experiment, it is too fine to allow a universal approximation on the space of all attributed graphs and to allow a robustness-type generalization theorem based on a finite covering of the space of attributed graphs. Our metric both leads to good accuracy in the experiment (though not the best) and allows obtaining the theoretical results.
>
>  - In the revised paper, we emphasize the goal of the experiments as experiments.
>
> "The goal of our experiments is to support the theoretical results which formulate a form of equivalence between GNN outputs and DIDM mover's distance. We emphasize that our proposed DIDM mover's distance metric is mainly a tool for theoretical analysis and the proposed experiments are not designed to compete with state-of-the-art methods. Although our metric is not intended to be used directly as a computational tool, our results suggest that we can roughly approximate the DIDM mover's distance between two graphs by the Euclidean distance between their outputs under random MPNNs. This Euclidean distance can be used in practice as it is less computationally expensive than DIDM mover's distance." (Page 41 Lines 2192-2199)
>
> "As a proof of concept, we empirically test the correlation between $\delta_{\rm DIDM}^L$ and distance in the output of MPNNs. We hence chose well-known and simple MPNN architectures, varying the hidden dimensions and number of layers. We do not claim that GIN [19] and GraphConv [20] are representative of the variety of all types of MPNNs. Nevertheless, they are proper choices for demonstrating our theory in practice."(Page 42, Lines 2247-2251)

---

> ### Author Response · Authors · 2024-11-24
> **Rebuttal Revisions**
>
> - We added background on the 1-NN classification task.
>
> "**M.1 1-Nearest-Neighbor classifier**
> The goal of the experiment in this section is to show that the geometry underlying the metric $\delta_{\rm DIDM}$ captures in some sense the underlying data-driven similarity related to the classification task.
> We consider the problem of classifying   attributed graphs, and a solution based on the 1-nearest neighbor.
>
> **Definition 73.**
> The *1-Nearest Neighbor* (1-NN) *classifier* is a non-parametric, instance-based machine learning method.  Given a dataset $\mathcal{D} $ consist of samples $((G_i,\mathbf{f}_i), y_i)$ for $0<i\leq n.$
>
>
> where $(G_i,\mathbf{f}_i)$ represents graph-signals and $y_i \in \mathcal{C}$ denotes class labels from a finite set $\mathcal{C}$, the goal is to classify a new input $(G_i,\mathbf{f}_i)$.  The classification process of classifying the input $(G,\mathbf{f})$ involves:
>
> 1. Computing the distance between the input $(G,\mathbf{f})$ and every point $(G_i,\mathbf{f}_i)$ in the dataset using a distance metric $d$.
>
>
> 2. Identifying the nearest neighbor $(G_k,\mathbf{f}_k)$ such that:
>
> $$(G_k,\mathbf{f}_k) $$
>
> $$= \arg\min_{i \in n} d((G,\mathbf{f}), (G_i,\mathbf{f}_i)).$$
>
> 3. Assigning the label $y_k$ of the nearest neighbor as the label $y$ of $(G,\mathbf{f})$ : $y \xleftarrow{} y_k.$" (Pages 41-42 Lines 2200-2218)
>
> - We included a new discussion on each of the "competing" metrics and compared its definition to our DIDM mover's distance.
>
> "Here we chose to compare $\delta_{\rm DIDM}^2$ with other optimal-transport-based iteratively defined metrics. Tree Mover's Distance $\mathbf{TMD}^L$ from [4] is defined via optimal transport between finite attributed graphs through a so called computation trees. The Weisfeiler-Lehman (WL) distance $d^{(L)}_{\text{WL}}$ and its lower bound distance
>
> $d^{(L)}_{\text{WLLB},\geq L}$
>
> from [5,6] are defined via optimal transport between finite attributed graphs through hierarchies of probability measures. The metric $\delta_{W,L}$ from [7] is define via optimal transport on a variant of IDMs and DIDMs where the IDMs are not concatenated. The metric $\delta_{W,\geq L}$, from [7], is a variation of $\delta_{W,L}$ where the maximum number of iterations performed after having obtained a stable coloring is bounded by 3. Unlike $\delta_{\rm DIDM}$, all the above metrics cannot be used to unify expressivity, uniform approximation and generalization for attributed graphs. Namely, the above pseudometrics are either restricted to graphs without attributes or are not compact." (Page 42 Lines 2219-2229)
>
> - We clarified the meaning of the word accuracy used in the explanation of the experiment.
>
> "Table 1 compares the mean classification accuracy of $\delta_{\text{W},3}$, $\delta_{\text{W},\geq 3}$ [7],
>
> $d^{(3)}_{\text{WL}}$,
>
> $d^{(3)}_{\text{WLLB},\geq 3}$ [5,6],
>
>  $\mathbf{TMD}^3$ [4], and $\delta_{\rm DIDM}^2$ in a $1$-NN classification task using node degrees as initial labels." (Page 42 Lines 2231-2234)
>
> - We converted the following information from being a footnote to being a part of the main text:
>
> "We used 90  percent of the data for training and 10 percent of the data for testing." (Page 42 Lines 2236-2237)
>
> - We added interpretation to the results
>
> " We note that the $1$-NN classification experiment "softly" supports our theory, in the sense that this experiment shows that our metric clusters the graphs quite well with respect to their task-driven classes. We stress that this experiment does not directly evaluates any rigorous theoretical claim. We moreover note that while the metric in [6] achieves better accuracy, the space of all graphs under their metric is not compact, so this metric does not satisfy our theoretical requirements: a compact metric which clusters graphs well." (Page 42, Lines 2237-2242)
>
> - We clarified the meaning of the word accuracy used in the captions of Table 1.
>
> "The table shows the mean classification accuracy of 1-NN using different graph distances using node degrees as initial labels." (Page 45 Lines 2383-2384)

---

> ### Author Response · Authors · 2024-11-24
> **Rebuttal Revisions**
>
> - We added details on the MPNN architectures we used.
>
> " **M.2.1 MPNN Architectures**
>     The **GIN\_meanpool** model is a variant of the Graph Isomorphism Network (GIN) (see Appendix A.2.1) designed for graph-level representation learning. Each layer consists of normalized sum aggregation and a multi layer perceptron (MLP). The first MLP consists of two linear transformations, ReLU activations, and batch normalization. Each MLP that follows has additionally a skip connection and summation of the input features and output features. The readout after $L$ layers is mean pooling (with no readout function).
>
> The **GC_meanpool** model is a realization of graph convolution network (GCN) for graph-level representation learning.  Each layer consists of normalized sum aggregation with a linear message and update functions (see Appendix C for the definition of message function and for the equivalency between MPNNs that use message functions and MPNNs with no message functions). All layers except the first layer have additionally a skip connection and a summation of the input features and output features. The readout after $L$ layers is mean pooling (with no readout function)." (Page 42, Lines 2253-2267)
>
> - We added another experiment on the correlation between DIDM mover's distance and MPNN's output distance, based on a modified SBM graph generative model. As opposed to the original experiment from the paper, we considered  a signal which has a different constant value on each community of the graph (defined via the communities of the SBM). Each value is randomly sampled from a uniform distribution over $[0,1]$. We added more detail on these experiments as follows:
>
> "**Correlation Experiments on Graphs Generated from a Stochastic Block Model**
> We extend here the experiments presented in Section 4, with the same experimental procedure and also offer an extended discussion and description of the experiments. We empirically test the correlation between $\delta_{\rm DIDM}^L$ and the distance in the output distance of an MPNN. We use stochastic block models (SBMs), which are generative models for graphs, to generate random graph sequences. We generated a sequence of $50$ random graphs $\{G_i\}^{49}_{i=0}$,
>
> each with 30 vertices. Each graph is generated from an SBM with two blocks (communities) of size $15$ with $p=0.5$ and $q_i=0.1+0.4i/49$ probabilities of having an edge between each pair of nodes from the same block different blocks, respectively. We denote $G:=G_{49}$, which is an Erdős–Rényi model. We plot $\delta_{\rm DIDM}^2(G_i,G)$ against distance in the output of randomly initialized MPNNs, i.e., once against $\|\mathfrak{H}(GIN\_meanpool,G_i) - \mathfrak{H}(GC\_meanpool,G)\|_2$ and once against $\|\mathfrak{H}(GIN\_meanpool,G_i)- \mathfrak{H}(GC\_meanpool,G)\|_2$. Note that in each experiment, we initialize **GIN\_meanpool** and **GC\_meanpool** only once with random weights and then compute the hidden representations of all graphs twice, once for each randomly weighted architecture.
>
> We conducted the entire procedure twice, once with a constant feature attached to all nodes and once with a signal which has a different constant value on each community of the graph. Each value is randomly sampled from a uniform distribution over $[0,1].$ In section 4 We present the results of the experiments when varying hidden dimension (see Figure 2). Figure 6 and Figure 7 show the results when varying the number of layers when the signal is constant and when the signal has a different randomly generated constant value on each community, respectively. The results still show a strong correlation between input distance and GNN outputs. When increasing the number of layer, the correlation slightly weakens.
>
> We conducted the experiment one more time with signal values sampled from a normal distribution $\mathcal{N}(\mu , \sigma_i)$ with mean $\mu = 1$ and variance $\sigma_i=\frac{49-i}{49}$. Figure 8 and Figure 9 show the results when varying the number of dimensions and the number of layers, respectively. The results still show a  correlation between input distance and MPNN outputs, but with a higher variance. As we interpret this result, the increased variance could be an artifact of using random noise as signal in our experiments. The specific MPNNs we used, have either a linear activation or a ReLU activation function. Thus, they have a "linear" averaging effect on the signal, which cancels in a sense the contribution of the noise signal to the output of the MPNN, while the metric takes the signal into full consideration. This leads to a noisy correlation." (Page 43, Lines (2268 - 2299)

---

> ### Author Response · Authors · 2024-11-24
> **Rebuttal Revisions**
>
> - We added another correlation experiment on PROTEINS database as well as further clarified the setting.
>
> "**M.2.3 Real Datasets Correlation Experiments**
> We empirically test the correlation between $\delta_{\rm DIDM}^L$  and distance in the output of MPNNs on MUTAG and PROTEINS databases. In the following, we present the results that showcase insightful relations.
>
> **Correlation experiments using a single randomized MPNN.**
>  Denote by $\mathcal{D}$ a generic dataset. For the entire dataset we randomly initialized one MPNN with random weights. We randomly picked an attributed graph from the dataset $(\hat{G},\hat{\mathbf{f}})\in \mathcal{D}$. For each $(G,\mathbf{f})\in\mathcal{D}$ we computed $\delta_{\rm DIDM}^2((\hat{G},\hat{\mathbf{f}}),(G,\mathbf{f})).$ We plotted the distance in the output of the randomly initialized MPNNs on each $(G,\mathbf{f})\in\mathcal{D}$ against $\delta_{\rm DIDM}^2((\hat{G},\hat{\mathbf{f}}),(G,\mathbf{f}))$. We conducted the experiment multiple times. Figure 4 and Figure 5 show the results on MUTAG when varying the number of dimensions and the number of layers, respectively. Figure 11 and Figure 12 show the results on PROTEINS when varying the number of dimensions and the number of layers, respectively." (Page 43, Lines 2302-2314)
>
> - We also added interpretation of an additional version of the experiment that depicts Corollary 12 more closely.
>
> "**Correlation between DIDM model's distance and maximal MPNN distance.**
>
> Corollary 12 states that "convergence in DIDM mover's distance" is equivalent to "convergence in the MPNN's output  *for all MPNNs*." The previous experiment depicts Corollary 12 only vaguely, since the experiment uses a single MPNN, and does not check the output distance for all MPNNs. Instead, in this experiment we would like to depict the "for all" part of Corollary 12 more closely. Since one cannot experimentally apply all MPNNs on a graph,  we instead randomly choose 100 MPNNs for the whole dataset. Denote the set containing the 100 MPNNs by $\mathcal{N}'$. To verify the "for all" part, given each pair of graphs, we evaluate the distance between the MPNN's outputs on the two graphs for each MPNN and return the maximal distance. We plot this maximal distance  against the DIDM mover's distance. Namely, we plot
>
> $\max_{\mathfrak{H}\in\mathcal{N}'}||\mathfrak{H}($*GIN\_meanpool*$,G_i) - \mathfrak{H}($*GC\_meanpool*$,G)||_2$
>
> and  $\max_{\mathfrak{H}\in\mathcal{N}'}||\mathfrak{H}($*GIN\_meanpool*$,G_i)- \mathfrak{H}($*GC\_meanpool*$,G)||_2$
>
> against $\delta_{\rm DIDM}^2(G_i,G)$.
>
> Note that in each experiment, we initialize *GIN\_meanpool* and *GC\_meanpool* only once with random weights and then compute the hidden representations of all graphs.
>
> In more details, we checked the extent to which $\delta_{\rm DIDM}^2$ correlates with the maximal distance of 100 MPNNs' vectorial representation distances on MUTAG dataset and marked the Lipschitz relation. Here, we randomly generated 100 MPNNs for the entire dataset. Figure 3 showcase different Lipschitz relation. Note that the results are normalized.
>
> From this, one can estimate a bound on the  Lipschitz constants of all MPNNs from the family." (Pages 43-44 Lines 2315-2342)
>
> - We added a conclusion to the above experiments, summarized as a conjecture
>
> "**The Random MPNN Distance conjecture.** Observe that in our experiments we plotted the MPNN's output distance for random MPNNs, not for "all MPNNs," and still got a nice correlation akin to Corollary 12. This leads us to the hypothesis that randomly initialized MPNNs have a fine-grained expressivity property: for some distribution over the space of MPNNs, a sequence of graph-signals converges in DIDM mover's distance if and only if the output of the sequence under a random MPNN converges in high probability. See Figure 10 for a comparison of $\delta_{\rm DIDM}^2$'s correlation with the maximal distance of 100 MPNNs' vectorial representation distances, with $\delta_{\rm DIDM}^2$'s correlation with the mean distance of 100 MPNNs' vectorial representation distances, and with $\delta_{\rm DIDM}^2$'s correlation with a single MPNN's vectorial representation distances on MUTAG dataset." (page 44, Lines 2334 - 2342)

---

> ### Author Response · Authors · 2024-11-24
> **Rebuttal Revisions**
>
> - Regarding generalization, we added a comment referring to relevant generalization experiments.
>
> "...First, experimentally, in [7: Tables 1, 3] and [10: Table 2], it is shown that MPNNs with normalized sum aggregation generalize well, comparably to MPNNs with sum aggregation...." (Page 10 Lines 525-528)
>
> - We added a discussion which emphasizes that the experiments are proof-of-concept experiments, designed to support the *theoretical* results.
>
> "The goal of our experiments is to support the theoretical results which formulate a form of equivalence between GNN outputs and DIDM mover's distance. We emphasize that our proposed DIDM mover's distance metric is mainly a tool for theoretical analysis and the proposed experiments are not designed to compete with state-of-the-art methods. Although our metric is not intended to be used directly as a computational tool, our results suggest that we can roughly approximate the DIDM mover's distance between two graphs by the Euclidean distance between their outputs under random MPNNs. This Euclidean distance can be used in practice as it is less computationally expensive than DIDM mover's distance." (Page 41 Lines 2192-2199)
>
> "As a proof of concept, we empirically test the correlation between $\delta_{\rm DIDM}^L$ and distance in the output of MPNNs. We hence chose well-known and simple MPNN architectures, varying the hidden dimensions and number of layers. We do not claim that GIN [19] and GraphConv [20] are representative of the variety of all types of MPNNs. Nevertheless, they are proper choices for demonstrating our theory in practice." (Page 42, Lines 2247-2251)
>
> - Overall, in Appendix M (Pages 41-50), we added more details about the experiments to make them more understandable and transparent to the reader.

---

> ### Author Response · Authors · 2024-11-24
> **Rebuttal Revisions**
>
> >>Question 3. Structuring the manuscript to more prominently summarize the major findings and their impact would improve readability. Perhaps revise the contribution part in the introduction to better summarize the major findings. Also, perhaps adding a section to state the broader impact of these findings more clearly and how each finding advances the field of GNN theory to better illustrate the importance of this work.
>
> **Response.** Thank you for this comment. We extensively revised the exposition and reorganized some parts of the paper.
>
> To emphasize the potential impact of our work and the motivation behind it, we added the following sentences.
>
> - We extended and modified the contributions' subsection as follows:
>
> "**Contributions.** We propose the first metric for attributed graphs under which the space of attributed graphs is compact and MPNNs are Lipschitz continuous and separate points. Our construction leads to the first theory of MPNNs that unifies expressivity, universality, and generalization on any data distribution of attributed graphs. In detail:
>
>  - We show a *fine-grained* metric version of the separation power of MPNNs, extending the results of [3] to attributed graphs}: two graph-signals are close in our metric if and only if the outputs of all MPNNs on the two graphs are close. Hence, the geometry of graphs (with respect to our metric) is equivalent to the geometry of the graphs' representations via MPNNs (with respect to the Euclidean metric).
>
>  - We prove that the space of attributed graphs with our metric is compact and MPNNs are Lipschitz continuous (and separate points). This leads to two theoretical applications: (1) A universal approximation theorem, i.e., MPNNs can approximate any continuous function over the space of attributed graphs. (2) A generalization bound for MPNNs, akin to robustness bounds [4], requiring no assumptions on the data distribution or the number of GNN weights." (Page 2 Lines 89-102)
>
> - To restate the broader impact of our findings more clearly, we revised Section 6 (Discussion).
>
> " MPNNs were historically defined constructively, as specific types of computations on graphs, without a proper theory of MPNN function spaces over properly defined domains of definition. Our work provides a comprehensive *functional basis* for MPNNs, which elegantly leads to *machine learning results* like universal approximation and generalization for attributed graphs. " (Page 10, Lines 519-522)
>
> - We list here some additional sentences that we added to the paper to make the motivation and contribution clearer.
>
> "In this paper, we define a pseudometric for attributed graphs which is highly related to the type of computation message passing GNNs (MPNNs) perform. Our  construction enables a unified analysis of expressivity, universal approximation and generalization of MPNNs on graphs with node attributes." (Page 1 Lines 44-47)
>
> "In this work, we close these gaps via a unified approach that allows for an analysis of expressivity, universal approximation, and generalization." (Page 2 Lines 76-77)
>
> - We updated Figure 1 to better exhibit the idea behind our metrics. (Page 5, Lines 244-250)
>
> - We changed the name of section 5 to "Theoretical Applications", as the previous "Main Results" was misleading. The results in Section 5 are better interpreted as applications of our main results, which are in Section 3,4. The main results are the compactness of the space of attributed graphs/DIDMs with respect to the DIDM mover's distance, the Lipschitz continuity of MPNNs with respect to the DIDM mover's distance, and the fine-grained expressivity (equivalency of convergence in DIDM mover's distance and in MPNN's outputs).
>
> "**5 Theoretical Applications**
> Next, we state the main theoretical applications of the compactness of the space of attributed graphs and the Lipschitz and separation power of MPNNs with respect to DIDM mover's distance." (Page 9, Lines 463 -466)

---

> ### Author Response · Authors · 2024-11-24
> **Rebuttal Revisions**
>
> >>>Question 4. The methods primarily focus on dense graphs, as noted in the conclusion that “sparse graphs are always considered close to the empty graph under our metrics.” A further explanation of this limitation—specifically, why sparse graphs exhibit this behavior under TMD—would benefit the reader. Additionally, a discussion of whether this limitation affects the method’s generality and its potential applications to sparse graph domains would strengthen the paper.
>
> **Response.**
> - We extended the explanation of this limitation in the revised version of the Section 6:
>
> "Our theory is meaningful only for dense graphs, as sparse graphs are always considered close to the
> empty graph under our metrics (see Appendix B.1). Future work may focus on deriving fine-grained expressivity analyses for sparse attributed graphs." (Page 10, Lines 532-534)
>
> - We extended the explanation of this limitation in the revised version of the appendix, in *Appendix B.1. The Limitation of Our Construction to Dense Graphs*:
>
> "**B.1 The Limitation of Our Construction to Dense Graphs**
> For sparse graphs, the number $E$ of edges  is much smaller than the number $N^2$ of vertices squared. As a result, the induced graphon is supported on a set of small measure. Since graphon are bounded by $1$, this means that induced graphons from sparse graphs are close in $L_1([0,1]^2)$ to the $0$ graphon.
>
> Therefore, we can only use our theory for datasets of graphs that roughly have the same sparsity level $S\in\mathbb{N}_0$, i.e., $N^2/E$ is on the order of some constant $S$ for most graphs in the dataset. For this, one can scale our distance by $S$, making it appropriate to graphs with $E \ll N^2$ edges, in the sense that the graphs will not all be trivially close to $0$. Our theory does not solve the problem of sequences of graph asymptotically converging to $0$.
>
> In future work, one may develop a fine-grained expressivity theory based sparse graph limit theories. There are several graph limit theories that apply to sparse graphs, including Graphing theory [20], Benjamini–Schramm limits [11,12,13], stretched graphons [14,15], $L^p$ graphons [16,17], and graphop theory [18], which extends all of the aforementioned
> approaches . Future work may extend our theory to sparse graph limits." (Page 21, Lines 1091-1109)
>
> Nevertheless, graphons are a well used geometric deep learning analysis tool, see e.g. [3,7,10,11,19].
>
> **References**
>
> [1] Oono, K. and Suzuki, T. Optimization and generalization analysis of transduction through gradient boosting and application to multi-scale graph neural networks. NeurIPS.
>
> [2] Li et al., 2022. Generalization guarantee of training graph convolutional networks with graph topology sampling. ICML.
>
> [3] Tang et al., 2023. Towards understanding the generalization of graph neural networks. ICML 2023.
>
> [4] Chuang, et al., 2022. Tree Mover’s Distance: Bridging Graph Metrics and Stability of Graph Neural Networks. NeurIPS.
>
> [5] Samantha et al. "The Weisfeiler-Lehman distance: Reinterpretation and connection with gnns." Topological, Algebraic and Geometric Learning Workshops 2023. PMLR.
>
> [6] Samantha et al. 2022. Weisfeiler-lehman meets gromov-Wasserstein. International Conference on Machine Learning. PMLR.
>
> [7] Böker, et al., 2023. Fine-Grained Expressivity of Graph Neural Networks. NeurIPS.
>
> [8] Xu, et al., 2019. How Powerful are Graph Neural Networks? ICLR.
>
> [9] Morris, et al., 2019. Weisfeiler and Leman Go Neural: Higher-order Graph Neural Networks. AAAI Conference on Artificial Intelligence.
>
> [10] Levie, 2023. A Graphon-signal Analysis of Graph Neural Networks. NeurIPS.
>
> [11] Abért, et al., 2014. Benjamini-Schramm convergence and pointwise convergence of the spectral measure.
>
> [12] Bollobás, et al., 2011. Sparse Graphs: Metrics and Random Models. Random Structures & Algorithms.
>
> [13] Hatami, et al., 2014 Limits of locally–globally convergent graph sequences. Geometric and Functional Analysis.
>
> [14] Xingchao, et al., 2023. Generalized Graphon Process: Convergence of Graph Frequencies in Stretched Cut Distance.
>
> [15] Feng, et al., 2024. Modeling Sparse Graph Sequences and Signals Using Generalized Graphons. IEEE Transactions on Signal Processing.
>
> [16] Borgs, et al., 2014. An $L^p$ theory of sparse graph convergence I: Limits, sparse random graph models, and power law distributions. Transactions of the American Mathematical Society
>
> [17] Borgs, et al., 2014. An $L^p$ theory of sparse graph convergence II: LD convergence, quotients, and right convergence. The Annals of Probability.
>
> [18] Backhausz et al., 2020. Action convergence of operators and graphs. Canadian Journal of Mathematics.
>
> [19] Ruiz, et al., 2020. Graphon Neural Networks and the Transferability of
> Graph Neural Networks. NeurIPS.
>
> [20] Lovász, 2020. Compact graphings. Acta Mathematica Hungarica.

---

> ### Comment · Reviewer_uw7j · 2024-11-26
>
> Thank you for the insightful and thorough responses, which address my concerns. I would like to raise my score.

---

> > ### Author Response · Authors · 2024-11-29
> >
> > We thank the reviewer again for their thorough and constructive comments. Their suggestions have significantly improved our paper.

---

### Official Review · Reviewer_Ku8V · 2024-11-02

**Soundness:** 4
**Presentation:** 2
**Contribution:** 3
**Rating:** 8
**Confidence:** 4

**Summary:**

The authors propose a metric based on optimal transport distances between graph representations.
Under this metric, the authors show the MPNNs are Lipschitz and graph rerpesentations are compact. They establish that GNNs are capable of separating attributed graphs that are distant in the defined metric, leading to a universal approximation theorem and generalization bounds for MPNNs. Building on these, the authors further developed a generalisation theorem.  Empirical results demonstrate the correlation between the new metric and MPNN output.

**Strengths:**

* The proposed a noval pseudometric for graphs based on iterated degree measures and optimal transport.
* The authors show MPNNs are universal and Lipschitz under this pseudometric.
* The authors show a bidirectional connections between the metric and MPNN output.
* A generalisation theorem is developed based the metric.
* The framework unifies MPNNs on graph and graphon.

**Weaknesses:**

* The proposed metric only applies to MPNN/1-WL GNNs, and cannot be used to analyse more powerful GNNs.
* Related works are insufficiently discussed. In particualr, the proposed metric seems realted to [1][2], but this connection has not been discussed.
* The metric is expensive to compute, while its complexity is linear to $L$, it is in the order of $n^5$ which hinders its practicality
* The MPNN analysis limits to normalized sum aggregation, which is less commonly used.
* There are too many similar notations and symbols, making it difficult to follow.
* The graphon-based approach means the results do not apply to sparse graphs.
* Emperiments are limited to one small real-world dataset (MUTAG), which makes the results less convincing.




[1] Chen, Samantha, et al. "The Weisfeiler-Lehman distance: Reinterpretation and connection with gnns." Topological, Algebraic and Geometric Learning Workshops 2023. PMLR, 2023.
[2] Chen, Samantha, et al. "Weisfeiler-lehman meets gromov-Wasserstein." International Conference on Machine Learning. PMLR, 2022.

**Questions:**

1. The proposed metric considers attributed graphs, but in Fig 2, why is the alignment worsen when added attributes?
2. Can the results be used to analyse more power GNNs? e.g. k-GNN, F-MPNN, etc
3. Unifying graph and graphon MPNN means you have to assume normalised sum aggregation, can this assumption be removed?
4. If not considering graphons, can this results apply to sparse graphs?
5. How is the metric related to the metric in "Weisfeiler-lehman meets gromov-Wasserstein."?
6. Why do you use Unbalance OT instead of OT?

---

> ### Author Response · Authors · 2024-11-24
> **Rebuttal Revisions**
>
> We thank the reviewer for their positive assessment of our paper and their insightful comments.
> *For the convenience of the reviewer, we merge related comments. In our responses, we quote any relevant revision we made in the paper. This led to a seemingly longer rebuttal, but we believe that it will streamline the work for the reviewer.*
>
>
> >**Weaknesses and Questions**
> >>Weakness 1. The proposed metric only applies to MPNN/1-WL GNNs, and cannot be used to analyse more powerful GNNs. >
>
> >>Question 2. Can the results be used to analyse more power GNNs? e.g. k-GNN, F-MPNN, etc
>
> **Response.** This is correct. More powerful GNNs will not be continuous with respect to our pseudo metrics, as such GNNs separate graphs within equivalence classes of graphs of zero distance. More than that, powerful GNNs will not even be functions (over equivalence classes of graphs with zero distance), as different representatives from the same class can map to different values under a powerful GNN. We slightly edited the text to clarify that the theory is designed to analyse MPNNs.
>
> "In this paper, we define a pseudometric for attributed graphs, which is highly related to the type of computation message passing GNNs (MPNNs) perform. This makes it suitable to enable a unified analysis of expressivity, universal approximation and generalization of MPNNs on graphs with node attributes. " (Page 1, Lines 44-47)
>
> >>Weakness 2. Related works are insufficiently discussed. In particualr, the proposed metric seems realted to [1],[2], but this connection has not been discussed.
>
> >>Question 5. How is the metric related to the metric in "Weisfeiler-lehman meets gromov-Wasserstein."?
>
> **Response.** In the new version of the paper, we extended the discussion in the introduction about the connection to [1,2], as suggested.
>
> "[1,2] prove that MPNNs are Lipschitz continuous over the Weisfeiler-Lehman (WL) distance between hierarchies of probability measures. Since the space of graphs is not compact under their metric, they achieve universal approximation by limiting the analysis to an arbitrary compact subspace." (Page 2, Lines 61-64)
>
> "Inspired by [2], [3] took..." (Page 2, Lines 65-66)
>
> "Similarly, hierarchical structures of measures have been used for the analysis of MPNNs [1,2]..." (Page 2, Lines 108-109)

---

> > ### Author Response · Authors · 2024-11-24
> > **Rebuttal Revisions**
> >
> > >>Weakness 3. The metric is expensive to compute, while its complexity is linear to $L$, it is in the order of $n^5$ which hinders its practicality
> >
> >  **Response.** We agree that the metric is expensive to compute in practice. However, the main application of our theory is proving both universal approximation and generalization theorems. Our metric is not intended to be used directly as a computational tool, but as a theoretical tool with the aim of helping us understand MPNNs better. Nevertheless, our results indicate that we can approximate our metric with random MPNNs outputs. This is less computationally expensive and can be used in practice.  To clarify these points, we revised the paper as follows:
> >
> >
> > a. We added a clarification for the goal of the experiments in the "Experimental Evaluation" appendix (Appendix M).
> >
> > "The goal of our experiments is to support the theoretical results which formulate a form of equivalence between GNN outputs and DIDM mover's distance. We emphasize that our proposed DIDM mover's distance metric is mainly a tool for theoretical analysis and the proposed experiments are not designed to compete with state-of-the-art methods. Although our metric is not intended to be used directly as a computational tool, our results suggest that we can roughly approximate the DIDM mover's distance between two graphs by the Euclidean distance between their outputs under random MPNNs. This Euclidean distance can be used in practice as it is less computationally expensive than DIDM mover's distance." (Page 41 Lines 2192-2199)
> >
> > b. We added a clarification for the goal of our theory in the revised version of the introduction.
> >
> > "In this paper, we define a pseudometric for attributed graphs which is highly related to the type of computation message passing GNNs (MPNNs) perform. Our construction enables a unified analysis of expressivity, universal approximation and generalization of MPNNs on graphs with node attributes." (Page 1, Lines 44-47)
> >
> > "In this work, we close these gaps via a unified approach that allows for an analysis of expressivity, universal approximation, and generalization." (Page 2, Lines 76-77)
> >
> > c. We extended and modified the contributions' subsection as follows:
> >
> > "**Contributions.** We propose the first metric for attributed graphs under which the space of attributed graphs is compact and MPNNs are Lipschitz continuous and separate points. Our construction leads to the first theory of MPNNs that unifies expressivity, universality, and generalization on any data distribution of attributed graphs. In detail:
> >
> >  - We show a *fine-grained* metric version of the separation power of MPNNs, extending the results of [3] to attributed graphs}: two graph-signals are close in our metric if and only if the outputs of all MPNNs on the two graphs are close. Hence, the geometry of graphs (with respect to our metric) is equivalent to the geometry of the graphs' representations via MPNNs (with respect to the Euclidean metric)."
> >
> >  - We prove that the space of attributed graphs with our metric is compact and MPNNs are Lipschitz continuous (and separate points). This leads to two theoretical applications: (1) A universal approximation theorem, i.e., MPNNs can approximate any continuous function over the space of attributed graphs. (2) A generalization bound for MPNNs, akin to robustness bounds [4], requiring no assumptions on the data distribution or the number of GNN weights." (Page 2 Lines 89-102)
> >
> > d. We emphasize that the main applications of our construction are theoretical in Section 5:
> >
> > "**5 Theoretical Applications**
> > Next, we state the main theoretical applications of the compactness of the space of attributed graphs and the Lipschitz and separation power of MPNNs with respect to DIDM mover's distance." (Page 9 Lines 463-466)

---

> ### Author Response · Authors · 2024-11-24
> **Rebuttal Revisions**
>
> >>Weakness 4.The MPNN analysis limits to normalized sum aggregation, which is less commonly used.
>
> >>Question 3. Unifying graph and graphon MPNN means you have to assume normalised sum aggregation, can this assumption be removed?
>
> >>Question 6. Why do you use Unbalance OT instead of OT?
>
> **Response.** These are good questions that relate to each other.
>
> - We added the following discussion in the appendix (Appendix B.2):
>
> "**Comparison of Sum Mean, and Normalized Sum Aggregations**"
>
> "See Appendix A.3 for the definition of sum and mean aggregation on attributed graphs. First, (unnormalized) sum aggregation (Definition 19)  does not work in the context of our analysis. Indeed, given an MPNN that simply sums the features of the neighbors and given the sequence of complete graphs of size $N\in \mathbb{N}_0$, then, the output of the MPNN on these graphs diverges to infinity as $n\to\infty$. As a result, equivalency of the metric at the output space of MPNNs with a compact metric on the space of graphs is not possible.
>
> Another popular aggregation scheme is mean aggregation, defined canonically on graphon-signal as follows [7,8].
>
> **Definition 38** (Mean Aggregation on Graphon-signals). Let $\varphi$ be an $L$-layer MPNN model, $(W,f)$ be a graph-signal, and $t\in[L-1]$. Then mean aggregation of $(W,f)$'s $t$-level features with respect to the node $x \in V(W)$ is defined as
>
> $$Agg(W,\mathfrak{f}(\varphi,W,f)^{(t)}_-)(x)$$
>
> $$=\frac{1}{\int_{[0,1]}W(x,y)d\lambda(y)} \int_{[0,1]}W(x,y)\mathfrak{f}(\varphi,W,f)^{(t)}_y d\lambda(y).$$
>
> The theory could potentially extend to mean aggregation using two avenues. One approach is to do this under a limiting assumption: restricting the space to graphs/graphons with minimum node degree bounded from below by a constant. This is like an idea outlined in [8].
>
> A second option is to redefine $\delta_{\rm DIDM}$ using balanced OT. In this paper, $\delta_{\rm DIDM}$ highly relates to the type of computation MPNNs with normalized sum aggregation perform. We used unbalanced OT (Definition 1) as the basis of $\delta_{\rm DIDM}$ due to the fact that MPNNs with normalized sum aggregation do not average incoming messages, which means they can separate nodes of different degrees within a graph. MPNNs with mean aggregation, in contrast, do average incoming messages. Hence, an appropriate version of optimal transport, in this case, could be based on averaging. I.e., using balanced OT on normalized measures could serve as a base for defining metrics in the analysis of MPNNs with mean aggregation." (Pages 21-22, Lines 1110-1137)
>
> - We added the following paragraph in Section 6:
>
> "The only nonstandard part in our construction is the choice of  normalized sum aggregation in our MPNN architecture, while most MPNNs use sum, mean, or max aggregation. We justify this choice as follows. First, experimentally, in [3: Tables 1, 3] and [5: Table 2], it is shown that MPNNs with normalized sum aggregation generalize well, comparably to MPNNs with sum aggregation. We believe that this is the case since in most datasets most graphs have roughly the same order of vertices. Hence, the normalization by $N^2$ is of a constant order of magnitude and can be swallowed by the weights of the MPNN, mimicking the behavior of an MPNN with sum aggregation. Future work may explore extensions of our theory to other aggregation schemes (see Appendix A.3 and Appendix B.2 for more details)." (Page 10 Lines: 523-531)

---

> ### Author Response · Authors · 2024-11-24
> **Rebuttal Revisions**
>
> >>Weakness 5. There are too many similar notations and symbols, making it difficult to follow.
>
> **Response** We agree that the original version of the paper had problematic/unclear notations. We modified some of the terminology and changed many notations to improve readability. Now, overall, we have fewer versions of H in different fonts denoting different things, and we give more meaningful names to the different terms. We list here the important changes:
>
> - We changed the notation of computation IDMs and DIDMs from $\mathscr{H}$ and $\mathscr{G}$ to $\gamma$ and $\Gamma$, as we typically denote generic IDM and DIDMs with Greek letters. These new symbols are more easily distinguishable from the rest of the notations in the paper.
>
> "We call $\gamma_{(W,f),L}$ a *computation iterated degree measure (computation IDM)* of order-$L$ and $\Gamma_{(W,f),L}$ a *distribution of computation iterated degree measures (computation DIDM)* of order-$L$." (Page 6, Lines 274-276)
>
> - We changed the name of our main metrics from the tree distance to IDM distance and tree mover's distance to DIDM mover's distance. These names describe the construction underlying the metrics more precisely and prevent confusion with the tree distance and tree mover's distance from the paper [6] and the tree distance from [3]. We also changed the notation of the metrics from $\mathbf{TD}$ and $\mathbf{TMD}$ to $d_{\rm IDM}$ and $\delta_{\rm DIDM}$, respectively, to be more meaningful. We took inspiration from standard notations of cut metric and cut distance. (Page 6, Lines 298, 312)
>
> - We updated the MPNN features' notations, to make it easier to follow the dependencies of each feature on his corresponding MPNN model and to clarify the dependence of the features on the input object. To clarify the feature definition purpose, we also named "_ features" as the "application of MPNNs on _ ".
>
> Graph-signals features: Given an $L$-layer MPNN model with readout $(\varphi,\psi)$ and a graph-signal $(G,\mathbf{f})$, we now denote the hidden node representations as  $\mathfrak{g}_-^{(t)}$ at layer $t$, with $1\leq t \leq L$ and the graph-level output as $\mathfrak{G} \in \mathbb{R}^d$. To clarify the dependence of $\mathfrak{g}$ and $\mathfrak{G}$ on $(\varphi,\psi)$, $(G,\mathbf{f})$, and the node $v\in V(G)$, we often denote $\mathfrak{g}(\varphi)_v^{(t)}$ or $\mathfrak{g}(\varphi,G,\mathbf{f})_v^{(t)}$, and $\mathfrak{G}(\varphi,\psi)$  or $\mathfrak{G}(\varphi,\psi,G,\mathbf{f})$.
>
> Graphon-signals features: Given an $L$-layer MPNN model with readout $(\varphi,\psi)$ and a graphon-signal $(W,f)$, we now denote the hidden node representations as $\mathfrak{f}_-^{(t)}$ at layer $t$, with $1\leq t \leq L$ and the graphon-level output as $\mathfrak{F} \in \mathbb{R}^d$. To clarify the dependency of $\mathfrak{f}$ and $\mathfrak{F}$ on $(\varphi,\psi)$, $(W,f)$ and the node $x\in V(W)$, we often denote $\mathfrak{f}(\varphi)_x^{(t)}$ or $\mathfrak{f}(\varphi,W,f)_x^{(t)}$, and $\mathfrak{F}(\varphi,\psi)$  or $\mathfrak{F}(\varphi,\psi,W,f)$.
>
> IDMs and DIDMs features: Given an $L$-layer MPNN model with readout $(\varphi,\psi)$. We now denote the hidden node representations as $\mathfrak{h}_-^{(t)}$ at layer $t$, with $1\leq t \leq L$ and the graph-level output as $\mathfrak{H} \in \mathbb{R}^d$. To clarify the dependency of $\mathfrak{h}$ and $\mathfrak{H}$ on $(\varphi,\psi)$, the IDM $\tau$, and the DIDM $\nu$, we often denote $\mathfrak{h}(\varphi)_v^{(t)}$ or $\mathfrak{h}(\varphi,W,f)_v^{(t)}$, and $\mathfrak{H}(\varphi,\psi)$  or $\mathfrak{H}(\varphi,\psi,W,f)$.
>
> (Pages 7-8, Lines 365-374, 377-385, 388-397)
>
> - We unify indexes and other notations across the paper. For example, in Theorem 6, we changed the notation of $max(|V(G)|,|V(H)|)$ from n to N, to fit our convention to make the number of nodes with a capitalized N rather than a noncapitalized N.
>
> "**Theorem 6.** For any fixed $L\in \mathbb{N}$, $\mathbf{TMD}^L$ between any two graph-signals $(G,\mathbf{f})$ and $(H,\mathbf{g})$ can be computed in time polynomial in $L$ and the size of $G$ and $H$, namely $O(L\cdot N^5\log(N))$ where $N= \max(|V(G)|,|V(H)|)$." (Page 6, Lines 319-321)
>
> - We changed the notation of the generic Lipschitz continuous function in Theorem 16 from $\mathfrak{L}$ to $\mathfrak{M}$, as the letter L was too heavily used in this paper (with different fonts denoting different things). (Page 10, Lines 498-507)
>
> - We added a section called "List of Notations" at the end of the appendix as Appendix N. (Page 54, Line 2700)
>
> - We changed the notation of the set of natural numbers from $\mathbb{N}$ to $\mathbb{N}_0$ to emphasize we include 0 in it. (Page 51, Line 2709)

---

> > ### Author Response · Authors · 2024-11-24
> > **Rebuttal Revisions**
> >
> > >>Weakness 6. The graphon-based approach means the results do not apply to sparse graphs.
> >
> > >>Question 4. If not considering graphons, can this results apply to sparse graphs?
> >
> > **Response.** This is correct. To clarify this, we offer the following discussion in Appendix *B.1 The Limitation of Our Construction to Dense Graphs* in the revised paper:
> >
> > "**B.1 The Limitation of Our Construction to Dense Graphs**
> > For sparse graphs, the number $E$ of edges  is much smaller than the number $N^2$ of vertices squared. As a result, the induced graphon is supported on a set of small measure. Since graphon are bounded by $1$, this means that induced graphons from sparse graphs are close in $L_1([0,1]^2)$ to the $0$ graphon.
> >
> > Therefore, we can only use our theory for datasets of graphs that roughly have the same sparsity level $S\in\mathbb{N}_0$, i.e., $N^2/E$ is on the order of some constant $S$ for most graphs in the dataset. For this, one can scale our distance by $S$, making it appropriate to graphs with $E \ll N^2$ edges, in the sense that the graphs will not all be trivially close to $0$. Our theory does not solve the problem of sequences of graph asymptotically converging to $0$.
> >
> > In future work, one may develop a fine-grained expressivity theory based sparse graph limit theories. There are several graph limit theories that apply to sparse graphs, including Graphing theory [10], Benjamini–Schramm limits [11,12,13], stretched graphons [14,15], $L^p$ graphons [16,17], and graphop theory [18], which extends all of the aforementioned
> > approaches . Future work may extend our theory to sparse graph limits." (Page 21, Lines 1091-1109)
> >
> > Nevertheless, graphons are a well used geometric deep learning analysis tool, see e.g. [3,5,9].

---

> ### Author Response · Authors · 2024-11-24
> **Rebuttal Revisions**
>
> >>Weakness 7. Experiments are limited to one small real-world dataset (MUTAG), which makes the results less convincing.
>
> >>Question 1. The proposed metric considers attributed graphs, but in Fig 2, why is the alignment worsen when added attributes?
>
> **Response.** Question 1 is a good question. As we interpret this result, it could be an artifact of using random noise as signal in our experiments. The specific MPNNs we used, have either a linear activation or a ReLU activation function. Thus, they have a "close to linear" averaging effect on the signal, which cancels in a sense the contribution of the noise signal to the output of the MPNN, while the metric takes the signal into full consideration. This leads to a noisy correlation.
> To test this hypothesis, we ran a modified experiment: instead of taking random noise as signal, we generated a "smoother" signal as follows. We conducted an additional version of the correlation experiment with a signal which has a different constant value on each community of the graph sampled from the SBM. Each value is randomly sampled from a uniform distribution over $[0,1]$. This gave a much cleaner correlation than the original noise-signal experiment.
>
> We moved the old experiment to the appendix, as it is less reflective of our theory. Instead, we replaced it by a new experiment where the signal is constant in each community.
>
> "..We conducted the experiment twice, once with a constant feature for all nodes and once with a signal which has a different constant value on each community of the graph. Each of these two values is randomly sampled from a uniform distribution over $[0,1].$ Figure 2, shows the results when varying the hidden dimension of the GNN."(Page 9, Lines 441-445)
>
> We added more details on these experiments in Appendix M, which is in pages 41-50. There, we interpret and compare the old experiment with the new.
>
> " ...The results still show a  correlation between input distance and MPNN outputs, but with a higher variance. As we interpret this result, the increased variance could be an artifact of using random noise as signal in our experiments. The specific MPNNs we used, have either a linear activation or a ReLU activation function. Thus, they have a "linear" averaging effect on the signal, which cancels in a sense the contribution of the noise signal to the output of the MPNN, while the metric takes the signal into full consideration. This leads to a noisy correlation." (Page 43, Lines 2293-2299)
>
> More details on the experiment are in the appendix (Appendix M.2 page 42-44).
>
> - We added another correlation experiment on PROTEINS database as well as further clarified the setting. See Page 43, Lines 2305 - 2315.
>
> - We also added interpretation of an additional version of the experiment that depicts Corollary 12 more closely.
>
> "**Correlation between DIDM model's distance and maximal MPNN distance.**
>
> Corollary 12 states that "convergence in DIDM mover's distance" is equivalent to "convergence in the MPNN's output  *for all MPNNs*." The previous experiment depicts Corollary 12 only vaguely, since the experiment uses a single MPNN, and does not check the output distance for all MPNNs. Instead, in this experiment we would like to depict the "for all" part of Corollary 12 more closely. Since one cannot experimentally apply all MPNNs on a graph,  we instead randomly choose 100 MPNNs for the whole dataset. Denote the set containing the 100 MPNNs by $\mathcal{N}'$. To verify the "for all" part, given each pair of graphs, we evaluate the distance between the MPNN's outputs on the two graphs for each MPNN and return the maximal distance. We plot this maximal distance  against the DIDM mover's distance. Namely, we plot
>
> $\max_{\mathfrak{H}\in\mathcal{N}'}||\mathfrak{H}($*GIN\_meanpool*$,G_i) - \mathfrak{H}($*GC\_meanpool*$,G)||_2$
>
> and  $\max_{\mathfrak{H}\in\mathcal{N}'}||\mathfrak{H}($*GIN\_meanpool*$,G_i)- \mathfrak{H}($*GC\_meanpool*$,G)||_2$
>
> against $\delta_{\rm DIDM}^2(G_i,G)$.
>
> Note that in each experiment, we initialize *GIN\_meanpool* and *GC\_meanpool* only once with random weights and then compute the hidden representations of all graphs.
>
> In more details, we checked the extent to which $\delta_{\rm DIDM}^2$ correlates with the maximal distance of 100 MPNNs' vectorial representation distances on MUTAG dataset and marked the Lipschitz relation. Here, we randomly generated 100 MPNNs for the entire dataset. Figure 3 showcase different Lipschitz relation. Note that the results are normalized.
>
> From this, one can estimate a bound on the  Lipschitz constants of all MPNNs from the family."(Pages 43-44 Lines 2315-2342)

---

> ### Author Response · Authors · 2024-11-24
> **Rebuttal Revisions**
>
> - We added a conclusion to the above experiments, summarized as a conjecture
>
> "**The Random MPNN Distance conjecture.** Observe that in our experiments we plotted the MPNN's output distance for random MPNNs, not for "all MPNNs", and still got a nice correlation akin to Corollary 12. This leads us to the hypothesis that randomly initialized MPNNs have a fine-grained expressivity property: for some distribution over the space of MPNNs, a sequence of graph-signals converges in DIDM mover's distance if and only if the output of the sequence under a random MPNN converges in high probability. See Figure 10 for a comparison of $\delta_{\rm DIDM}^2$'s correlation with the maximal distance of 100 MPNNs' vectorial representation distances, with $\delta_{\rm DIDM}^2$'s correlation with the mean distance of 100 MPNNs' vectorial representation distances, and with $\delta_{\rm DIDM}^2$'s correlation with a single MPNN's vectorial representation distances on MUTAG dataset." (page 44, Lines 2334 - 2342)
>
> - Regarding generalization, we added a comment referring to relevant generalization experiments.
>
> "...First, experimentally, in [7: Tables 1, 3] and [10: Table 2], it is shown that MPNNs with normalized sum aggregation generalize well, comparably to MPNNs with sum aggregation...."(Page 10 Lines 525-528)
>
> - We added a discussion which emphasizes that the experiments are proof-of-concept experiments, designed to support the *theoretical* results.
>
> "The goal of our experiments is to support the theoretical results which formulate a form of equivalence between GNN outputs and DIDM mover's distance. We emphasize that our proposed DIDM mover's distance metric is mainly a tool for theoretical analysis and the proposed experiments are not designed to compete with state-of-the-art methods. Although our metric is not intended to be used directly as a computational tool, our results suggest that we can roughly approximate the DIDM mover's distance between two graphs by the Euclidean distance between their outputs under random MPNNs. This Euclidean distance can be used in practice as it is less computationally expensive than DIDM mover's distance." (Page 41 Lines 2192-2199)
>
> "As a proof of concept, we empirically test the correlation between $\delta_{\rm DIDM}^L$ and distance in the output of MPNNs. We hence chose well-known and simple MPNN architectures, varying the hidden dimensions and number of layers. We do not claim that GIN [19] and GraphConv [20] are representative of the variety of all types of MPNNs. Nevertheless, they are proper choices for demonstrating our theory in practice."(Page 42, Lines 2247-2251)
>
> - Additionally, in Appendix M (Pages 41-50), we added more details about the experiments to make them more understandable and transparent to the reader. For a  longer discussion about these changes, see our response to questions 1 and 2 of reviewer uw7j.

---

> ### Author Response · Authors · 2024-11-24
> **Rebuttal Revisions**
>
> **References.**
>
> [1] Samantha et al. The Weisfeiler-Lehman distance: Reinterpretation and connection with gnns. Topological, Algebraic and Geometric Learning Workshops 2023. PMLR.
>
> [2] Samantha et al. "Weisfeiler-lehman meets gromov-Wasserstein." International Conference on Machine Learning. PMLR.
>
> [3] Böker, et al., 2023. Fine-Grained Expressivity of Graph Neural Networks. NeurIPS.
>
> [4] Xu, et al., 2012. Robustness and generalization. COLT.
>
> [5] Levie, 2023. A Graphon-signal Analysis of Graph Neural Networks. NeurIPS.
>
> [6] Chuang, et al., 2022. Tree Mover’s Distance: Bridging Graph Metrics and Stability of Graph Neural Networks. NeurIPS.
>
> [7] Maskey, et al., 2022. Generalization analysis of message passing neural networks on large random graphs. NeurIPS.
>
> [8] Maskey, et al., 2024. Generalization Bounds for Message Passing Networks on Mixture of Graphons.
>
> [9] Ruiz, et al., 2020. Graphon Neural Networks and the Transferability of Graph Neural Networks. NeurIPS.
>
> [10] Lovász, 2020. Compact graphings. Acta Mathematica Hungarica.
>
> [11] Abért, et al., 2014. Benjamini-Schramm convergence and pointwise convergence of the spectral measure.
>
> [12] Bollobás, et al., 2011. Sparse Graphs: Metrics and Random Models. Random Structures & Algorithms.
>
> [13] Hatami, et al., 2014 Limits of locally–globally convergent graph sequences. Geometric and Functional Analysis.
>
> [14] Xingchao, et al., 2023. Generalized Graphon Process: Convergence of Graph Frequencies in Stretched Cut Distance.
>
> [15] Feng, et al., 2024. Modeling Sparse Graph Sequences and Signals Using Generalized Graphons. IEEE Transactions on Signal Processing.
>
> [16] Borgs, et al., 2014. An $L^p$ theory of sparse graph convergence I: Limits, sparse random graph models, and power law distributions. Transactions of the American Mathematical Society
>
> [17] Borgs, et al., 2014. An $L^p$ theory of sparse graph convergence II: LD convergence, quotients, and right convergence. The Annals of Probability.
>
> [18] Backhausz et al., 2020. Action convergence of operators and graphs. Canadian Journal of Mathematics.
>
> [19] Xu, et al., 2019. How Powerful are Graph Neural Networks? ICLR.
>
> [20] Morris, et al., 2019. Weisfeiler and Leman Go Neural: Higher-order Graph Neural Networks. AAAI Conference on Artificial Intelligence.

---

> > ### Comment · Reviewer_Ku8V · 2024-11-26
> > **Thanks for the detailed response**
> >
> > I would like to thank the reviewers for the very detailed response. My questions are generally adequately answered. The paper is insightly and I would like to to seen it accepted. I will keep my current positive score, as the mentioned limitations around assumptions and constraints still exist.

---

> > > ### Author Response · Authors · 2024-11-29
> > > **Thanks**
> > >
> > > We thank the reviewer again for their thorough and constructive comments. Their suggestions have significantly improved our paper.

---

### Meta-Review · Area_Chair_M2zF · 2024-12-19

**Metareview:**

This paper proposes a new pseudo-metric on message passing GNNs on attributed graphs, based on the Wasserstein distance between distributions of computation tree analogs over graphons. The paper proves that message passing GNNs are universal approximations over attributed graphons. A generalization capability bound is derived exploiting the compactness and the Lipschitzness properties of the model. A experimental evaluation is proposed.


Strengths:
- a new pseudometric metric on graphs based on optimal transport,
- message passing GNNs are universal and lipschitz,
- generality of the framework,
- solid theoretical work with a generalization bound,
- unifies generalization and expressivity for GNN.

Weaknesses:
- framework limited to a subclass of GNN and better suited to dense graph,
- the metric is costly,
- experiments limited,
- discussion with related work could be improved,
- the presentation can be improved.

During the rebuttal, authors have provided multiple answer to the issues raised by the reviewers.
The reviewers have acknowledged the various answers provided and they were all satisfied with the answers.

Overall, there was a consensus for saying that the paper must be accepted.
I propose then acceptance.

I encourage the authors to include in the final version all the comments provided by the reviewers.

**Additional Comments On Reviewer Discussion:**

Reviewers xDjn and Ku8V were very positives and support the paper acceptance with a score of 8.
Reviewers uw7j was satisfied with answers that addressed his concerns and wanted to increase his score, final score is 6.
Reviewer 9Cqp acknowledged the answers and increased his score to 6.
Overall, reviewers were unanimous for acceptance.

---

### Decision · Program_Chairs · 2025-01-22

Accept (Poster)